# Active transcription and epigenetic reactions synergistically regulate meso-scale genomic organization

Aayush Kant [1,2], Zixian Guo[1,3], Vinayak Vinayak[1,2], Maria Victoria Neguembor [4], Wing Shun Li[5,6], Vasundhara Agrawal[6,7], Emily Pujadas[6], Luay Almassalha [6,8], Vadim Backman [6,7], Melike Lakadamyali [1,9], Maria Pia Cosma [4,10,11] & Vivek B. Shenoy [1,2,3] ✉

In interphase nuclei, chromatin forms dense domains of characteristic sizes, but the influence of transcription and histone modifications on domain size is not understood. We present a theoretical model exploring this relationship, considering chromatin-chromatin interactions, histone modifications, and chromatin extrusion. We predict that the size of heterochromatic domains is governed by a balance among the diffusive flux of methylated histones sustaining them and the acetylation reactions in the domains and the process of loop extrusion via supercoiling by RNAPII at their periphery, which contributes to size reduction. Super-resolution and nano-imaging of five distinct cell lines confirm the predictions indicating that the absence of transcription leads to larger heterochromatin domains. Furthermore, the model accurately reproduces the findings regarding how transcription-mediated supercoiling loss can mitigate the impacts of excessive cohesin loading. Our findings shed light on the role of transcription in genome organization, offering insights into chromatin dynamics and potential therapeutic targets.

The three-dimensional organization of chromatin within the nucleus is key to understanding the biophysical origin of critical cellular activities ranging from cell fate decisions to migration, proliferation, and metabolism. The existence of a multiscale chromatin organization has been observed not only from sequencing and contact-mapping techniques[1,2], but also super-resolution imaging[3–8]. At the microscale, chromatin is organized into transcriptionally distinct compartments – a transcriptionally active, loosely packed euchromatin phase and a tightly packed, predominantly silent heterochromatin phase. Finer resolution of the chromatin conformation reveals the existence of a more detailed spatial organization ranging from self-interacting topologically associated domains (TADs) to chromatin loops, a feature of the chromatin polymer mediating direct contact between gene regulatory elements bound by the CCCTC-binding factor (CTCF) and the cohesin complex[1,2]. The chromatin fibers can be trapped in and pushed through cohesin rings via a process called loop extrusion, until either CTCF bound sites are encountered or cohesin is unloaded[9–11]. In addition to direct extrusion of DNA loops via cohesin motor activity[12–19], RNA polymerase II (RNAP II), a protein complex essential for DNA transcription, has been identified to play a significant role in

[1]Center for Engineering Mechanobiology, University of Pennsylvania, Philadelphia, PA 19104, USA. [2]Department of Materials Science and Engineering, University of Pennsylvania, Philadelphia, PA 19104, USA. [3]Department of Mechanical Engineering and Applied Mechanics, University of Pennsylvania, Philadelphia, PA 19104, USA. [4]Centre for Genomic Regulation (CRG), The Barcelona Institute of Science and Technology, 08003 Barcelona, Spain. [5]Department of Applied Physics, Northwestern University, Evanston, IL 60208, USA. [6]Center for Physical Genomics and Engineering, Northwestern University, Evanston, IL 60202, USA. [7]Department of Biomedical Engineering, Northwestern University, Evanston, IL 60208, USA. [8]Department of Gastroenterology and Hepatology, Northwestern Memorial Hospital, Chicago, IL 60611, USA. [9]Department of Physiology, Perelman School of Medicine, University of Pennsylvania, Philadelphia, PA 19104, USA. [10]ICREA, Barcelona 08010, Spain. [11]Universitat Pompeu Fabra (UPF), Barcelona 08003, Spain. ✉e-mail: vshenoy@seas.upenn.edu

enabling the movement of the chromatin fiber resulting in chromatin loop extrusion through cohesin[19,20]. Specifically, by altering the DNA winding thereby supercoiling it, transcriptional activity has been proposed to play a role in in-vivo chromatin loop extrusion[19,20]. A recent experimental study combined super-resolution imaging of chromatin and single molecule tracking of cohesin with various biological perturbations, such as pharmacological and genetic inhibition of transcription, supercoiling, and loop extrusion. This approach provided compelling evidence that transcription-mediated supercoiling regulates loop extrusion, as well as the spatial organization of chromatin within the nucleus[19]. These observations present a promising avenue of crosstalk between chromatin's multiscale structural organization and its transcriptional status. This indicates that a bi-directional coupling exists, such that not only do the distinct phases of chromatin organization regulate transcription, but transcriptional activity can also affect genome organization via chromatin tethering, extrusion, and decompaction[19,21]. While the local microscopic effects of transcription on spatial DNA organization have been previously investigated, a fundamental quantitative understanding of the physical mechanisms involved in the global genomic organization, due to transcriptional and epigenetic regulation, is not yet fully understood.

Here, we propose a mesoscale coarse-grained, polymer physics-based mathematical model to capture the formation of chromatin domains while incorporating the spatiotemporal role of transcription-driven chromatin extrusion kinetics. Chromatin-chromatin interactions establish an energy landscape which drives a separation of hetero- and euchromatin phases. The dynamics of this evolution are governed by the diffusion of nucleoplasm and epigenetic reactions. Such evolution leads to the formation of functionally distinct heterochromatin domains of characteristic sizes. Chromatin-lamina interactions along the nuclear periphery give rise to lamina associated heterochromatin domains. The supercoiling-driven chromatin loop extrusion through active transcription is captured via the conversion of inactive compacted heterochromatin into transcriptionally active euchromatin loops along the chromatin phase boundaries. Essential and unique to our model is the interplay of the epigenetic and transcriptional kinetics in governing meso-scale chromatin organization – including the size of heterochromatin domains and their spacing in the interior and periphery of the nucleus.

Using this model, we make quantitative predictions that offer a mechanistic explanation for the emergence of size scaling of compacted heterochromatin domains with the rate of supercoiling-mediated loop extrusion at the domain interfaces. Importantly, by including the interactions of chromatin with the nuclear lamina, we show the quantitative dependence of the sizes of lamina-associated domains (LADs) as well as those of interior chromatin domains on the level of transcriptional activity. The predictions on the size scaling of heterochromatin domains made by the model are agnostic to specific interactions, and thus are not limited to a particular cell type. Indeed, the model predictions are qualitatively validated experimentally on five different cell lines and using two different nanoscopic imaging approaches. We used partial wave spectroscopy (PWS), which enables high-throughput, label-free, live cell imaging, in conjunction with scanning transmission electron microscopy tomography with ChromEM staining (ChromSTEM), which allows 3-dimensional high-resolution quantification of chromatin mass distribution, to quantify statistical domain properties upon inhibition of transcription. We, further, quantitatively validated our predictions by analyzing the length scales of compacted chromatin domains previously reported using stochastic optical reconstruction microscopy (STORM) imaging[19]. In conjunction with super-resolution microscopy and nano-imaging techniques, our model establishes a foundation for a predictive framework with broad implications for understanding the role of transcriptional and epigenetic crosstalk in defining mesoscale genome organization.

## Results

### Numerical simulations capture experimentally observed features of chromatin organization

We have developed a mathematical model to capture dynamic chromatin organization in the nucleus, in terms of its compaction into the heterochromatic phase or decompaction into the euchromatic phase (Fig. 1a). We treat the meso-scale genomic organization as a dynamic, far-from-equilibrium process, governed by the energetics of phase-separation in conjunction with the kinetics of epigenetic reactions and the formation of chromatin loops aided by supercoiled DNA extrusion through cohesin due to RNAPII-mediated transcription. The model ingredients are depicted schematically in Fig. 1a. We begin by defining the energetics of the chromatin distribution in terms of the entropic-enthalpic balance of chromatin-chromatin interactions, the chromatin-lamina interactions as well as the penalty on the formation of phase boundaries via Eq. (6) (refer Methods, and Supplementary Section S1.2 in the SI). The gradients in the free-energy landscape, defined as the chemical potential (refer Supplementary Eq. (S3)), drive the dynamic evolution of chromatin towards the two energy wells corresponding to the euchromatin and heterochromatin phases via Eq. (7a, b) (refer Methods, Supplementary Section (S1.4) in the SI). Interconversion of the two phases of chromatin can occur via (a) epigenetic regulation of histone acetylation and methylation (Fig. 1b), and (b) supercoiling-driven extrusion of chromatin loops from heterochromatin into euchromatin along the phase boundaries (Eq. (7b)) as shown in Fig. 1c.

The process of phase separation is initiated by adding a random perturbation to the initially uniform chromatin configuration (as shown in Fig. 2a, left panel) which captures the intrinsic intranuclear heterogeneities. As the simulation progresses heterochromatin domains (in red, center panel of Fig. 2a) spontaneously nucleate and grow. The evolution ultimately stabilizes resulting in a steady state (right panel of Fig. 2a) with a quasi-periodic distribution of stable domains of heterochromatin rich phase ($\phi_h = \phi_h^{max}$) in red and euchromatin rich phase ($\phi_h = 0$) in blue. Each of these domains are nearly circular (see Supplementary Section S2 of SI for a discussion on non-circular lamellar domains) with characteristic sizes. Concomitantly, heterochromatin domains localized to the nuclear lamina (called LADs) of comparable sizes appear in our simulations (Fig. 2a).

The meso-scale distribution of chromatin throughout the nucleus predicted by the mathematical model presents a striking qualitative similarity with the experimentally observed distribution of DNA in the nucleus using ChromSTEM, and STORM as reported previously[19] (Fig. 2b). Domains of compacted chromatin with a characteristic size are observed via a high histone density distinguished from regions of low histone density (Fig. 2b). Lastly, the preferential accumulation of heterochromatin domains along the nuclear periphery seen via STORM imaging (Fig. 2b), again with similar size scale, is also in excellent agreement with the experiments.

When defining the free energy density of chromatin organization in the nucleus (see Supplementary Eq. (S1) in SI), we penalized the formation of sharp interfaces via an interface penalty $\eta$, defined as the energy cost associated with the formation of the interfaces between heterochromatin and euchromatin phases. As we show in the SI (Supplementary Section S1.5), the energy penalty $\eta$ results in the formation of a smooth rather than a sharp interface between the heterochromatin and the euchromatin phases. Numerical simulations of chromatin organization exhibit such smooth interfaces around chromatin domains, as shown in the zoomed in image in Fig. 2c (right panel). The width of the interface $\delta$ is controlled by the competition between the interfacial and bulk energy contributions (refer Supplementary Section S1.5).

Smooth chromatin phase boundaries are indeed observed in-vivo via Chrom-STEM imaging (Supplementary Section S1.11). We characterized the 3D chromatin density around individual

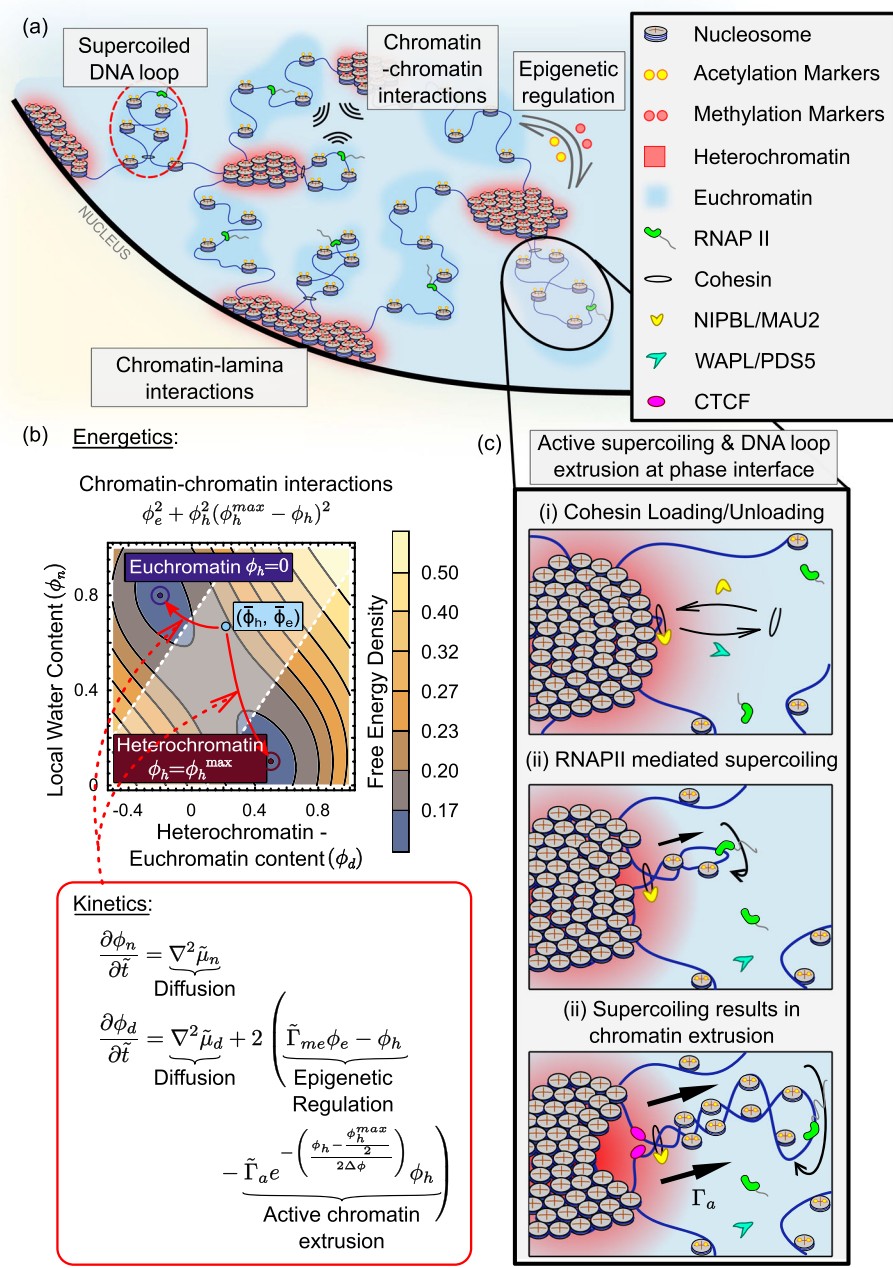

**Fig. 1 | Schematic description of model ingredients. a** Schematic of a portion of a nucleus showing the multiple mechanisms involved in chromatin organization such as chromatin-chromatin interactions, the chromatin-lamina interactions and epigenetic regulation. Additionally, extrusion of chromatin loops due to DNA supercoiling – which is increased by transcriptional activity – also plays a role in mesoscale genomic organization. While this may occur within either chromatin phases (red circle), we further explore the role of chromatin loop extrusion at the heterochromatin-euchromatin interface (black circle). **b** The model captures the chromatin-chromatin interaction energetics via a double well free energy description as shown in the contour plot. The two wells correspond to the heterochromatin (red circle) and euchromatin phases (blue circle). Any initial configuration (light blue circle) spontaneously decomposes into these wells at steady state. The dynamics of this transition are governed by diffusion and reaction kinetics comprising of epigenetic regulation and kinetics of supercoiling-driven chromatin extrusion (red box inset). **c** Loading of cohesin assisted by NIPBL/MAU2 initiates the formation of chromatin loops. Cohesin can also be dynamically unloaded via unloading factors viz. WAPL/PDS5. Active processes such as RNAPII mediated transcription further drive the extrusion of trapped DNA, supercoiling it into chromatin loops.

heterochromatin domains in a BJ fibroblast nucleus using Chrom-STEM (Fig. 2c, left panel; Supplementary Fig. S5). We estimated the average chromatin density within concentric circles emerging from the center of individual domains to the periphery (Fig. 2c, Supplementary Fig. S5). The chromatin density was highest at the core of the domain and dropped slowly from the center of the domain to the periphery. The smooth decrease in radial density indicates that the

chromatin domain boundaries are not abrupt (Fig. 2c), in agreement with the numerical simulations.

We next investigate how the size scaling of the heterochromatin domains is regulated by the epigenetic reactions – acetylation and methylation of histones – and supercoiling-driven chromatin extrusion which together can lead to interconversion between heterochromatin and euchromatin. First, we see that in the absence of the epigenetic

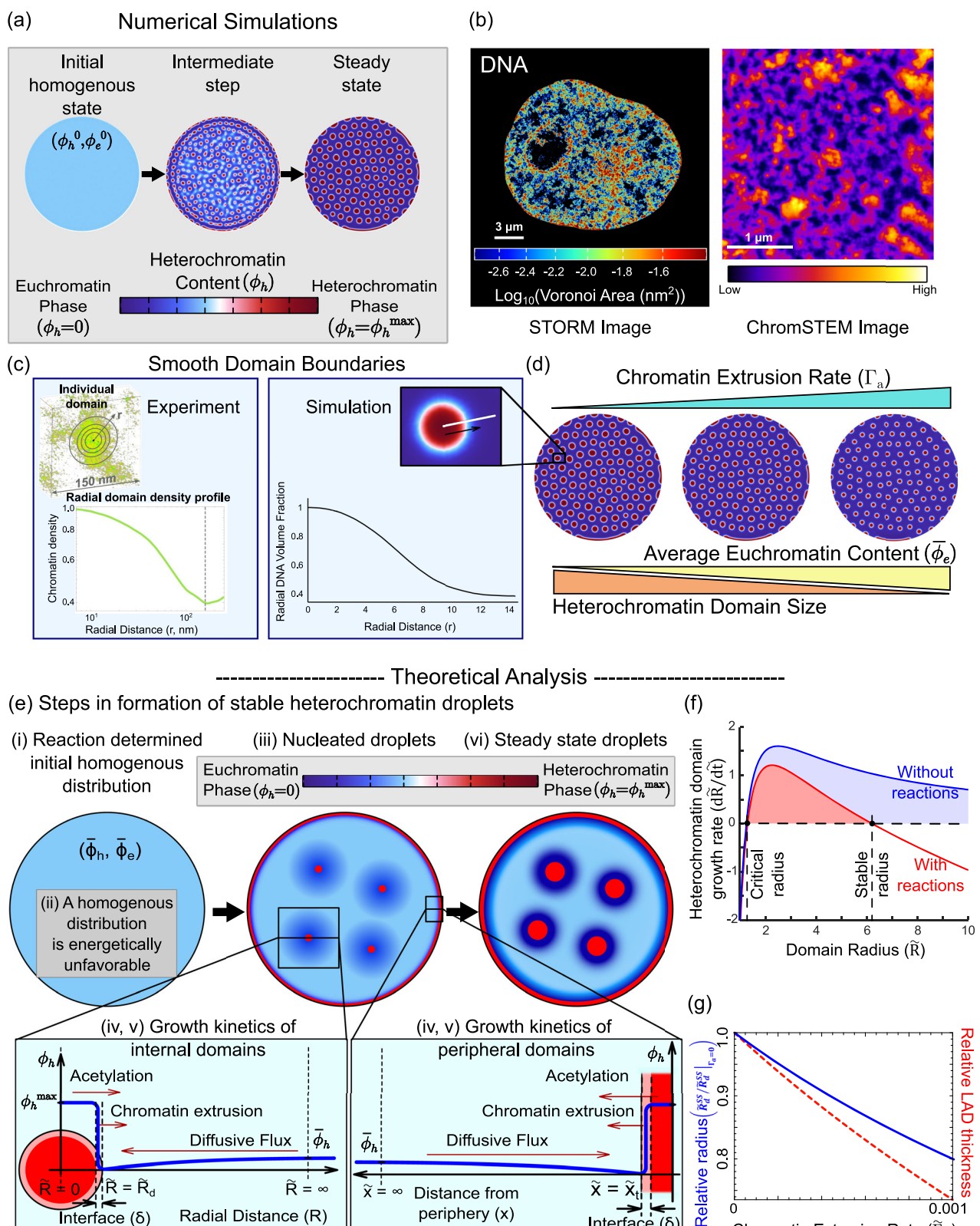

reactions and chromatin extrusion multiple domains of a characteristic size are not obtained as shown in Supplementary Fig. S10 (detailed discussion in Supplementary Section S5). In this case, although nucleation of multiple heterochromatin domains occurs even without reactions (Supplementary Fig. S10a), all of them merge into a single large cluster driven by Ostwald ripening so as to minimize the interface formation.

The model also predicts that the size of the heterochromatin domains in the interior and periphery can be regulated by the epigenetic reaction rates of acetylation and methylation as shown in Supplementary Fig. S6 (Supplementary Section S2). We see that as methylation increases the size of the interior domains increases too. On the other hand, increase in acetylation results in the formation of smaller heterochromatin domains. The trends followed by the

**Fig. 2 | Numerically predicted chromatin distribution in the nucleus captures the salient features of in-vivo chromatin organization. a** Visualization of the chromatin organization obtained from the simulations. The initial chromatin organization is a homogenous distribution with a small perturbation added, resulting in nucleation of heterochromatin domains (center panel) which grow into heterochromatin domains of characteristic sizes at a steady state. **b** Super-resolution visualizations of chromatin organization observed in-vivo via STORM imaging of HeLa nuclei (left panel, scale bar 3 μm, data previously reported in ref. 19, n = 19 nuclei) and ChromSTEM imaging of BJ fibroblast nuclei (right panel, scale bar 1 μm, n = 1 nucleus) show that chromatin organization in nucleus is characterized by interspersed heterochromatic domains of comparable sizes. **c** The smooth boundaries of the chromatin packing domains as seen in ChromSTEM observations

are captured by the model. **d** Numerically predicted trend of sizes of heterochromatin domains as the transcription-mediated chromatin extrusion rate increases. **e** Schematic diagrams of the step-by step events (events 'i' through 'vi') involved in the nucleation, growth and stabilization of heterochromatin domains at a steady state. **f** Plot of theoretically evaluated growth rate of heterochromatin domains with (red) and without (blue) reactions. Reactions give rise to a stable domain radius. In the absence of reactions, no stable heterochromatin domain length scales are observed. **g** The evaluation of stable radius (blue) and stable LAD thickness (red) as transcription mediated surface reactions are changed. Here, the relative radius is defined as the steady state radius relative to its value when transcription is zero, i.e., relative radius = $\tilde{R}_d^{SS}/\tilde{R}_d^{SS}|\Gamma_a = 0$. The relative LAD thickness is similarly defined.

domains towards the interior of the nucleus are replicated by the LADs as well. Lastly, we identify that the size scales of the domains – the domain radii in the interior of the nucleus and the LAD thickness along its periphery – depend on the level of transcription governed supercoiling-driven chromatin extrusion rate $\tilde{\Gamma}_a$ (Fig. 2d, Supplementary Fig. S6). We note that, as the transcription ($\tilde{\Gamma}_a$) is increased, the sizes of the heterochromatin domains decrease, both in the interior as well as at the periphery. At the same time, we also note that as chromatin extrusion rate is increased, the average volume fraction of heterochromatin ($\bar{\phi}_h$) in the nucleus decreases, while that of euchromatin ($\bar{\phi}_e$) increases.

## Theoretical analysis predicts how the heterochromatic domain and LAD sizes depend on epigenetic and transcriptional regulation

Next, we theoretically predict an explicit dependence of the sizes of interior heterochromatic domains and LADs on epigenetic and transcription reactions and the diffusion kinetics of the epigenetic marks.

Intuitively, in the presence of more repressive methylation the overall heterochromatin content in the nucleus should increase, while in higher histone acetylation conditions the overall euchromatin content will increase. Thus, the epigenetic reactions can independently determine the average volume fractions of each form of chromatin, thereby breaking the detailed balance condition where the free energies of each phase determine their relative abundance in a thermodynamic equilibrium. A mathematical relation between the average volume fraction of each chromatin phase and the epigenetic reaction parameters can be determined by averaging the chromatin evolution equation (Eq. (7b)) at a steady state (i.e. $\frac{\partial \phi_d}{\partial t} = 0$). In the absence of transcription driven chromatin extrusion (i.e. $\tilde{\Gamma}_a = 0$), we see that the epigenetic kinetics regulates the average heterochromatin content of the nucleus as, $\bar{\phi}_h \approx \frac{\tilde{\Gamma}_{me}(1-\bar{\phi}_n)}{\tilde{\Gamma}_{me}+1}$ (Supplementary Eq. (S23), refer Supplementary Section S3 for more details).

The presence of transcription-mediated loop extrusion kinetics (i.e., $\tilde{\Gamma}_a \neq 0$ in Eq. (7b)) further augments the deviation from thermodynamic equilibrium (i.e., the breaking of detail balance) via surface reactions that actively extrude DNA at the interface of heterochromatic domains. In the presence of transcription, the average heterochromatin (and euchromatin) content in the nucleus becomes (refer Supplementary Eq. (S22)),

$$\bar{\phi}_h \approx \frac{\tilde{\Gamma}_{me}(1-\bar{\phi}_n)}{\tilde{\Gamma}_{me}+1+\kappa\tilde{\Gamma}_a}, \quad \bar{\phi}_e \approx \frac{(1+\kappa\tilde{\Gamma}_a)(1-\bar{\phi}_n)}{\tilde{\Gamma}_{me}+1+\kappa\tilde{\Gamma}_a}, \quad (1)$$

where $\kappa$ is a function of $\phi_h^{max}$, volume fraction change across the interface $\Delta\phi$, and the length of the interface between the two chromatin phases (refer Supplementary Section S3 for derivation). Since supercoiling-mediated chromatin extrusion converts the tightly packed heterochromatin into low density transcriptionally active

euchromatin phase, as extrusion rate $\tilde{\Gamma}_a$ increases, the average heterochromatin content decreases.

Thus, the overall mean chromatin composition of the nucleus ($\bar{\phi}_h, \bar{\phi}_e$) is determined by the reaction kinetics of epigenetic regulation along with transcription. The reaction kinetics alone would drive a homogenous chromatin organization with ($\bar{\phi}_h, \bar{\phi}_e$). On the ($\phi_d, \phi_n$) phase space we see that the average composition (shown as a light blue circle in Fig. 1b) determined by reactions is energetically unfavorable – it does not lie in the energy wells – and hence must evolve in time.

Next, we show that the average composition of the two chromatin phases, shown in Fig. 2e(i), plays a key role in the emergence of the characteristic sizes of the heterochromatin domains. To illustrate this, we first observe that the mean chromatin composition ($\bar{\phi}_h, \bar{\phi}_e$) lies in neither of the energy wells as shown in Fig. 1b (light blue circle) and is thus energetically unfavorable. The need to reduce the total free energy in the nucleus drives the system to phase separate by nucleating heterochromatin domains (Fig. 2e(iii)) corresponding to the red energy well labeled heterochromatin in Fig. 1b surrounded by euchromatin domains corresponding to the dark blue energy well labeled euchromatin. The events entailing the individual steps in the nucleation and growth of a single droplet of heterochromatin due to phase separation, as shown in Fig. 2e, are as follows:

1. Due to phase separation, the heterochromatin volume fraction immediately outside the droplet is $\phi_h = 0$ corresponding to the euchromatic energy well. Far away from the droplet, the mean composition ($\bar{\phi}_h, \bar{\phi}_e$) remains undisturbed. The resulting spatial gradient in the chromatin composition (blue curve in Fig. 2e(iv)) sets up a diffusive flux of heterochromatin into the droplet, allowing it to grow.

2. On the other hand, within the heterochromatin droplet (with $\phi_h = \phi_h^{max}$) histone acetylation reactions will allow conversion of heterochromatin inside the droplet into euchromatin outside. Active supercoiling-mediated chromatin loop extrusion further adds to the heterochromatin outflux. Together loop extrusion and acetylation oppose the diffusive influx of heterochromatin and thereby reduce the size of the droplet (Fig. 2e).

3. Based on the above observations, the rate at which the nucleated heterochromatin droplet grows can be written in terms of the balance of reaction-diffusion gradient driven influx and acetylation and transcription driven outflux of heterochromatin as (refer Supplementary Section S4, Supplementary Eq. (S25)),

$$4\pi\tilde{R}_d^2\frac{d\tilde{R}_d}{d\tilde{t}} = \underbrace{4\pi\tilde{R}_d\bar{\phi}_h}_{\text{inwards diffusion}} - \underbrace{\frac{4}{3}\pi\tilde{R}_d^3\phi_h^{max}}_{\substack{\text{Acetylation working} \\ \text{against inwards} \\ \text{diffusion}}} - \underbrace{4\pi\tilde{R}_d^2\frac{\delta}{2}\tilde{\Gamma}_a\phi_h^{max}}_{\substack{\text{Chromatin extrusion working} \\ \text{against inwards} \\ \text{diffusion}}}$$

$$(2)$$

where $\delta$ is the rescaled width of the interface, which is in turn related to the length scale obtained via the competition between the interfacial energy and chromatin-chromatin interaction (refer Supplementary

Section S1.5). The resulting evolution of the droplet growth rate $\left(d\widetilde{R}_d/d\widetilde{t}\right)$ as the radius of the droplet increases is shown in Fig. 2e. Notice the two fixed points (Fig. 2f, labeled critical and stable radius) where $d\widetilde{R}_d/d\widetilde{t} = 0$. Beyond the critical radius the domains grow in size.

4. The second fixed point (stable radius) corresponds to the rescaled steady state (i.e., $d\widetilde{R}_d/d\widetilde{t} = 0$) heterochromatin domain size as determined by the active epigenetic and the transcriptional regulation in tandem with passive diffusion, and can be written as (derivation shown in Supplementary Section S4, Supplementary Eq. (S27)),

$$\widetilde{R}_d^{ss} = -\frac{3\widetilde{\Gamma}_a\delta}{4} + \sqrt{\left(\frac{3\widetilde{\Gamma}_a\delta}{4}\right)^2 + \frac{3}{\phi_h^{\max}}\frac{\widetilde{\Gamma}_{me}(1-\bar{\phi}_n)}{1+\widetilde{\Gamma}_{me}+\kappa\widetilde{\Gamma}_a}}. \quad (3)$$

From Eq. (3), we observe that the steady state droplet radius $\left(\widetilde{R}_d^{ss}\right)$ depends on both diffusion and reaction kinetics. With increase in methylation, $\widetilde{R}_d^{ss}$ increases implying bigger heterochromatin domains. On the other hand, with increase in either the acetylation or transcription-mediated loop extrusion the steady state radius decreases. The quantitative dependence of the steady state radius on transcriptional kinetics is shown in Fig. 2g (blue solid line). Note that the steady state radius shown in Fig. 2g is normalized relative to the steady state radius with no transcription. Thus, our theory predicts an increase in the sizes of compacted chromatin domains in the interior of the nucleus upon inhibition of transcription.

The size dependence of chromatin domains along the nuclear periphery can be similarly determined by the balance of reaction, transcription, and diffusion kinetics for the LADs. The affinity of chromatin to the nuclear periphery due to the chromatin-lamina interactions in Eq. (6) induces a preferential nucleation of LADs. A schematic representation of heterochromatin compaction along the nuclear periphery resulting in LAD growth is shown in Fig. 2e. As with the interior heterochromatin droplet, phase-separation drives the heterochromatin compaction $\left(\phi_h = \phi_h^{\max}\right)$ within the LADs, while the chromatin immediately outside corresponds to the euchromatin energy minimal well $\left(\phi_h = 0\right)$. Far away from the peripheral LAD nucleation sites, the chromatin composition remains undisturbed at the average composition of $\left(\bar{\phi}_h, \bar{\phi}_e\right)$. The variation of chromatin composition with distance from nuclear periphery is shown in Fig. 2e (blue line). Like in the case of the interior heterochromatin droplets, the heterochromatin composition gradient driven diffusive influx is balanced by the epigenetic and transcriptional regulated heterochromatin outflux, which determines the rescaled steady-state thickness of the LADs (refer to the Supplementary Section S7, Supplementary Eq. (S34)),

$$\widetilde{x}_t^{ss} = \frac{\widetilde{\Gamma}_{me}\left(1-\bar{\phi}_n\right)}{\phi_h^{\max}\left(1+\widetilde{\Gamma}_{me}+\kappa\widetilde{\Gamma}_a\right)} - \frac{\delta\widetilde{\Gamma}_a}{2} \quad (4)$$

As with the interior domains, we observe that the LADs become thicker with increase in methylation, while they become thinner with increasing acetylation or chromatin extrusion rates. A quantitative dependence of steady state LAD thickness on transcription rate based on Eq. (4) is plotted in Fig. 2g (red dashed line). Our theory predicts an increase in the sizes of LADs along the nuclear periphery upon inhibition of transcription. While the theoretical analysis helps develop a fundamental biophysical understanding of the role of energetics and kinetics in chromatin phase separation, a nucleus-wide chromatin organization and its dynamic evolution can only be obtained numerically.

## Loss of transcription results in increase in heterochromatin domain size and LAD thickness

Next, we use the in-silico model to make testable quantitative predictions of the meso-scale chromatin organization in the nucleus. We also report the in-vivo nuclear chromatin reorganization upon transcription inhibition using complimentary STORM[19] and ChromSTEM – on nuclei from multiple cell lines. The choice of the parameters for rates of acetylation $\widetilde{\Gamma}_{ac}$, methylation $\widetilde{\Gamma}_{me}$, and the strength of chromatin-lamina interactions $\widetilde{V}_L$, were held constant for all the following simulations, and the choice of the level of spatial noise is discussed in the Supplementary Section S8. We calibrate the active chromatin supercoiling-driven loop extrusion rate $\Gamma_a$ to obtain an in-silico change in the interior domain sizes quantitatively comparable to that observed upon transcriptional inhibition. The calibrated model is then used to predict the change in LAD thickness due to inhibition of transcription, which upon comparison with experimental images serves to validate the model. A schematic for the workflow utilized to calibrate and cross-validate the model predictions in the interior and along periphery of the nucleus is shown in Supplementary Fig. (S14) (Supplementary Section S8).

ChromSTEM was used to obtain super-resolution images in terms of statistical descriptions of chromatin packing domains for BJ fibroblasts. ChromSTEM allows the quantification of 3D chromatin conformation with high resolution[22]. ChromSTEM mass density tomograms were collected for BJ fibroblasts treated with Actinomycin D (ActD) (Fig. 3a, center) and compared to DMSO treated mock controls (Fig. 3a, left) to evaluate the average size and density of chromatin packing domains. We have previously demonstrated that chromatin forms spatially well-defined higher-order packing domains and that, within these domains, chromatin exhibits a polymeric power-law scaling behavior with radially decreasing mass density moving outwards from the center of the domain[23]. As the ChromSTEM intensity in the reconstructed tomogram is proportional to the chromatin mass density, we estimated the size of the domains based on where the chromatin mass scaling and the radial chromatin density deviate from their predicted behavior (discussed in Supplementary Section S1.11). Based on the statistical analysis of individual packing domains, in a single tomograph shown in Fig. 3a, we observed 71 domains in DMSO and 48 domains in the ActD-treated nucleus. Of the identified domains, the average domain radius (± S.E) of BJ cells treated with DMSO and ActD was estimated to be 103.5 ± 4.73 nm and 129.7 ± 6.78 nm, respectively (Fig. 3a, right panel), representing a 20.2% increase in size. Overall, fewer domain centers, and larger chromatin packing domains were experimentally observed upon ActD treatment compared to the control.

In addition to evaluating domain properties using ChromSTEM, we utilized live-cell partial wave spectroscopy (PWS) imaging to observe the change in chromatin organization after transcription inhibition in various cell lines (Fig. 3b). The PWS images demonstrate a significant reduction in average chromatin packing scaling upon ActD treatment in live cells across four different cell types. Next, the size of the domains is quantitatively approximated via polymer scaling relationships discussed in Supplementary Section S1.13[22,24]. The quantification of the domain sizes (boxplots in Fig. 3b) shows that, for all cell types studied, packing domains are larger for upon transcription inhibition with ActD treatment – in agreement with the ChromSTEM results on BJ fibroblasts.

Additionally, we have previously used STORM imaging to observe the nucleus wide changes in chromatin organization caused by transcription abrogation in HeLa nuclei after ActD treatment[19]. Heatmaps of chromatin density obtained via Voronoi tessellation-based color-coding of STORM images (see[19] for analysis) are shown in Fig. 3c. The zoomed in images of heatmaps of the chromatin cluster density (Fig. 3f) clearly show the increasing heterochromatin domain sizes when RNAPII activity is inhibited, in agreement with our

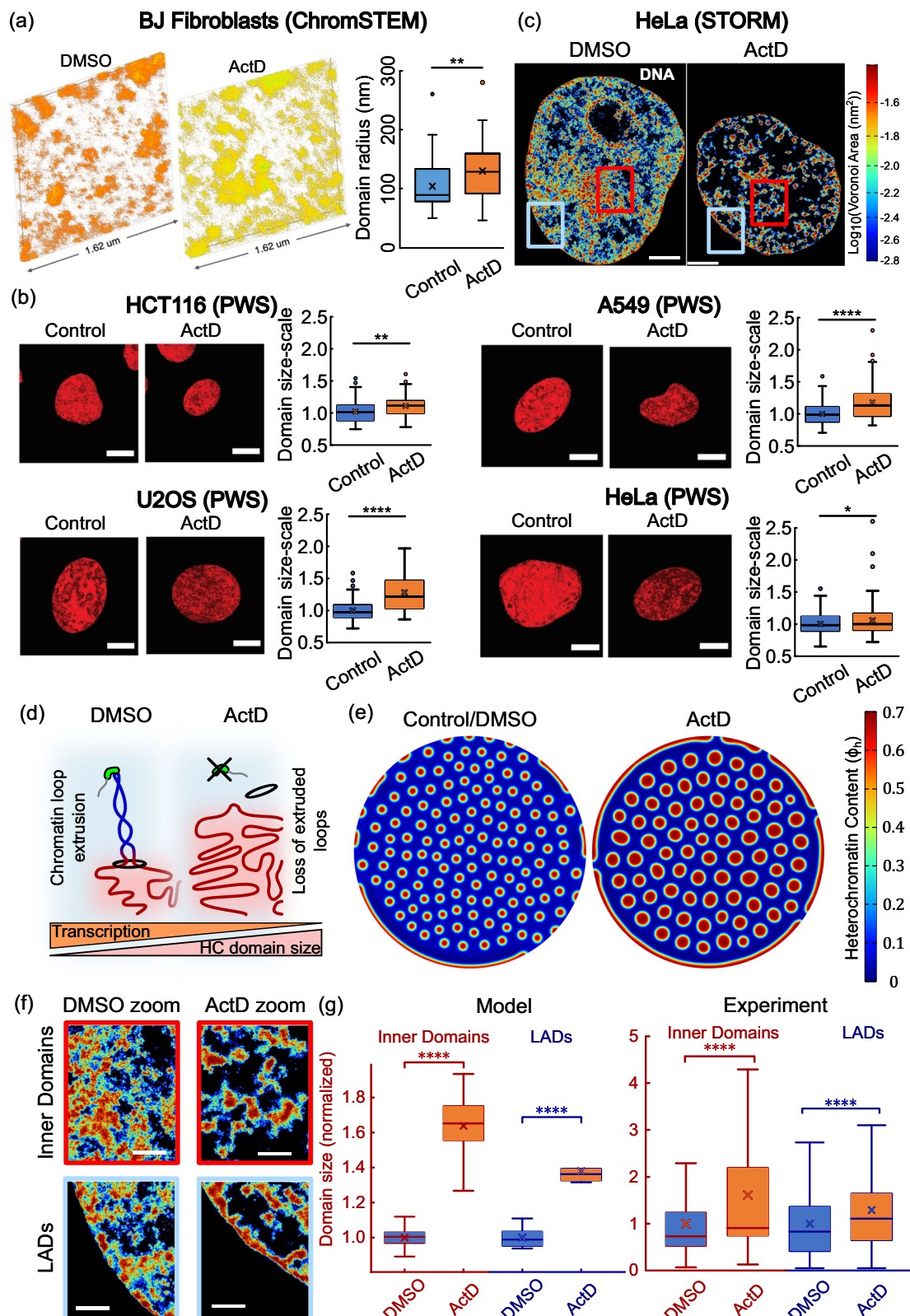

theoretical and numerical predictions (Fig. 2d, e). Importantly, we see that the changes in chromatin organization occur not only in the interior domains of the nucleus but also along its periphery (Fig. 3f, g).

Altogether these complementary imaging techniques establish that nucleus wide increase in sizes of compacted chromatin domains occurs upon the loss of transcription in a wide range of cell lines.

The chromatin cluster density maps obtained from STORM imaging were further analyzed to quantify the sizes of heterochromatin domains after DMSO and ActD treatment. A density-based threshold was used to isolate the high-density heterochromatin regions, which were then clustered via a density based spatial clustering algorithm (see Supplementary Section S1.8) and further sub-classified into LADs and interior domains depending on the distance from nuclear

**Fig. 3 | Heterochromatin domains grow after transcription inhibition.**
**a** ChromSTEM tomogram reconstructions for DMSO (left panel) and ActD treated (center panel) BJ fibroblasts. The domains radii for BJ cells treated ActD (right panel, $n = 48$ domains) show 1.25 times (unpaired two tail t-test, $p = 0.002$) increase compared to control ($n = 71$ domains). **b** Representative live-cell PWS images (1-hour ActD treatment). Scale bars = 5 μm. Box plots compare the domain sizes between DMSO control and ActD treated cells. Sample size – HCT116: $n = 63$ nuclei (control), 65 (ActD), $p = 0.05$; A549: $n = 102$ (control), 84 (ActD), $p = 1e−7$; U2OS: $n = 116$ (control), 75 (ActD), $p = 1e−12$; $n = 103$ (control), 150 (ActD), $p = 0.04$. **c** Heatmap density of DNA super-resolution images in DMSO control (left panel, $n = 19$ nuclei) and ActD (right panel, $n = 20$ nuclei) treated HeLa nuclei. All scale bars – 3 μm. **d** Loss of chromatin loop extrusion due to absence of RNAPII results in increased heterochromatin domain size (in red, nucleosomes not shown for clarity). **e** Numerical prediction of chromatin organization in DMSO control and

ActD treated nucleus. **f** Zoomed in views of DMSO and ActD treated nuclei localized to the nucleus interior (top panels) and the periphery (bottom panels). Red and blue boxes shown in **c** are zoomed into. All scale bars −1 μm. **g** Left: Simulations predict domains in ActD nuclei are on average 1.63 times larger than in DMSO nuclei ($n = 127$ (DMSO), 77 (ActD) unpaired two tail t-test, $p = 0$) while LADs are 1.37 times thicker ($n = 38$ (DMSO), 15 (ActD); unpaired two tail t-test, $p = 0$). Right: Domain radii observed experimentally in ActD treated nuclei ($n = 3584$ loci, 20 nuclei) are 1.61 times (unpaired two tail t-test, $p = 0$) larger than in DMSO nuclei ($n = 5830$ loci,19 nuclei), while LADs are 1.3 times thicker ($n = 1082$ loci (DMSO), 1015 loci (ActD), unpaired two tail t-test, $p = 0.0006$). All boxplots show the mean (cross), median (horizontal line), upper and bottom quartiles (box outlines) and the maximum and minimum non-outlier data points (whiskers). All source data are provided as a source data file.

periphery (Supplementary Section S1.9). The quantitatively extracted distribution of interior heterochromatin domain radii for DMSO and ActD treated nuclei shows that their mean radius after transcription inhibition was nearly 1.61 times that in DMSO controls (Fig. 3g).

Indeed, our model (Eq 3-4, Fig. 2d, g) predicts that loss of transcription results in increased heterochromatin domain size. This is because under control conditions, extrusion of heterochromatin phase into euchromatin occurs. We assume, based on previous experimental findings[19], that the presence of RNAPII activity drives the supercoiling of the DNA loop, thereby extruding it from the heterochromatin phase into the euchromatin phase at the phase boundaries (Fig. 3c, left panel). However, when RNAPII is inhibited with ActD treatment (Fig. 3c, right panel), the absence of this driving force for supercoiling-mediated loop extrusion keeps more DNA in the heterochromatin phase thereby increasing the domain sizes. The in-silico chromatin distribution predicted under control (left panel) and transcription inhibited ($\Gamma_a = 0$, right panel) conditions is shown in Fig. 3e. The phase separated heterochromatin domains ($\phi_h = \phi_h^{max}$) are shown in red in a loosely compacted euchromatin background (blue, $\phi_h = 0$). We quantify the change in the sizes of the heterochromatin domains predicted by the model as the active extrusion rate $\Gamma_a$ is parametrically varied. The value of $\Gamma_a$ under control conditions is chosen (Supplementary Table S2) such that the change in the interior domain sizes with respect to transcription inhibition (with $\Gamma_a = 0$) is quantitatively the same as observed experimentally.

**The model predicts changes in LAD thickness due to transcriptional inhibition with no additional parameters.** Next, we quantitatively validate the choice of $\Gamma_a$ under control conditions by comparing the predicted change in LAD thickness against that quantified from the STORM images. Our theoretical predictions (Eq. (4)) show that the reduction in transcription increases the thickness of the LADs reflecting the behavior predicted in the interior of the nucleus (Fig. 2d, g). Our simulations of chromatin distribution in the nucleus (Fig. 3e) show that inhibition of transcription ($\Gamma_a = 0$) results in thicker LADs. Of note, the chromatin-lamina interaction strength ($V_L$) stays unchanged between the two simulations. Yet, we see a higher association of chromatin with the periphery. Upon quantitative comparison (Fig. 3g, left panel) we see that the LADs grow approximately 1.37 times thicker upon loss of transcription.

To validate this prediction, we compare the predicted change in LAD thickness with that quantified from in-vivo STORM imaging. (Fig. 3g, refer to Supplementary Sections S1.8 and S1.9 for procedure). The quantified comparison of LAD thickness between DMSO and ActD nuclei (Fig. 3g) shows nearly 1.3 times increase upon ActD treatment, in close quantitative agreement with the model prediction. Overall, with both model predictions and cellular observations, our results suggest that impairment of transcription plays a significant role in determining the size scaling of the interior heterochromatin domains and LADs.

**Transcription inhibition results in movement of DNA from the euchromatic into heterochromatic regions**
We next enquire how, in addition to altering the size of the compacted domains, abrogation of transcription changes the extent of DNA packing. For this we analyzed the chromatin distribution in HeLa nuclei under DMSO and ActD treatments from STORM images previously generated[19]. Under control conditions the distribution of DNA is qualitatively more homogenous while ActD treated nuclei exhibit more isolated distinct domains of compacted chromatin surrounded by region of very low chromatin density (Fig. 4a). For quantification, we plot the chromatin intensity along a horizontal line chosen to run across two heterochromatin domains with euchromatin between them (see zoomed images in Fig. 4b, blue and red horizontal line). The chromatin intensity, plotted in Fig. 4c (in blue) shows that even in the euchromatin region, the DNA presence is substantial. On the other hand, chromatin intensity across a horizontal line chosen across a heterochromatin domain in ActD nucleus (Fig. 4b, c; in red) shows a much steeper gradient outside the domain.

The increased presence of DNA in the euchromatic phase in presence of transcription as observed experimentally is captured by the simulations. The in-silico distribution of DNA (measured as the sum of volume fractions of the chromatin phases, $\phi_e + \phi_h$) in a nuclear region far from LADs is plotted in Fig. 4d for control and transcription inhibited in-silico nuclei. We see that the euchromatic phase (outside white circles) is darker when transcription is inhibited, indicating the presence of much lesser DNA than in control euchromatin. A quantification of the total DNA along cut-lines chosen in the control and ActD in-silico nuclei confirm the observations (Fig. 4e).

Since the lack of transcription inhibits supercoiling-mediated chromatin loop extrusion from heterochromatin into euchromatin, we see a reduced density of DNA in the euchromatin phase of the nucleus under ActD conditions. Further, due to the lack of chromatin extrusion out of the heterochromatin domains when transcription is inhibited, we also observe that they are larger in size. Thus, transcription, via chromatin loop extrusion, results in removal of DNA from compacted heterochromatin region by converting it into active euchromatin form.

Taken together, our results suggest that transcription not only affects the scaling of the lengths (radius or thickness) of the heterochromatin domains, but also significantly changes the relative amounts of DNA in the euchromatin and heterochromatin phases.

**Excessive chromatin loop extrusion reduces the sizes of chromatin domains**
We have established that change in transcription activity affects the global chromatin organization of the nucleus via altered supercoiling mediated loop extrusion. In turn, chromatin loop extrusion is initiated by the loading of cohesin onto DNA via a balance between cohesin loaders such as NIPBL and cohesin unloaders like WAPL (Fig. 1c[2,12,25,26]). If the chromatin loop extrusion is responsible for the global chromatin reorganization, altering the cohesin loading/unloading balance must

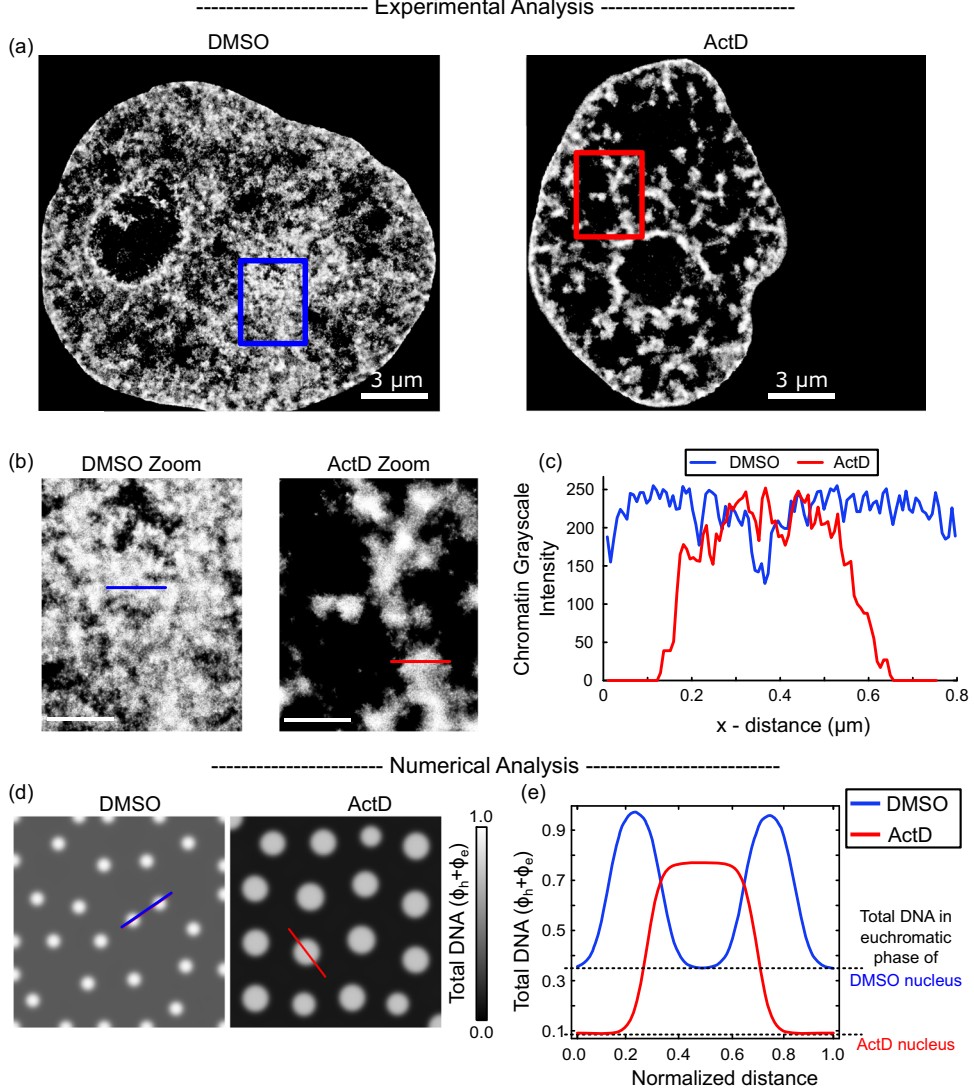

**Fig. 4 | Loss of transcription reduces the amount of DNA in euchromatic phase.**
**a** Grayscale heatmap density rendering of super-resolution images of DNA in control (DMSO, left panel, $n = 19$ nuclei) and actinomycin D (ActD, right panel, $n = 20$ nuclei) treated HeLa nuclei. All scale bars – 3 μm. **b** Zoomed in views of DMSO and ActD treated nuclei. Boxes shown in **a** are zoomed into. All scale bars - 1 μm. **c** Along the blue (DMSO) and red (ActD) line segments, we plot the chromatin heatmap intensity (corresponding to the total DNA content) for the DMSO-treated control nucleus (in blue) and ActD-treated nucleus (in red). The DMSO-treated

nucleus shows a wider distribution of small heterochromatin domains, while the ActD treated nucleus shows a greater compaction with isolated large heterochromatin domains. **d** Numerical prediction of distribution of total DNA (in grayscale) in a nucleus with (DMSO) and without (ActD) transcription mediated chromatin extrusion. **e** Distribution of total DNA content along the blue (red) line in **d** under DMSO (ActD) treatment. The black dashed line shows the level of total DNA predicted in the euchromatin phase of DMSO and ActD treated nuclei.

also result in chromatin reorganization. Thus, next, we study the chromatin arrangement in WAPL-deficient (WAPLΔ) nuclei marked by increased levels of loaded cohesin.

In vivo, WAPL depletion causes an accumulation of large amounts of cohesin on chromatin[27]. This results in a much more homogenous distribution of DNA, which was previously termed "blending" due to excessive extrusion of chromatin loops, as shown schematically in Fig. 5a[19]. In our mathematical model, WAPL deficiency is simulated as an increase in the rate of chromatin extrusion ($\Gamma_a$). Based on the theoretical size scaling of the interior heterochromatin domains and LADs, as seen from Eq. (3) and Fig. 2g, our model predicts that increase in $\Gamma_a$ would result in a decrease in the radius of the steady state heterochromatin domains (Fig. 5b).

STORM images of HeLa nuclei without (labeled Cas9) and with WAPL-deficiency previously revealed genome-wide changes in the chromatin organization induced by excessive loading of cohesin

(Fig. 5c, d)[19]. A visual comparison between representative zoomed-in regions (white boxes in Fig. 5c) demonstrates the reduction of heterochromatin domain sizes in the interior of the nuclei in WAPLΔ nuclei (Fig. 5d). Using clustering analysis (refer Supplementary Section S1.8 and S1.9), we quantify the altered chromatin domain sizes in control and WAPLΔ HeLa cell nuclei. We observe that WAPLΔ nuclei with increased chromatin blending have heterochromatin domains with a mean radius approximately 15% smaller than control nuclei (Fig. 5e).

In-silico, we parametrically vary the active chromatin extrusion rate $\Gamma_a$ above the control level (Supplementary Table S2, determined for control treatment). The value of $\Gamma_a$ for WAPLΔ nuclei is chosen (Supplementary Table S2) such that the decrease in the size of interior heterochromatin domains reduces by 15% (Fig. 5f) to agree with the experimental observation (Fig. 5e).

As discussed previously (Fig. 2g), the model predicts that the effects of chromatin extrusion observed in the interior domains of the

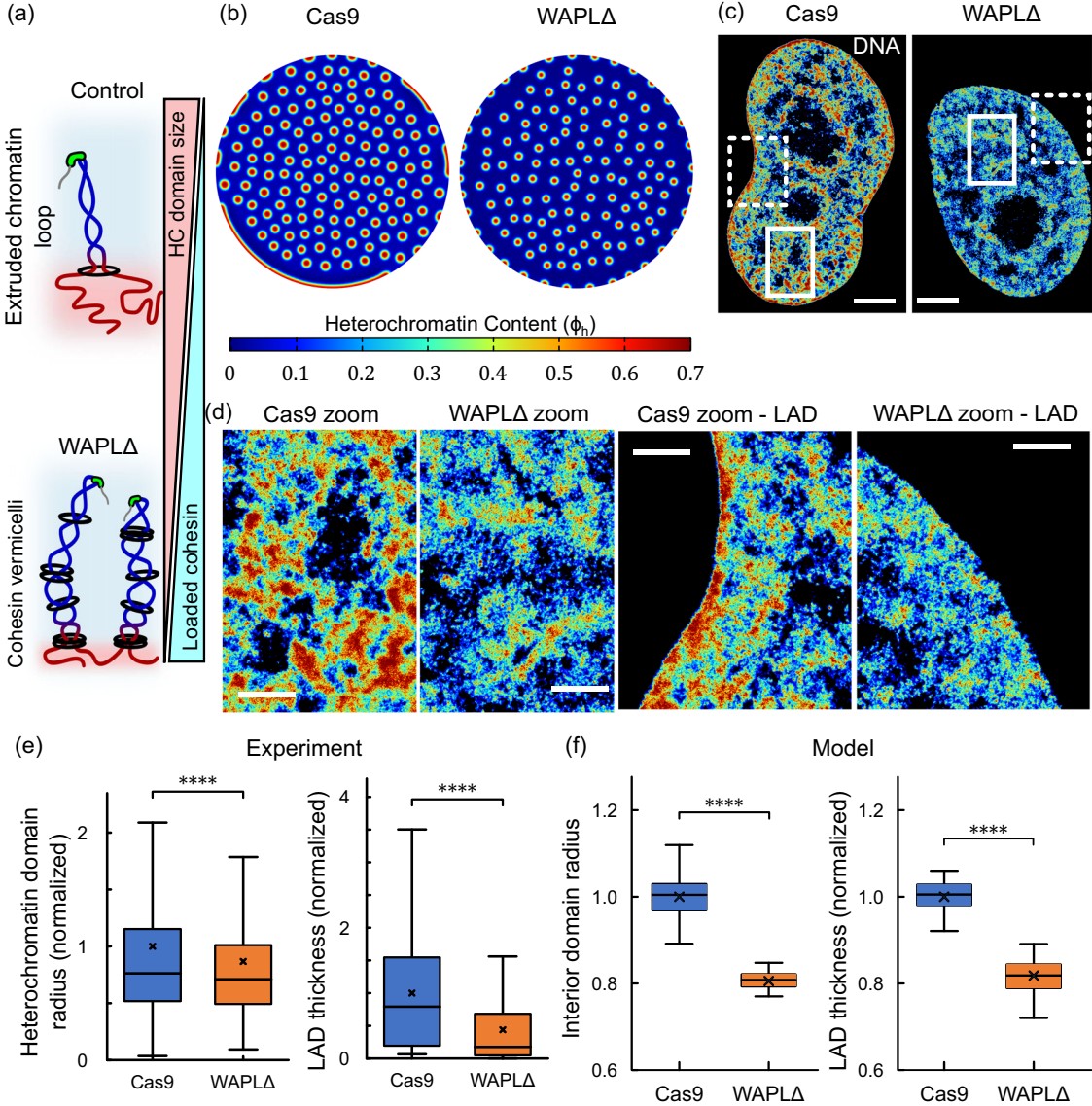

**Fig. 5 | Heterochromatin domains become smaller upon WAPL-depletion.**
**a** Schematic representation of chromatin loop extrusion. WAPL-depletion results in increased cohesin loading and excessive transcription-driven chromatin loop extrusion. Note that nucleosomes, despite being present, are not shown to improve clarity. **b** Numerical prediction of distribution of heterochromatin domains in the interior and the LADs along the periphery (all domains in red) in a nucleus without (Cas9) and with (WAPLΔ) cohesin unloading disruption. **c** Heatmap density of DNA super-resolution images in **d** control (Cas9, left panel) and WAPL knock-out (WAPLΔ) treated HeLa nuclei. All scale bars - 3 μm. **d** Left: Zoomed in views of Cas9 and WAPLΔ treated nuclei focusing on the interior heterochromatin domains. White solid boxes shown in **c** are zoomed into. All scale bars - 1 μm. Right: Zoomed in views of Cas9 and WAPLΔ treated nuclei along the nuclear periphery. White dashed boxes shown in **c** are zoomed into. All scale bars - 1 μm. **e** Quantification of heterochromatin domain radius in the interior of Cas9- and WAPLΔ - treated nuclei. ($n = 2386$ loci in 6 nuclei for Cas9-treatment and 2416 loci in 7 nuclei for WAPLΔ

treatment). WAPLΔ treated nuclei exhibit a significantly lower ($\sim 0.86$ times) mean heterochromatin radius (unpaired two-tailed t-test, $p = 6e{-}10$). Quantification of LAD thickness along the periphery of Cas9- and WAPLΔ - treated nuclei. ($n = 219$ loci in 6 nuclei for Cas9-treatment and 169 loci in 7 nuclei for WAPLΔ treatment). WAPLΔ treated nuclei exhibit a significantly lower ($\sim 0.43$ times) mean LAD thickness (unpaired two-tailed t-test, $p = 1e{-}13$). **f** Boxplot in left panel shows the distribution of domain radii predicted numerically. WAPLΔ nuclei have a mean domain radius 0.8 times that of Cas9-treated nuclei (unpaired two-tailed t-test, $p = 0$). Boxplot in right panels shows the distribution of LAD thicknesses predicted numerically. WAPLΔ nuclei have a mean LAD thickness 0.82 times that of Cas9-treated nuclei. All boxplots show the mean (cross), median (horizontal line), upper and bottom quartiles (box outlines) and the maximum and minimum non-outlier data points (whiskers) of the plotted distribution. All source data are provided as a source data file.

nucleus are replicated along the nuclear periphery. Simulation of nuclear chromatin organization (Fig. 5b) reveals that by changing only the rate of chromatin extrusion $\Gamma_a$, keeping all other parameters including chromatin-lamina interaction potential $V_L$ constant, we see a reduction in the association of chromatin with the lamina. Specifically, a 2.5-fold increase in $\Gamma_a$ calibrated to occur due to WAPL-deficiency predicts a 51.2% decrease in the average LAD thickness, as shown in Fig. 5f.

The predicted change in LAD thickness is consistent with previous experimental observations and was further quantitatively validated by

measuring the thickness of LADs in STORM images of control and WAPLΔ nuclei (Fig. 5e)[19]. A reduction in the sizes of domains, as seen in the nucleus interior, can also be observed at the nuclear periphery, as shown in a representative zoomed in region (white dashed boxes in Fig. 5c) in Fig. 5d. The mean thickness of the LADs at the nuclear periphery is approximately 20% smaller for WAPLΔ nuclei (Fig. 5h) as compared to the control-treated nuclei.

Together, these results confirm that the meso-scale spatial chromatin organization is strongly regulated by the chromatin loop

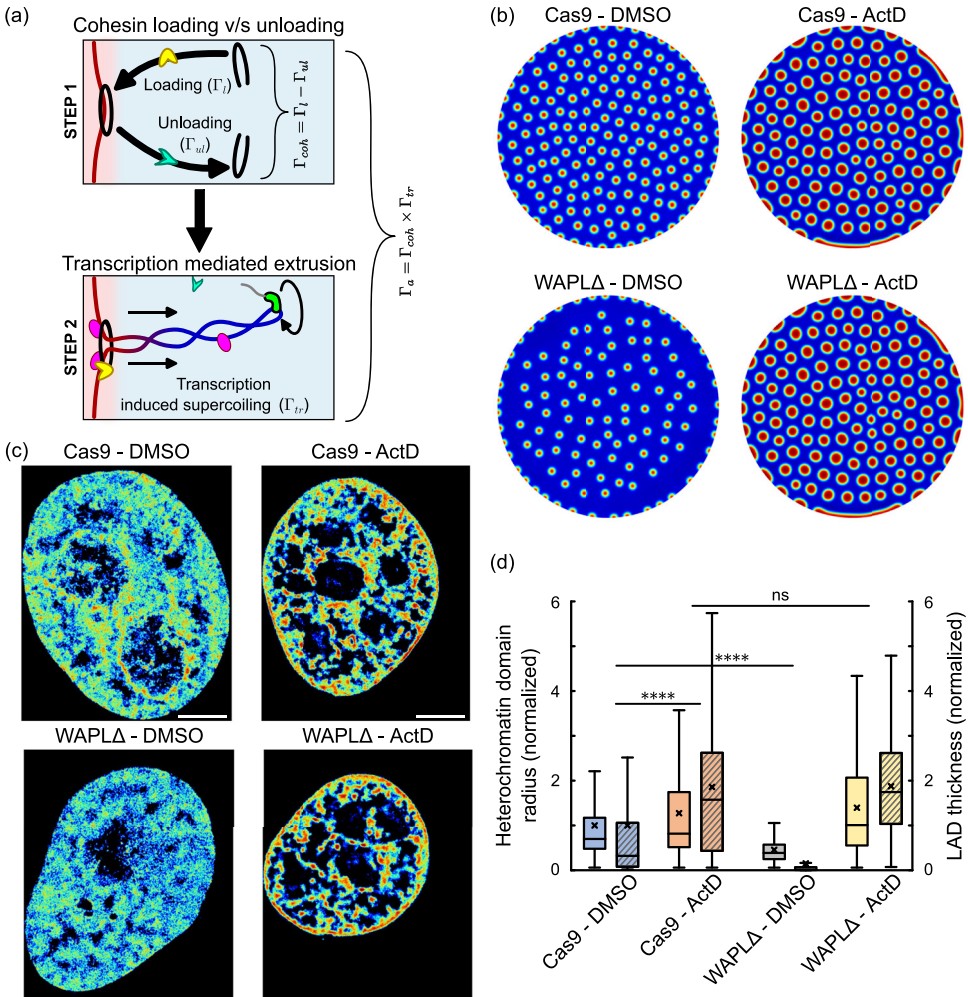

**Fig. 6 | Simultaneous roles of transcription inhibition and cohesin imbalance (via disabling cohesin unloading WALPΔ). a** Schematic showing the associative sub-steps of chromatin extrusion incorporating cohesin loading v/s unloading balance and active transcriptional work done by RNAPII. The rate of active extrusion of chromatin loops ($\Gamma_a$) is determined by both sub-steps. Note that nucleosomes, despite being present, are not represented in this schematic to better display the chromatin loops. **b** Numerical prediction of distribution of heterochromatin domains in the interior and the LADs along the periphery (all domains in red) in a nucleus in control (Cas9-DMSO treatment, top-left panel), transcription inhibited (Cas9-ActD, top right), WAPL knock-out treated (WAPLΔ-DMSO, bottom left) and simultaneous WAPL knock-out along with transcription inhibition treated (WAPLΔ-ActD, bottom right). **c** Heatmap density rendering of super-resolution images of DNA in control (Cas9-DMSO treatment, left panel), transcription inhibited (Cas9-ActD, center left), WAPL knock-out treated (WAPLΔ-DMSO, center right)

and simultaneous WAPL knock-out along with transcription inhibition treated (WAPLΔ-ActD) HeLa nuclei. All scale bars − 3 μm. **d** Quantification of heterochromatin domain radius in the interior (plain colored boxes) as well as the LAD thickness along the nuclear periphery (hatched boxes) of Cas9-DMSO (3328 loci in 13 nuclei), Cas9-ActD (4042 loci in 11 nuclei), WAPLΔ-DMSO (1548 loci in 10 nulcei) and WAPLΔ-ActD (1926 loci in 11 nuclei) treated nuclei. As previously, ActD treated nuclei exhibited a significantly increased domain size (unpaired two-tailed t-test, $p = 0$) while WAPLΔ treated nuclei exhibit a significantly lower mean heterochromatin radius (unpaired tw-tailed t-test, $p = 0$). However, the differences between Cas9-ActD treated and WAPLΔ-ActD treated nuclei was insignificant (unpaired two-tailed t-test, p ∼ 0.9). All boxplots show the mean (cross), median (horizontal line), upper and bottom quartiles (box outlines) and the maximum and minimum non-outlier data points (whiskers) of the plotted distribution. All source data are provided as a source data file.

formation, and this effect can be modulated not only by the transcription activity, but also by altering the extent of loading or unloading of cohesin rings on the DNA. These results provide further evidence for the link between transcriptional regulation and nucleus-wide chromatin distribution via transcription-driven supercoiling mediated chromatin loop extrusion.

## Chromatin blending in WAPL deficient cells is blocked by transcription inhibition

Since we have established, via both quantitative analysis of experimental data and simulations, that extrusion of chromatin loops is governed by both cohesin loading/unloading balance and RNAPII mediated transcription, a question of their tandem role emerges.

To simulate the individual effects of cohesin loading and transcriptional activity, we decompose the overall active chromatin extrusion rate into its distinct constitutive steps. The individual steps involved in the process of supercoiling mediated chromatin loop extrusion from heterochromatin into euchromatin (as discussed previously in Section "Introduction") are shown in Fig. 6a. As a first step, a balance between the loading of cohesin via NIPBL/MAU2[25] on chromatin occurring at a rate $\Gamma_l$ and its unloading via by WAPL/PDS5[2,12,26] occurring at a rate $\Gamma_{ul}$ results in the association of cohesin rings with chromatin at an overall rate $\Gamma_{coh} = \Gamma_l - \Gamma_{ul}$. In other words, $\Gamma_{coh}$ denotes the overall rate of cohesin loading on DNA. The entrapment of DNA by cohesin is followed by the extrusion of supercoiled loops of chromatin via DNA supercoiling by the RNAPII mediated transcription, at a rate denoted by $\Gamma_{tr}$. Thus, as shown in Fig. 6a, by assuming a first-order

reaction kinetics for both steps, the overall rate of active chromatin extrusion $\Gamma_a$ at the interface of heterochromatin and euchromatin is proposed to be multiplicatively decomposed as,

$$\Gamma_a = \Gamma_{tr}\Gamma_{coh} = \Gamma_{tr}(\Gamma_l - \Gamma_{ul}) \tag{5}$$

In addition to the extrusion of loops via RNAPII mediated DNA supercoiling activity[12,13,19,28–30], in vitro experiments proposed that cohesin once transiently loaded onto DNA, could independently drive the formation of loops via its ATPase machinery[9,11,31–33]. Cell based experiments demonstrated that in WAPLΔ cells, clusters of cohesin in WAPLΔ cells assemble together into vermicelli-like structures and these structures disappear upon transcription inhibition, but not upon partial loss of cohesin[19]. These results, taken together, present strong evidence for the important role of transcription in powering cohesin mediated loop extrusion. While the relative role of cohesin's motor activity and transcription in loop extrusion inside cells remains to be determined, here we focus on the latter given the previous in vivo experimental findings. We indeed show that a kinetic model captured by Eq. (5) sufficiently explains the effect of extrusion of the specific chromatin loops extending from transcriptionally silenced heterochromatin into genetically active euchromatin on determining the meso-scale chromatin domain sizes.

The chromatin organization is simulated in a nucleus under control and transcription inhibition treatments for nuclei with and without WAPL deficiency. The chromatin organization in a control nucleus (labeled Cas9-DMSO), simulated via parameters listed in Supplementary Table S1 is shown in Fig. 6b, top-left panel. The individual inhibition of transcriptional activity without affecting the cohesin loading (Cas9-ActD) results in a chromatin organization with increased heterochromatin domains sizes and LAD thickness, as shown in Fig. 6b, top-right panel. On the other hand, the simulation of chromatin distribution in nucleus with depleted cohesin unloading, without disturbing the transcriptional activity, (WAPLΔ-DMSO) is shown in Fig. 6b, bottom-left panel. Finally, the chromatin distribution predicted in a WAPLΔ nucleus with inhibited transcription (WAPLΔ-DMSO-treatment) is shown in Fig. 6b, bottom-right panel. As shown in Fig. 3e and Fig. 3g, ActD (mathematically, $\Gamma_{tr} = 0$ in Eq. (5)) results in larger heterochromatin domains and thicker LADs, while WAPLΔ nuclei (increased cohesin loading; mathematically, $\Gamma_{ul}/\Gamma_l$ increases in Eq. (5)) show the opposite effect with smaller heterochromatin domains and LADs. For a WAPLΔ nuclei in which transcription is inhibited (WAPLΔ − ActD; mathematically, $\Gamma_{tr} = 0$ and $\Gamma_{ul}/\Gamma_l$ increases in Eq. (3)), the model predicts that inhibition of transcription returns the chromatin organization to the control (Cas9-ActD) levels. Transcription inhibition thus blocks the reduction in chromatin domain sizes induced due to WAPL deficiency due to lack of impetus for chromatin supercoiling.

To quantitatively validate the model predictions, we investigate the in-vivo chromatin organization under individual and tandem changes in transcription and cohesin unloading by re-analyzing previously reported super-resolution images shown as heatmap density plots in Fig. 6c[19]. Visual inspection of this data agrees with the model predictions that transcriptional inhibition counteracts the chromatin blending observed in DMSO treated WAPLΔ nuclei, which was also previously reported[19]. We thus focused on extracting the radius of heterochromatin domains and LAD thickness to further validate the model results quantitatively (Fig. 6d). Cas9 − ActD treated nuclei show an increased heterochromatin domain radius compared to control while WAPLΔ nuclei show a significant reduction in domain radius and LAD thickness (Fig. 6d). However, WAPLΔ − ActD treated nuclei show no significant difference in comparison to Cas9 − ActD treated nuclei (Fig. 6d), in quantitative agreement with the numerical predictions.

These results further confirm that the effect of transcription on global chromatin distribution occurs via supercoiling mediated chromatin loop extrusion, especially at the interface of heterochromatin and euchromatin phases. Furthermore, these results also present a significant validation of the mathematical phase-field model of chromatin organization in the nucleus.

## Discussion

Significant inroads into mechanistic modeling of chromatin organization as physically and functionally distinct states with finer architectural sub-features such as topologically associated domains (TADs) and chromatin loops have been made from a polymer physics perspective. Such models were developed with different levels of fine-graining to capture biophysics of chromatin organization at different length-scales spanning single or multiple nucleosomes[34–36], multiple nucleosome clutches[37–42], single and multiple TADs with salient sub-TAD features[43–46], single and multiple chromosomes[45,47–52] and the whole genome[53–56]. Depending on the focus on the chromatin functionalities or structure being simulated, any of these models can be adopted. For instance, first-principles thermodynamics driven approach may capture chromatin as a copolymer with two states, whereas a data-driven approach trained on conformation capture (e.g., Hi-C) or sequencing (e.g., CHIP-seq) data may incorporate over 50 states spanning the entire genome.

The experimentally observed role of RNAPII-mediated transcription in DNA supercoiling and subsequent loop extrusion[2,12,19,20,33,57–60] has also been studied using molecular dynamics simulations and polymer physics-based models at nanoscale[20,61–66]. However, quantitative predictions of sizes of heterochromatin domains which organize at a nucleus-wide meso-scale level are beyond the purview of such models. Furthermore, to the best of our knowledge, polymer models lack the far from equilibrium kinetic considerations of active epigenetic regulation, chromatin extrusion and diffusion kinetics, which we find are intricately involved in the spatiotemporal regulation of heterochromatin domain sizes. The current study, incorporating coarse-grained continuum model of chromatin organization at a mesoscale, presents the following advantages over previous polymer-based models:

a. Nucleus-wide characteristic size distribution of heterochromatic domains.

b. LADs of finite thickness co-existing with interior heterochromatin domains and their dynamic size-regulation.

c. The kinetic interplay of diffusion, epigenetic reactions and transcription in regulation of meso-scale organization.

Thus, here we present a non-equilibrium thermodynamic continuum model of the meso-scale chromatin organization in the nucleus to bridge the gap in the understanding of the mechanistic relation between transcriptional and epigenetic regulation and the size-scaling of the meso-scale heterochromatin domains. Our model incorporates the energetics of chromatin-chromatin interactions which is constructed as a double-well function allowing the phase-separation of chromatin into compartments of distinct compactions. Along the nuclear periphery, the effect of chromatin-anchoring proteins such as LAP2$\beta$ is captured via energetic chromatin-lamina interactions leading to the formation of LADs. Concomitant with the energetics, the chromatin organization is temporally driven by diffusion kinetics of nucleoplasm and the effective diffusion-like evolution of epigenetic marks. While the diffusion of nucleoplasm determines the level of chromatin compaction, such that higher local nucleoplasm content results in lesser chromatin compaction, diffusion of epigenetic marks results in accumulation of acetylated and methylated nucleosomes driving their segregation (Supplementary Fig. S4, Supplementary Sections S1.6, S1.7 of the Supplementary Information). Most importantly, we also account for the active reaction kinetics, which allow the interconversion of heterochromatin into euchromatin and vice-versa. The chromatin phase-interconversion can occur via the epigenetic

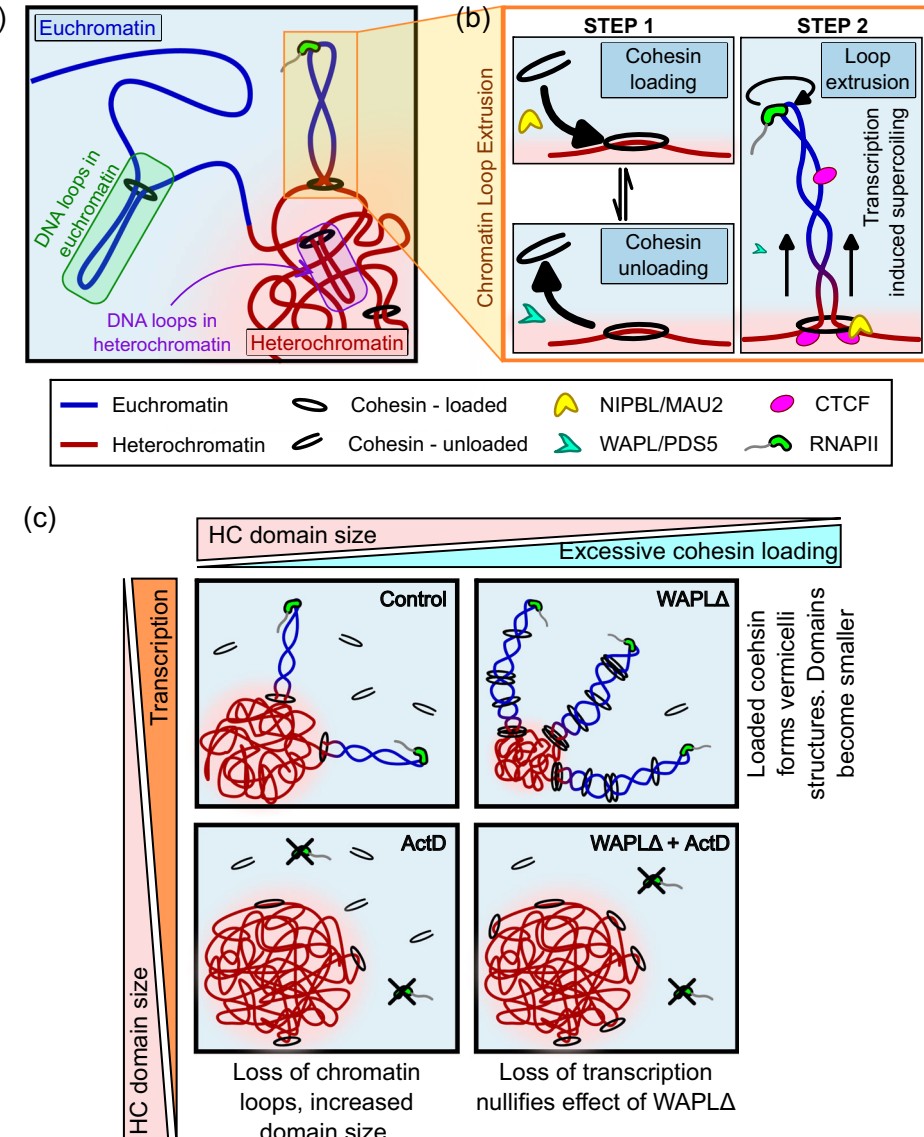

**Fig. 7 | Schematic summary of the effect of supercoiling mediated chromatin loop extrusion on chromatin distribution in the nucleus. a** A schematic of the roles of epigenetic and transcriptional regulation in chromatin compaction and **b** the steps in extrusion of a chromatin loop (Eq. 5). **c** Schematics showing the alterations in chromatin compaction and supercoiling mediated loop extrusion upon inhibition of the proteins involved in the individual steps of loop extrusion. Note that nucleosomes, despite being present, are not represented to better display the chromatin loops.

regulation of chromatin in the nucleus via the acetylation or methylation of the histones. Finally, to capture the role of transcription mediated supercoiling-driven loop extrusion in the determination of the heterochromatin domain sizes, we incorporate a kinetic conversion of compacted chromatin into transcriptionally active euchromatin in presence of RNAPII (Fig. 7a, b).

Together the active transcriptional kinetics and epigenetic regulation determine the active interconversion of hetero- and euchromatin, thereby taking the chromatin organization in the nucleus to a dynamic steady-state configuration. Specifically, our theoretical analysis reveals that the active reaction kinetics alone – independently of energetic interactions – offers a significant control over the average extent of chromatin compaction in the nucleus, thereby breaking the detail balance of thermodynamic equilibrium. At the meso-scale, spanning individual heterochromatin domains, we theoretically observe that the distribution of epigenetic marks relevant to chromatin compaction exhibit a radial gradient which would drive an inward heterochromatin flux leading to ripening of the

phase-separated domains. However, the presence of epigenetic and transcriptional regulation offers an opposition to the influx via – (a) acetylation of heterochromatin into euchromatin which is then pushed out via the diffusion of epigenetic marks, and (b) extrusion of loops of chromatin from the heterochromatin phase into euchromatin phase (refer Fig. 2e–g). The steady-state balance between the opposing fluxes leads to intrinsic emergence of a characteristic size-scaling of heterochromatin domains. Upon translating the theoretically and numerically obtained stable heterochromatin domain size-scale into physical dimensions (Supplementary Section S9, SI), we note that the predicted characteristic domain size is equivalent to that observed using multiple super-resolution imaging techniques[5,7,67–70]. It is essential to note that without the active chromatin phase interconversion – at thermodynamic equilibrium – no inherent size-scale of heterochromatin domains would be observed.

Thus, our model predicts that transcriptional activity, synergistically with epigenetic regulation, controls the size and morphologies of

the heterochromatin domains. The key predictions of the model as summarized in Fig. 7c are:

1. Upon transcriptional inhibition, the characteristic sizes of heterochromatin domains increase due to loss of supercoiling mediated DNA loop extrusion (Fig. 7c, left panels).
2. The increased size of heterochromatin domains upon transcription abrogation are also observed in the vicinity of the nuclear lamina.
3. Transcriptional inhibition leads to reduction of DNA in the euchromatic phase.
4. Conversely, upon increased loop extrusion due to excessive cohesin loading, the size of the heterochromatin domains reduces in the interior as well as periphery of the nucleus (Fig. 7c, top panels).
5. Transcriptional inhibition in nuclei with excessive cohesin loaded, results in loss of loop extrusion resulting in increased domain sizes (Fig. 7c, bottom right panel).

Being founded on fundamental non-equilibrium thermodynamic principles, the predictions made by our model are cell-type agnostic. To validate cell-type independence, complementary techniques such as Chrom-STEM (for high-resolution chromatin conformation imaging) and PWS (for high-throughput nano-scale sensitive live-cell imaging) are carried out for BJ fibroblast cells and multiple epithelial cancer cell-lines – U2OS, HeLa, A549 and HCT116. We found that in vivo alterations in chromatin organization under transcriptional inhibition conditions are consistent with our model's predictions across all studied cell lines. A quantitative analysis of previously reported[19] super-resolution STORM images of nuclei further gives a direct quantitative validation of the predicted effects of transcription abrogation on heterochromatin domain sizes. Of note, in all the reported cells, growth of condensed heterochromatin domains after ActD treatment is seen throughout the nucleus, including along the nuclear periphery where an increased LAD thickness is observed both in-silico and in cells. Lastly, our predictions on the changes in heterochromatin domain sizes upon over-extrusion of chromatin loops with and without transcription are quantitatively validated by domain size analysis of the previously reported[19] super-resolution STORM images of control and WAPLΔ nuclei after DMSO or ActD treatments.

In addition to imaging techniques using multiple modalities, previously reported observations of chromatin reorganization via chromatin conformation capture studies further confirm our model predictions[71,72]. While the loss of RNAPII only had a limited effect on the presence of chromatin loops, observed as off diagonal peaks on Hi-C contact maps[73–75], investigations of contact maps at a finer resolution using Micro-C[72,76,77] revealed the existence of RNAPII-associated chromatin loops which are indeed disrupted upon transcription inhibition. Specifically, in agreement with our predictions, it was reported that depletion of RNAPII decreased genome-wide histone acetylation (specifically H3K27ac) levels, reduced local chromatin accessibility, and lead to loss of chromatin loops upon RNAPII depletion[72]. Further, Hi-C contact maps revealed an increase in chromatin loops in WAPL deficient nuclei[71]. However, it should be noted that not all the loops observed using Hi-C or Micro-C were RNAPII associated.

Beyond transcription induced supercoiling-driven loop extrusion, cohesin itself can play an active role in the formation, extrusion and maintenance of chromatin loops at different physiological length scales. Loops of chromatin, identified as peaks in chromatin contact mapping techniques like Hi-C and Micro-C[9–11,72] are formed at a length scale of topologically associated domains (TADs) or below. Such loops are observed within both active A[72] and inactive B[78] compartments. The mechanism of extrusion of such loops, could be distinct from RNAPII transcription induced supercoiling-driven loop extrusion, such as cohesin subunit SMC motor activity[20,79,80] or a passive cohesin diffusion along the chromatin polymer[81,82]. These multiple mechanisms of loop formation could be convergently cooperating forming chromatin loops at multiple physiological length scales.

However, here we are specifically interested in the meso-scale roles played by supercoiling-driven extrusion of chromatin loops from the silenced heterochromatin phase into transcriptionally active euchromatin region (Fig. 7a). These loops are specifically considered since the interconversion of heterochromatin to euchromatin will alter the sizes of the heterochromatin domains. Recent experimental evidence, as well as computational models, present strong evidence in favor of DNA loop extrusion mediated by the RNAPII driven transcription induced DNA supercoiling[12,14,19,20,57,58,79,83]. Negatively supercoiled DNA regions are particularly rich in transcription start sites (TSS) with a strong correlation seen between transcription and supercoiling[30]. Indeed, super-resolution images show high presence of RNAPII at the heterochromatin-euchromatin phase boundaries where loops would extrude from heterochromatin into euchromatin phase[22].

Intriguingly, previous observations[19] also show that in HeLa nuclei WAPL deficiency introduces abnormalities in the peripheral distribution of lamin A/C. Since lamin A/C plays an integral role in the chromatin-lamina interactions via chromatin anchoring proteins such as LAP2$\beta$ and emerin, it can be conjectured that WAPL treatment may affect the LAD organization. In the current study we have ignored such effects focusing purely on the role of supercoiling mediated chromatin loop extrusion. Our model can be easily modified to address the LAD alterations by introducing WAPL deficiency dependent modulations in the chromatin lamina interaction parameter $V_L$ in Eq. (2). Experiment guided modifications in the model will further strengthen our predictions of LAD formation. Further, the transcriptional machinery involves a highly complicated multi-stage process comprising recruitment of multiple transcription factors, RNAPII and gene regulatory elements, we have assumed the cohesin loading and RNAPII mediated supercoiling to be the rate defining steps which thereby govern the timescale for chromatin loop extrusion. A more refined kinetic model of transcription and loop extrusion could possibly be incorporated to predict the spatiotemporal chromatin arrangement in the nucleus. However, even without these inclusions, we believe that our model lays a fundamental computational framework to better understand the mechanistic role of transcription, and in general chemo-mechanical cell-signaling, on the meso-scale chromatin organization.

## Methods
### Mathematical description of genomic organization in the nucleus
At the meso-scale, chromatin is organized into distinct transcriptionally dissimilar phases of euchromatin and heterochromatin as depicted schematically in Fig. 1a. We incorporate the energetic interactions between the nucleosomes depending on their epigenetic state, as discussed below. While on one hand entropic contributions push chromatin towards a homogenous organization, enthalpy arising from nucleosome-bridging via HP1 proteins[84], via ionic interactions within chromatin phases[85,86] or local activity[87] oppose it. The entropic-enthalpic competition comprising the chromatin-chromatin interactions drives the phase separation of chromatin domains. The emergent formation of domains occurring thermodynamically in our model is similar to the chromatin domains qualitatively postulated based on super-resolution imaging[5,68,69]. Near the nuclear periphery, heterochromatin can be further anchored to the nuclear lamina via anchoring proteins such as LAP2$\beta$[88–90]. This drives the formation of peripheral LADs. The interior and peripheral heterochromatin domain formation occurs spatiotemporally via free energy lowering diffusion of nucleoplasm, and diffusion-like evolution of acetylation or methylation marks on the histones. We incorporate the active interconversion between the eu- and heterochromatic phases in two ways: histone methylation

or acetylation reactions can change the epigenetic distribution or transcription mediated supercoiling-driven chromatin loop extrusion.

As shown in Fig. 1a, supercoiling-driven DNA loop extrusion, can occur broadly in two regions where RNAPII is present[23,68,69,91]: within the euchromatin domains (red dashed circle in Fig. 1a) or at the interface of heterochromatin and euchromatin phases (black circle in Fig. 1a). Since the chromatin extrusion in the euchromatin phase maintains its transcriptionally active status and does not lead to any significant mesoscale changes in the epigenetic distribution, we focus on the domain interface. The chromatin extrusion at the interface is instrumental in the regulation of size of heterochromatic domains at the periphery to form euchromatin.

## Free energy considerations for the hetero- and euchromatic phases

At any point $x$ in the nucleus, at a time $t$, we consider three nuclear constituents, namely the nucleoplasm and the two phases of chromatin, euchromatin and heterochromatin with their volume fractions (refer Supplementary Information SI, Supplementary Section S1.1 for detailed definition) $\phi_n(x,t)$, $\phi_e(x,t)$ and $\phi_h(x,t)$. We assume that these three constituents are space filling, and their volume fractions add up to unity, i.e., $\phi_e + \phi_h + \phi_n = 1$ (derived in SI, Supplementary Section S1.1). Hence, if the volume fractions of two of the constituents is known, the volume fraction of the third is determined by this constraint. The composition of the constituents can thus be defined in terms of two independent variables (refer to the methods for details) – (i) $\phi_n(x,t)$ volume fraction of the nucleoplasm, and (ii) $\phi_d(x,t) = \phi_h(x,t) - \phi_e(x,t)$ which is the difference of the volume fractions of heterochromatin and euchromatin. Note that $\phi_d < (>)0$ for the euchromatin (heterochromatin) rich phase and is therefore analogous to an order parameter. In terms of the chromatin composition variables, $\phi_n$ and $\phi_d$, the free energy density at any point $x$ can be expressed non-dimensionally as (refer Supplementary Section S1.5 for details on non-dimensionalization),

$$\widetilde{W} = \underbrace{\left[\phi_e^2 + \phi_h^2(\Phi_h^{max} - \phi_h)^2\right]}_{\text{chromatin−chromatin interactions}} - \underbrace{\widetilde{V}_L \phi_h e^{-\frac{d}{d_0}}}_{\text{chromatin−lamina interactions}}$$
$$+ \underbrace{\frac{\delta^2}{2}|\nabla\phi_n|^2 + \frac{\delta^2}{2}|\nabla\phi_d|^2}_{\text{Interfacial energy}} \tag{6}$$

The construction of the free energy density function is discussed in more detail in the Supplementary Section S1.2. The first term, which is a Flory-Huggins type free energy density for chromatin, defines the competition between the enthalpy of the chromatin-chromatin interactions and entropic contributions of chromatin configuration. We discuss the choice of the form of chromatin-chromatin energetic interactions, and its similarity to the Flory-Huggins form of free energy density in the Supplementary Section S1.3. This term gives rise to the double-well potential describing the energy landscape of the possible chromatin distribution. The potential surface is visualized in Fig. 1b as a contour plot with well locations as $\phi_h = 0$ (euchromatin phase) and $\phi_h = \phi_h^{max}$ (heterochromatin phase). The well towards the bottom in Fig. 1b corresponds to the heterochromatin phase with a low water content and a higher chromatin compaction.

The methylated histone tails in heterochromatin phase can mediate inter-chromatin interactions via chromatin cross-linkers such as HP1$\alpha$[92–94]. Such chromatin crosslinking lowers the enthalpy resulting in a heterochromatin phase well with a densely packed chromatin. On the other hand, the euchromatin well, corresponding to the energy minimum with a higher water content is marked with a more acetylated histone tails with a loosely packed chromatin conformation corresponding to a higher entropy.

The second term captures the interactions between the chromatin and the lamina via chromatin anchoring proteins (LAP2$\beta$, emerin, MAN1, etc.)[88–90] with parameter $\widetilde{V}_L$ denoting the rescaled strength of these anchoring interactions. Notably, these interactions are most robust at the nuclear periphery (distance from lamina $d = 0$) and vanish exponentially over a length scale $d_0$. Since the chromatin domains preferentially associating with the nuclear lamina are linked to transcriptional repression and an increased histone methylation[89,95–97], the chromatin-lamina interactions are captured specifically towards heterochromatin phase. Lastly, the negative sign permits an energetic preference for the peripheral association of heterochromatin. Analogous discrete descriptions of chromatin-lamina interactions, via formation of strong bonds when the chromatin is within a characteristic distance from the lamina have been previously implemented[50,51] in polymer models of chromatin, although without the epigenetic or transcriptional kinetics.

The last term accounts for the interfacial energy which is not accounted in a Flory-Huggins model and penalizes the formation of sharp interfaces between the dissimilar phases (refer Supplementary Sections S1.2 and S1.5). The interfacial penalty competes with the energy of chromatin-chromatin interactions forming smooth interfaces of non-dimensional width $\delta$ (Supplementary Section S1.5).

## Diffusion kinetics of the nucleoplasm

Thus, the energetic considerations dictate that an initial chromatin configuration (light blue circle in Fig. 1b) spontaneously phase-separates into the two energy wells to minimize the total free energy of the system. The driving force pushing the chromatin composition towards the energy wells is a measure of the gradients of the energy landscape and is called the chemical potential. Thus, the chemical potentials are obtained at each point in space by considering changes in energy density for small changes in the local volume fractions (labeled n or d): $\widetilde{\mu}_{n(d)}(x,t) = \frac{\delta W}{\delta \phi_{n(d)}}$, as derived in Supplementary Eq. (S12). Here, the operator $\delta$ denotes the functional derivative, or the change in free energy density with respect to the volume fraction. Spatial gradients of chemical potential drive the diffusive flow of nucleoplasm to reduce the overall free energy of the system giving rise to nucleoplasm kinetics via Supplementary Eq. (S6) (Supplementary Section S1.4). By rescaling the evolution equation, via the methodology described in Supplementary Section S1.5, we obtain the non-dimensional nucleoplasm kinetics as (Supplementary Eq. (S13),

$$\frac{\partial \phi_n}{\partial \widetilde{t}} = \nabla^2 \widetilde{\mu}_n \tag{7a}$$

Note that nucleoplasm diffusion kinetics in Eq. (7a) is conservative in nature, i.e., the net amount of water in the nucleus is conserved over time as long as no water enters or exits the nucleus.

## Reaction-driven spatiotemporal kinetics of histone marks

The kinetics of epigenetic marks on the histones – acetylation or methylation – can have two contributions. Primarily, the epigenetic regulation via enzymes such as histone deacetylase (HDAC), histone methyltransferase (HMT), histone acetyltransferase (HAT) and histone demethylase (HDM) can result in interconversion of heterochromatin and euchromatin phases via acetylation and methylation reactions as shown in Supplementary Fig. S3. The reaction kinetics, inherently non-conservative (discussed in Supplementary Section S1.4) are captured via the second term in Eq. (7b').

The reaction kinetics however should also incorporate the contribution of chromatin-chromatin interactions, which determine how favorable the euchromatin-heterochromatin interconversion is depending on the epigenetic marks on the other nucleosomes in vicinity. In the SI, we qualitatively (Supplementary Section S1.6) and theoretically (Supplementary Section S1.7) describe how neighborhood dependent reaction-kinetics is effectively equivalent to diffusion-

like evolution of epigenetic marks, which we incorporate in our model as 'diffusion of epigenetic marks' (first term in Eq. 7b'). Thus, the reaction kinetics also give rise to an effectively conservative contribution which allows for evolution of epigenetic marks without changing the overall amounts of heterochromatin and euchromatin in the nucleus.

Lastly, the kinetics of transcription-mediated supercoiling-driven chromatin extrusion localized at the heterochromatin domain boundaries is incorporated into the dynamics of epigenetic marks, giving rise to a second evolution equation of the form,

$$\frac{\partial \phi_d}{\partial \tilde{t}} = \underbrace{\nabla^2 \tilde{\mu}_d}_{\text{Diffusion of epigenetic marks}} + \underbrace{2\left(\tilde{\Gamma}_{me}\phi_e - \phi_h\right)}_{\text{Epigenetic regulation}} - \underbrace{2\tilde{\Gamma}_a(\tilde{x})\phi_h}_{\text{Active chromatin loop extrusion}}$$

(7b')

Note that Eq. (7b') is non-dimensionalized by rescaling all time variables with respect to the rate of histone tail acetylation (i.e., $\tilde{t} = \Gamma_{ac}t$) and all spatial variables with respect to the characteristic reaction diffusion length defined in Supplementary Section S1.5 (i.e., $\tilde{x} = x/\ell_{RD}$). The second term in Eq. (7b') incorporates active first-order reaction kinetics of histone tail acetylation (see Eqs. (S7) and (S8) before rescaling) and that of histone methylation $\tilde{\Gamma}_{me}$ leading to interconversion between hetero- and eu-chromatin.

The last term in Eq. (7b') accounts for the supercoiling-driven chromatin extrusion kinetics and the chromatin state changes resulting from it (Fig. 1c). Being transcription mediated, the kinetic rate of supercoiling-driven extrusion $\tilde{\Gamma}_a(\tilde{x})$ must be spatially dependent on local availability of RNAPII, which is prominently present at the boundaries of the compacted heterochromatin phase[23,68,69]. Although supercoiling-driven loop extrusion may also occur within the euchromatin phase, it does not contribute to interconversion of chromatin phases as euchromatin is already transcriptionally active. In contrast, at the interface of heterochromatin and euchromatin, supercoiling-driven loop extrusion can result in activation of otherwise inactive genes. Considering this spatial localization to the heterochromatin domain boundaries, we rewrite Eq. (7b') as,

$$\frac{\partial \phi_d}{\partial \tilde{t}} = \underbrace{\nabla^2 \tilde{\mu}_d}_{\text{diffusion}} + 2\left(\underbrace{\tilde{\Gamma}_{me}\phi_e - \phi_h}_{\substack{\text{epigenetic}\\\text{regulation}}} - \underbrace{\tilde{\Gamma}_a e^{-\left(\frac{\phi_h - \frac{\phi_h^{\max}}{2}}{2\Delta\phi}\right)^2}\phi_h}_{\substack{\text{active chromatin}\\\text{extrusion}}}\right)$$

(7b)

Note that $\phi_h^{\max}/2$ is the volume fraction of heterochromatin at the domain boundary. A deviation of $\Delta\phi$ from this value defines the width of the domain boundary, and the supercoiling-driven loop extrusion is spatially restricted to a narrow region at the boundary of heterochromatin domains. The last two terms of Eq. (7b) are responsible for the non-conservative dynamics and can alter the global heterochromatin to euchromatin ratio of the system. More detailed derivation of the chemical potential, contribution of passive diffusion kinetics, epigenetic and active loop extrusion can be found in the extended methods section in the SI (Supplementary Sections S1.1–S1.7).

Having developed the model to capture the spatiotemporal organization of chromatin in the nucleus, we numerically solve Eqs. (7a) and (7b) along with the equation defining the chemical potential (Supplementary Eq. (S3)). As a boundary condition we ensure no exchange of water and chromatin between the nucleus and the surroundings. This condition can be suitably adjusted to allow flow of water from or into the nucleus. The parameters used in the model along with the initial and boundary conditions are described in detail in the SI (Supplementary Section S8) and listed in Supplementary

Table S2. Note that the epigenetic rates $\tilde{\Gamma}_{me}$ and the strength of chromatin-lamina affinity $\tilde{V}_L$ are not modified throughout any of the simulations carried out, unless explicitly stated. This is to ensure that any predicted changes in chromatin organization occur specifically due to changes in supercoiling-driven chromatin loop extrusion.

### Reporting summary
Further information on research design is available in the Nature Portfolio Reporting Summary linked to this article.

## Data availability
The data supporting the findings of this study are available from the corresponding authors upon request. The data generated in this study are provided in the Source Data file.

## Code availability
The code used for measurement of sizes of heterochromatin domain obtained from STORM imaging is freely available through github (https://github.com/ShenoyLab/STORM_Analysis)[98]. The Python module for PWS acquisition and analysis is also publicly available on GitHub (https://github.com/BackmanLab/PWSpy).

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

## Acknowledgements

This work was supported by NIH Award U54CA261694 (V.B.S.); NCI Awards R01CA232256 (V.B.S.); NSF CEMB Grant CMMI-154857 (V.B.S.); NSF Grants MRSEC/DMR-1720530 and DMS-1953572 (V.B.S.); NIBIB Awards R01EB017753 and R01EB030876 (V.B.S.). This work made use of the BioCryo facility of Northwestern University's NUANCE Center, which has received support from the SHyNE Resource (NSF ECCS-2025633), the IIN, and Northwestern's MRSEC program (NSF DMR-1720139). We acknowledge the support from NIH grants U54CA268084 (V.B.), R01CA228272 (V.B.), R01CA225002 (V.B.), NSF grant EFMA-1830961 (V.B.), and philanthropic support from Rob and Kristin Goldman (V.B.). We acknowledge the support from the European Union's Horizon 2020 Research and Innovation Programme (no. 964342 to M.P.C.); Ministerio de Ciencia e Innovación (grant no. 008506-PID2020- 114080GB-I00 to M.P.C.; AGAUR grant from Secretaria d'Universitats i Recerca del Departament d'Empresa iConeixement de la Generalitat de Catalunya (grant no. 006712 BFU2017-86760-P (AEI/FEDER, UE) to M.P.C.).

## Author contributions

A.K. and V.B.S. conceived the chemo-mechanical model and carried out the theoretical and numerical analyses. A.K., V.V. and V.B.S. performed the polymer equivalence analysis. M.V.N, M.L. and M.P.C carried out the STORM imaging. A.K. and Z.G. performed the quantitative analysis of STORM images. W.S.L., V.A., E.P., L.A and V.B. carried out the Chrom-STEM and PWS imaging and analyzed the data. A.K., Z.G., V.V., M.V.N., W.S.L., V.A., E.P., L.A., V.B., M.L., M.P.C. and V.B.S wrote the paper.

## Competing interests

The authors declare no competing interests.
