## [Peer Review File · Nature Communications]

Active Transcription and Epigenetic Reactions Synergistically Regulate Meso-Scale Genomic OrganizationREVIEWER COMMENTS

Reviewer #1 (Remarks to the Author):

It is now well known that cell-state specific genome regulation correlates with the non-random organization of the chromatin. However, we are yet to learn the interplay of the chromatin organization and the genome regulation, which includes genetic transcription, in a comprehensive way. The current manuscript presents an important piece of work towards addressing that grand research question. It conceives a meso-scopic theoretical continuum model which incorporates (i) chromatin-chromatin and chromatin-lamina interactions, (ii) the reaction kinetics of epigenetic modifications leading to interconversion of eu- and heterochromatic segments of the chromatin, along with the diffusion of the several constituents of the system, and (iii) enzymatic activity of a type of transcription machinery, RNAP-II, that converts heterochromatic segments to euchromatic type. Using this fundamentally out-of-equilibrium model, the authors exhibit how the interplay of the above-mentioned model elements determine the mesoscopic chromatin organization. Their theory predicts that a balance between diffusive and reactive fluxes underly the mesoscopic chromatin organization with characteristic microphase separated morphology determining the length-scale of the heterochromatin domains. They supplement their theoretical study with experimental data obtained from several cell-lines using complementary experimental approaches. Both the theoretical and the experimental results presented in the current manuscript support the conclusions drawn, the research methodologies are upto the mark of the current standards in the research community, the knowledge gained from this research is beneficial for the field. Altogether, this is a good piece of work. Unfortunately, the current manuscript full of typographical errors, mis-references to the sub-figures and sections, inconsistent symbols. So, I suggest a careful revision of the manuscript before its publication. The authors may pay attention to the points mentioned below.

1. I did not understand how the authors conceive the mathematical expression for the chromatin-chromatin interactions (the first term on the right hand side, R.H.S.) in Eq. 1? Do they write it phenomenologically or deduce it from something else? They describe it as a term similar to Flory-Huggins terms. However, this comparison is not straight forward to me. It is better to explicitly describe how their term compare with the Flory-Huggins terms, especially, focusing on the enthalpic and entropic contributions. Also, in col. 74, they describe their theoretical model as a polymer physics-based one. Since, I did not understand the first term on R.H.S. of Eq. 1, I am doubtful about this statement.
2. The major obstruction in following this manuscript is its mis-references and inconsistent usage of symbols. See for example Eq. 1 and compare it with Eqs. S1 and S1' in the supplementary information (SI). Δ is not defined in the main text, ϕ_w should be read as ϕ_n , the tilde over W is missing in Fig. 1b (I guess, if otherwise, please mention it in the caption). There is a repetition of text in col. 150. Check all the texts where subfigures have been referenced (e.g., Fig. 2e should be Fig. 2f on col. 365, and many others). There are two Eq. 6 in the manuscript. I guess, the authors meant to refer the extended methods in SI on col. 144, 212, etc. There is no Sec. 2.5 in methods as being referred to on col. 427. The authors should check the SI as well (e.g., Eq. 13 has been cited on col. 236 of SI, which I could not find). Please note that, I do not intend to provide a complete list of this kind of errors. Better, the authors do a thorough revision of all the text including the figure-captions, check its consistency with the figures, check individual equations, the symbols, and whether the symbols have been defined in the main text (e.g., the rescaled quantity Γ^{\sim}_{me} should be defined in the main text).
3. The phrase 'diffusion of epigenetic marks' may not be obvious to all the readers. It has been defined in sec. 2.3-2.4; however, used earlier (see col. 78). The authors may think about better way of presentation on this matter.
4. I understand that only the chromatin-chromatin interactions has been considered to draw the contour plot in Fig. 1b. But, I do not understand what the authors meant to say by referring to it in the sentence ending on col. 334.
5. I do not understand what the authors meant by 'relative' in Fig. 2g. It has been referred as steady-

state radius in Fig. 2f (wrong numbering I believe) on col. 376.

6. Compare the first the Eq. 6 on col. 659 with the expression written in Fig. 6a.

7. I believe some of the preprints cited in the manuscript might have already published.

Reviewer #2 (Remarks to the Author):

General Comments

1) In contrast to present polymer theories of nuclear genome structure and their relation to transcriptional activity, this appears to be the first large-scale attempt to quantitatively predict essential structural features of the condensation/decondensation of nuclear chromatin caused synergistically by epigenetic markers and transcriptional complexes (corresponding to some qualitative models previously discussed in the Literature) within the framework of a unified differential equation of the chemical potential of the entire nuclear genome. This appears also to be the first time that the role of epigenetic structure related elements is postulated to be equivalent to the biochemistry of transcription: Accordingly, genome structure is no longer regarded as a "downstream" phenomenon of transcription processes, but emerges to be also essential for gene regulation. Its results are in accordance with a wealth of experimental results obtained by both electron and super-resolution microscopy.

2) For the readership of Nature Communications (mostly with a biological/biochemical background) it should be helpful to add a short section indicating this highly innovative perspectives, and also to refer to previous qualitative models referred to in the literature.

3) The Theoretical Physics effort used for this purpose is impressive; it is understandable that the detailed theoretical description given in the Supplementary Material will mostly be reserved to specialists in the field. In the main text, however, it might be helpful for readers with an experimental background to explain in a more „basic language“ the general highlights of the theory. In the Supplementary Information, a list of the mathematical symbols (well known to Physicists but perhaps not all familiar to others) might be added.

4) A peculiarity of the Chemical Potential theory of nuclear chromatin is that so far no sequence-specific distinctions are made. In order to still come up with something useful (decondensation of the surface of domains upon transcription, condensation without transcription), as a starting point the authors assume a qualitative model very similar to Miron et al. (2020) and other papers in the literature as an initial condition, however without reference to them. To convince the biologically/biochemically oriented reader, it should be helpful to explain a little bit more the assumptions made, and to quote some of the earlier, qualitative models some of them fitting well to the results of this MS.

5) The impressive differential equation for the chemical potential of the nucleus in its present form yielded usable solutions only by additionally applying a set of experimentally determined parameters that were, moreover, "dimensionless": This means that, for example, only relative densities of DNA in the domains (e.g., radial relative distribution) could be calculated; thus, absolute densities (Mbp/ μm^3) could not be given. However, according to polymer based numerical simulations (e.g. Maeshima et al., 2015), as well as to qualitative models referred to in the literature (so far not cited by the authors), absolute densities of chromatin domains may control the accessibility of macromolecular complexes to target sequences essential for the initiation of transcription.

6) A first very rough estimate of the absolute densities relevant for the theory presented might already be derived from the initial conditions assumed by the authors that the heterochromatic

domains have such a high density that even water cannot diffuse. From this (and also from the schematic images shown) it is clear that the assumed density of the compact domains will be at least 200 Mbp/ μm^3 , i.e. such domains should be inaccessible for macromolecules. Hence, transcription must take place at the surface of the domains, as the authors also explicitly postulate.

7) Recent published reports of experimental measurements of the absolute DNA density in compact chromatin domains obtained by single molecule localization microscopy (SMLM) revealed a range of absolute densities sufficient to allow an accessibility controlled fine regulation of transcription. Altogether, the basic theoretical approach presented offers novel and fascinating perspectives for a substantially improved understanding of functional nuclear genome structure, based on a rigid application of the fundamental laws of Physics and Chemistry.

From the side of "theoretical nuclear genome physics", there are thus two basic approaches that can be further developed:

- a) Numerical models based on polymer chains (developed e.g. by the Nicodemi, Safran, de Rosas, Everaers labs); and
- b) Differential equation systems built on the thermodynamics of chemical potentials, as presented in this MS.

Within their specific limits, both quantitative modelling approaches will be extremely valuable to provide a theoretical frame to experimental nuclear genome structure research. Another, most important perspective will be the wealth of experimentally testable predictions (an example for this documented also in this ms). This will not only stimulate the application of present methodology, from Hi-C to electron microscopy and super-resolution, but also the development of new experimental approaches.

Eventually, this may even contribute to the development of epigenetic pharmaceuticals, e.g. by specifically changing transcription related chromatin compaction. It might be helpful for the reader to add a short section on the comparison of the Chemical Potential model to polymer based theories.

Specific Comments

- 1) „Transcription-driven loop extrusion“: This is a favorite topic of biochemically oriented papers. However, models shown on this subject in the literature are extremely “qualitative”. The more quantitative nanostructural data are made available, the more realistic such models should become; eventually it should be possible to understand what the biochemically inspired word „loop extrusion“ really means in terms of space and time.
- 2) “Loss of transcription results in increased chromatin compaction and larger heterochromatin domain sizes“: This result would fit extremely well to earlier qualitative models described in the literature which postulate that the probability of transcription and chromatin compaction is intimately correlated.
- 3) „The existence of a multiscale chromatin organization is well-established [1, 2]“: Both initial references describe well the present state of biochemical analysis (Hi-C, Cohesin loops etc.) but do not provide an adequate description of the essential contributions of microscopy. Some additional references might be helpful.
- 4) Altogether, the mathematical theory presented looks very interesting as a first attempt to describe the space-time structure of the nuclear genome in terms of a general theory based on Thermodynamics. To allow some meaningful testable conclusions, some radical “boundary” assumptions had to be made. It may be useful to provide a little more technical information about the calculations made.

5) An alternative basic approach to the Chemical Potential theory would be a more generalized polymeric theory. For example, using Monte Carlo approximations, relations between absolute chromatin density and accessibility (i.e. between Euchromatin and Heterochromatin) may be predicted; assuming interaction potentials between a polymeric chromatin chain and lamins, structural predictions of the distribution of heterochromatin and euchromatin in the nucleus may be calculated (e.g. Bajpai et al. 2021; Amiad-Pavlov et al., 2021): one might add to such schemes also diffusion and binding characteristics of epigenetic factors etc. In the discussion, some references to such alternative theoretical approaches might be useful. Which model is adapted to solve specific problems of nuclear organization has to be explored in future studies. Probably both will have their importance. Concerning this ms, it is suggested to add a section to discuss the relationship of the Chemical Potential model chosen to Polymer Theory approaches.

6) Maintext & Supplementary Material: How is the "local" volume fraction defined? Is it the ratio of e.g. [nucleoplasm volume]/[nuclear volume]? Or is it ratio of [nucleoplasm volume]/[domain volume]? Generally, the legibility of the ms might be improved by a little more „redundancy“ in the explanation of the parameters shown (if due to length limits this is not possible in the main text, it might be done in the Supplementary Information).

7) To use the thermodynamic theory of free energy density is very elegant. The consequence of such a general theory using differential operators is that in principle it assumes a very large number of molecules and does not consider the special conditions of molecular configurations (e.g. number of base pairs; number of H2B molecules, proteins, loops etc.). Hence it is expected that the theory by itself will be able to describe general relationships (e.g. relative densities) but not specific molecular conditions. This might be obtained by experimental calibration, e.g. to translate relative radial intensity distributions of compact domains into absolute ones (in terms of Mbp/ μm^3).

8) The assumption of a water devoid heterochromatin phase would physically be possible only at extremely high absolute densities (estimated to be in the range above 200 Mbp/ μm^3). This would imply

a) that only a small part of the nuclear volume is in this state: Assuming e.g. a density of 200 Mbp/ μm^3 and a total cellular DNA content of 6,000 Mbp, the heterochromatin volume $V = [\text{DNA content}]/\text{density}$ would be $6,000/200 \mu\text{m}^3 = 30 \mu\text{m}^3$, i.e. only a few percent of a typical nuclear volume;

b) that target sequences in the interior of such extremely compact structures would definitely be excluded from any accessibility of transcription factor complexes. In contrast to these considerations, the schematic figures shown suggest a lot of space even between the chromatin fibers in the interior of heterochromatic domains. This appears to be correct if one assumes that the DNA chain is shown without the nucleosomes; but this may not provide an idea of the real density relations which are of utmost importance for accessibility. Hence it should be helpful to state in the legend whether DNA or chromatin fibers (nucleosome chains) are intended.

9) „Interactions between the chromatin and the lamina“: A reference to the polymeric chromatin-lamin interaction model of Bajpai et al. (2021) and Amiad-Pavlov et al. (2021) may be helpful. An advantage of these and related polymer theories to predict the consequences of chromatin-lamin interactions is that their approach is based on a discrete polymer chain with a large but finite number of chromatin particles ("beads"), each "bead" representing a definite small amount of DNA which may be given in absolute units (No. of bp, No. of nucleosomes). On the other side, so far such polymeric models regarded only chromatin-chromatin and chromatin lamin interactions and hence did not include explicitly epigenetic diffusing factors, as in the present ms. In the future, both approaches will probably be valid, depending on the special problems to be solved.

10) How are the small heterochromatic domains in the nuclear interior obtained? By assuming a certain amount of lamina also in the nuclear interior?

11) Maintext & Supplementary Material, „energy penalty associated with forming phase boundaries between the euchromatin and heterochromatin phases“: Might this term be used to simulate various degrees of "chromosome territory intermingling" (see e.g. Pombo et al. 2006). For this, it would be required to specify the "chromatin particles" belonging to a given CT. Generally, it appears to be that in the Chemical Potential theory presented, the existence of chromosome territories (CTs) is neglected. Would it be possible to extend the theory in such a way that individual chromatin particles are assigned to specific CTs (e.g. by assuming that most of the chromatin particles of e.g. #1 form a cluster with a radius of 1 μm in a nuclear volume of radius 5 μm ?).

12) Maintext & Supplementary Material: Might a quantitative theory of absolute fluxes (in terms of water volume (nm^3) per unit surface area (μm^2) be established on the basis of diffusion simulations using experimental absolute density chromatin distributions ?

13) Maintext & Supplementary Material: What would be required to provide time scales also in absolute terms (s)?

14) Conversion of Euchromatin to Heterochromatin and vice vs.: This theory of transition of heterochromatin into euchromatin appears to be based on the assumption that the chromatin modelling factors responsible are not substantially impaired by the extremely high density assumed for heterochromatic domains. A possibility to understand this in the context of the theory given would be to assume that the decondensation process starts at the surface of the heterochromatic domains (for a previous qualitative model of this see e.g. Miron et al. 2020). Such a model would also nicely explain the observation of increased condensation of heterochromatic domains: To function on the surface of dense domains, transcription complexes would have to disentangle somewhat the surface chromatin, as shown in Miron et al. (2020), Fig. 6. In the absence of such factors, chromatin-chromatin interactions would result in a "collapse" of these external loops and hence be visible as increased condensation. The model presented by Miron et al. (2020) as well as related qualitative models presented in the literature also suggest a mechanism how the inhibition of transcription may result in an increase of the size of heterochromatic domains.

15) „Time-dependent mathematical set of governing equations describing the spatio-temporal evolution of chromatin organization in the nucleus“: In the equation presented, the epigenetic regulation appears to be restricted to two factors (for methylation and acetylation). Would it be possible to extend the theory to include also other epigenetic factors, like other DNA/histone modellers, Polycomb proteins etc.?

16) „Rescaling the governing equations“: A non-dimensional set of governing equations means that the results are also intrinsically non-dimensional, i.e. provide relations but not absolute figures (e.g. on absolute densities, e.g. in terms of $\text{Mbp}/\mu\text{m}^3$). What experimentally determined parameters would be required to translate non-dimensional predictions into absolute dimensional ones ?

17) „Polymer analogy of the roles played by reaction and diffusion kinetics“: The conservation of the net amount of the DNA of heterochromatin plus euchromatin in the context of the theory presented appears to be evident for a given state of the cell cycle (e.g. G1); but what happens during G2 ? Would it be possible to extend the theory to cell states with replicating DNA ? Since in G2 the number of DNA bp changes, also the terms for chromatin-chromatin interactions and for chromatin-lamin interaction in the equation should be modified. Assuming that the total DNA content remains constant: What is

the expected outcome if other molecule numbers change, e.g. of epigenetic diffusing marks (for example methylated or acetylated histone variants and proteins involved): How would this modify the ratio between euchromatin and heterochromatin and their spatial distribution? This would be highly interesting, e.g. with respect to the known change in methylation during ageing. In other words: Would it in principle be possible by this theory to predict age dependent nuclear genome structure?

18) Schematic Figures: If e.g. in Figure S2 one follows the trajectories of an individual interphase chromosome, one gets the impression that a the enveloping surface of a specific chromosome territory may fill a large part of the nuclear volume, which is in contradiction with the results of experimental observations. To reduce such misleading impressions (often found in biochemical papers on chromatin organisation), the fibers may be just appropriately shortened.

19) „Conservation of epigenetic factors via Eq (S5) dictates that the total amount of DNA in heterochromatic or euchromatic phase in the nucleus does not change“: This would describe a given transcriptional state of a nucleus. But what happens if gene activity changes ? If by such activation the domains with the sequences to be transcribed have to be made accessible (see e.g. the Model in Miron et al., 2020, or other models, e.g. ANC/INC model, BioEssays 2020), and hence decondensation occurs, THEN the amount of Euchromatin will increase and the amount of heterochromatin will diminish.

20) „Relocalization of methylation or acetylation histones without changing the overall amounts of heterochromatin and euchromatin“: This appears to be plausible under the assumption that the number of epigenetic factors remains constant, and all chromatin particles have the same interaction potentials. This is a valid first assumption to predict the general euchromatin-euchromatin distribution. However, how to integrate the fact that cell specific genes are turned on/off in a specific way ? Would it be possible to create instead of one equation a set of coupled differential equations for each set of „particles“ (according to their specific activity status) ?

21) Fig. S2: Using the assumption of very compact heterochromatin domains excluding even water (as indicated in Figure S2a below, then in the schematic depiction of the nuclear DNA density landscape (Fig. S2 abc above), the heterochromatin blocks may graphically better be indicated by homogeneously stained red regions, instead of Spaghetti fibers.

22) „In addition to methylation and acetylation, transcriptional activity also regulates the sizes of the heterochromatin domains“: Is transcriptional activity regulating the size of the domains; or does the size/structure of the domains regulate transcriptional activity? In other words: Is chromatin nanostructure modification a consequence of the biochemical reactions involved in transcription ? Or is the initiation of transcription a consequence of the chromatin nanostructure modification? (e.g. by allowing increased accessibility by decompaction to a target sequence in the interior). In the first case, chromatin structure changes would be just a "down stream effect" produced by otherwise functionally independent biochemical reactions; in the second, the domain nanostructure would play an essential role; chromatin nanostructure would be of fundamental importance for the understanding of transcriptional regulation.

23) List of parameters: Altogether, the parameters used to solve the Chemical Potential equation were chosen to fit the basic experimental expectations. This is of course a valid first approach. The dimension less parameter values indicate that in this way, relative distributions will be obtained. The model would become even more interesting, if it should become possible to introduce absolute values, e.g. the absolute DNA content, the absolute amount/density of H2B, the absolute number of chromatin-chromatin interactions and chromatin-lamin interactions, the absolute number of epigenetic factors etc. Using appropriately calibrated SMLM approaches, this should become possible at least for some basic parameters, e.g. for the absolute density of eu- and heterochromatin domains (see e.g. a recent analysis in Cell Reports 2023). One might argue that eventually also chromatin structure is regulated by the fundamental laws of Physics and Chemistry. This is of course true. But under

evolutionary pressure chromatin nanostructure may have developed in such a way to allow an optimum of transcriptionally useful dynamic chromatin structures.

Recommendation

Nuclear chromatin structure and its relation to gene regulation has emerged as one of the most challenging topics of modern cell biology; for a better understanding of the mechanisms involved, theoretical contributions to this field are highly welcome. Due to the importance of developing theoretical approaches to understand the synergistic action of transcription and epigenetics, a revised version of the ms is recommended for publication.

Reviewer #3 (Remarks to the Author):

This manuscript describes a mesoscale whole-nucleus reaction-diffusion model of how chromatin is packed into spatial domains of active and inactive chromatin with various sizes, and how perturbations can alter this organization. The model incorporates: chromatin-chromatin interactions, heterochromatin-lamina interactions, interfacial energy, and kinetic parameters controlling the exchange between heterochromatin and euchromatin. From these, the model generates spatial distributions for heterochromatin, euchromatin, and nucleoplasm and how they evolve over time. The authors compare simulated spatial domain distribution of domains to those obtained experimentally from five cell lines and two imaging technologies, chromSTEM and STORM.

Meso-scale biophysical models are of great interest for the interpretation of perturbations on genome folding. In general, the details of this model are nicely presented with both schematics and annotated equations. However, the current presentation is confusing for several major reasons, as well as important missing details.

Major points

First, the proposed model assumes that loop extrusion converts euchromatin (ϕ_e) into heterochromatin (ϕ_h). This is not consistent with descriptions of loop extrusion in previous polymer modeling literature, or in related biology experiments. In previous polymer modeling descriptions of extrusion and phase separation (e.g. Nuebler et al., <https://pubmed.ncbi.nlm.nih.gov/29967174/>, Conte et al. <https://pubmed.ncbi.nlm.nih.gov/35831310/>, Salari et al., <https://genome.cshlp.org/content/32/1/28.full>), the total amount of heterochromatin or euchromatin is not changed by the level of loop extrusion: what is altered is their mixing. That description aligns with experimental results for depletion of cohesin (e.g. Vian et al. <https://pubmed.ncbi.nlm.nih.gov/29706548/>, Schwarzer et al., <https://pubmed.ncbi.nlm.nih.gov/29094699/>), which do not show more inactive chromatin, but instead show a de-mixing of active and inactive chromatin. While the model presented here can make interesting predictions, it will be confusing to the same words to describe a distinct process. Much clarity could be gained if the authors use a different term to refer to the process modeled throughout the paper (e.g. "active interfacial heterochromatin conversion"). Drawing conclusions about the processes referred to as loop extrusion in the existing literature would require extensive comparative analysis between the models here and previous models in the literature. While such an extensive analysis may be beyond the scope of this paper, clarity and consistency with prior literature should not be.

The biological motivation behind the assumption that cohesin converts euchromatin into heterochromatin at phase boundaries is also unclear. Recent work (Zhang et al., <https://www.nature.com/articles/s41588-023-01364-4>) shows that RNAPII counteracts extrusion as assayed by Hi-C. Other recent work shows experimental Hi-C data are consistent with cohesin

modifying folding inside the heterochromatin domains, and not just at their edges (<https://www.nature.com/articles/s41594-022-00892-7>). The authors should address such existing literature when motivating their modeling approach and presenting their conclusions.

Finally, the authors do not clearly show how their analysis support changes in extrusion versus changes in the rate of acetylation following transcription inhibition. The model's similar dependence on acetylation and active chromatin extrusion (Γ_{ac} and Γ_a equation 2b), raises uncertainty if observed changes following ActD treatment are due to lowered extrusion versus lowered acetylation. Are there differences in distribution of size of modeled heterochromatin domains in these two scenarios? To draw firm conclusions, it would be informative to decrease the expected activity of extrusion (e.g. via RAD21 depletion) and then inhibiting transcription. Such model inquiries and experiments may be beyond the scope of this study, but without it the authors should heavily revise how their conclusions are presented.

Key details:

- as numerical solution of the reaction-diffusion PDEs are central to this manuscript, code and software used should be specified in the manuscript and supplement.
- the reporting summary has a link to a GitHub repository, but the repository does not provide an explanation of how the files could be used. This repository should include a sample dataset of how an image is processed. An example simulated image would be helpful as well. Image data can be stored on OSF, Zenodo, or elsewhere.
- it would be helpful to justify the exponential decay used to model the chromatin-lamina interaction.
- it is important to indicate the resolution in nanometers at which they model chromatin or solve their equations.
- while experimental data appears to display a broad range of domain sizes, the model appears to predict domains with a characteristic size. How can this discrepancy be resolved? If it cannot with the existing approach, the authors should mention this as a limitation in the discussion.
- the distribution of chromatin is not necessarily at steady-state. Can the authors provide evidence that this is a reasonable simplifying assumption?
- if Γ_{tr} is small but non-zero, then Γ_a for combined WAPL and ActD should be smaller than AcdD alone, and domain sizes may be further increased. The authors should discuss how their conclusions would differ if this is the case.

Minor details:

- 2.4: Unclear what authors mean by: "However, locally, the diffusion kinetics can result in exchange of epigenetic marks – either acetylation or methylation – between neighboring nucleosomes, as shown in the inset in Figure S1. ". Is it really the nucleosomes that are changing from diffusion or the spatial density / positions of the two types of marks?
- 2.4: the spatial dependence (x) in $\Gamma_a(x)$, is easy to miss, and it would be useful to emphasize that this makes this term distinct from the epigenetic regulation term in the first sentence after equation 2b.
- Page 12 line 12: instead of figure 2f it should be figure 2g.
- Figure 4b-e: It is not clear how the blue line in experiments, which stays quite high, is analogous to the blue line in simulations. It is also not clear why (e) indicates euchromatin rather than total DNA (heterochromatin + euchromatin) as seems to be plotted for experiments (caption for (a)).
- S11: γ_a is in the legend, but γ_{me} is in the panel.

We would like to thank the referees for their careful review of our manuscript and for finding our work interesting. We found the comments/suggestions of the referees very useful. We have carefully considered each comment and have provided a detailed point-by-point response to them. We feel that the additional information we have now included in the revised manuscript and the supplementary information addresses all the concerns raised by the referees. We firmly believe that the revised MS is stronger because of the reviewers' suggestions.

(The reviewer comments are in blue, our response in plain text, and the edits made to the manuscript are highlighted in yellow)

REVIEWER #1

Remarks to the Author: It is now well known that cell-state specific genome regulation correlates with the non-random organization of the chromatin. However, we are yet to learn the interplay of the chromatin organization and the genome regulation, which includes genetic transcription, in a comprehensive way. The current manuscript presents an important piece of work towards addressing that grand research question. It conceives a meso-scopic theoretical continuum model which incorporates (i) chromatin-chromatin and chromatin-lamina interactions, (ii) the reaction kinetics of epigenetic modifications leading to interconversion of eu- and heterochromatic segments of the chromatin, along with the diffusion of the several constituents of the system, and (iii) enzymatic activity of a type of transcription machinery, RNAP-II, that converts heterochromatic segments to euchromatic type. Using this fundamentally out-of-equilibrium model, the authors exhibit how the interplay of the above-mentioned model elements determine the mesoscopic chromatin organization. Their theory predicts that a balance between diffusive and reactive fluxes underly the mesoscopic chromatin organization with characteristic microphase separated morphology determining the length-scale of the heterochromatin domains. They supplement their theoretical study with experimental data obtained from several cell-lines using complementary experimental approaches. Both the theoretical and the experimental results presented in the current manuscript support the conclusions drawn, the research methodologies are up to the mark of the current standards in the research community, the knowledge gained from this research is beneficial for the field. Altogether, this is a good piece of work. Unfortunately, the current manuscript is full of typographical errors, mis-references to the sub-figures and sections, inconsistent symbols. So, I suggest a careful revision of the manuscript before its publication. The authors may pay attention to the points mentioned below.

Response: At the outset we would like to thank the reviewer for their perspective, positive feedback and detailed examination of our manuscript. We have carefully revised our manuscript to remove all errors. Our point-wise responses to the reviewer's comments are detailed below.

Point 1: I did not understand how the authors conceive the mathematical expression for the chromatin-chromatin interactions (the first term on the right hand side, R.H.S.) in Eq. 1? Do they write it phenomenologically or deduce it from something else? They describe it as a term similar to Flory-Huggins terms. However, this comparison is not straight forward to me. It is better to explicitly describe how their term compare with the Flory-Huggins terms, especially, focusing on the enthalpic and entropic contributions. Also, in col. 74, they describe their theoretical model as a polymer physics-based one. Since, I did not understand the first term on R.H.S. of Eq. 1, I am doubtful about this statement.

Response: We would like to clarify that the energy density term used to describe the chromatin-chromatin interactions in our model (Eq 1, Eq S1) is constructed to be similar to the Flory-Huggins descriptions while providing certain advantages. Here, we give a short comparison of the two energy descriptions – Flory-Huggins, and biquadratic construction used by us.

The free energy density W is defined in terms of the local volume fraction of nucleoplasm ϕ_n , euchromatin ϕ_e and heterochromatin ϕ_h . Note that $\phi_e + \phi_h + \phi_n = 1$. The Flory-Huggins type energy description results from the competition of enthalpy and entropic contributions which have the scale of $k_B T / \Omega$. Modifying the

standard Flory-Huggins form for a binary mixture to that suitable for a ternary mixture with two stable phases, the energy of mixing can be written in the following two forms:

Flory Huggins form (De Gennes, 1979):

$$\frac{W\Omega}{k_B T} = \phi_e^2 + \underbrace{\frac{\phi_h}{N} \ln \phi_h + \phi_n \ln(\phi_n)}_{\text{Entropic Contribution}} + \underbrace{\chi \phi_h \phi_n}_{\text{Enthalpic Contribution}} \quad (R1)$$

where, N is the number of monomers in the chromatin polymer. For a particularly large polymer like chromatin, $N \rightarrow \infty$, and the entropic contribution arises from the second term.

Simplified biquadratic form (Chen, 2002; Kim et al., 2017; Ramanarayan & Abinandanan, 2003; Tanaka, 2000):

$$\frac{W\Omega}{k_B T} = \phi_e^2 + \phi_h^2 (\phi_h^{max} - \phi_h)^2 \quad (R2)$$

A graphical comparison of these two equations brings out their similarity as shown in Figure S2, reproduced here as Figure R1. The following section (new Section S1.3, Pages 4-5 of SI) is added to the SI for clarification and comparison of similarity between the choice of the energy descriptions:

Before discussing the role of chemical potentials μ_d and μ_n in spatiotemporal evolution of chromatin organization, we describe the specific form of energetic contributions $W_{CCI}(\phi_d, \phi_n)$ arising from the chromatin-chromatin interactions in Eq S1. The chosen form of energetic contribution must account for a ternary mixture of the three nuclear constituents, with volume fractions - ϕ_e , ϕ_h and ϕ_n . An energy landscape can suitably be constructed to obtain two coexisting phases:

- (i) Water rich euchromatin phase with $\phi_n > \phi_e, \phi_h$, and $\phi_h \rightarrow 0$.
- (ii) Water poor heterochromatin phase with $\phi_h > \phi_e, \phi_n$ and $\phi_h \rightarrow \phi_h^{max}$. Also, ϕ_n is very small.

The Flory Huggins model for the energetics of a mixture of polymer in a solvent can be used as a

Figure R1: A comparison of the double well potential functions as given by Flory-Huggins (Eq R1), and the biquadratic function (Eq R2) used in the manuscript.

for entropic and enthalpic contributions is given as [7, 8],

$$W_{CCI}(\phi_d, \phi_n) = \frac{k_B T}{\Omega} \left[\phi_e^2 + \underbrace{\frac{\phi_h}{N} \ln \phi_h + \phi_n \ln(\phi_n)}_{\text{Entropic Contribution}} + \underbrace{\chi \phi_h \phi_n}_{\text{Enthalpic Contribution}} \right] \quad (S1a)$$

where, N is the degree of polymerization. For a particularly large polymer such as chromatin in the nucleoplasm, $N \rightarrow \infty$ thereby lowering the entropic contribution to the free energy. On a (ϕ_d, ϕ_n) phase space, the contour plot of the energy density $\tilde{W}(\phi_d, \phi_n)$ is shown in Figure S2, left. Also note that, k_B is the Boltzmann constant, T is the temperature and Ω is the volume of individual chromatin particles. The coefficient $k_B T / \Omega$ acts as an energy scaling factor.

The Flory-Huggins form of free energy is appropriate when considering only the enthalpic and entropic contributions to the free energy. *It has been shown that locally driven energy consuming/producing activity, such as motor activity or transcription, or ATP consumption, can drive an ‘activity-induced’ segregation of the chromatin phases [9]. To account for such, more general, mechanisms of phase-separation it becomes more suitable to adopt a generalized simplified biquadratic form of energy landscape. Several polynomial descriptions of double-well energy functions emulating the Flory-Huggins form have been proposed and utilized previously [10-13].* To capture the biphasic (hetero- and euchromatin) phase-separation of a ternary mixture (ϕ_e, ϕ_h, ϕ_n) we adopt a simplified biquadratic double-well energy as,

$$W_{CCI}(\phi_d, \phi_n) = \frac{k_B T}{\Omega} [\phi_e^2 + \phi_h^2 (\phi_h^{max} - \phi_h)^2] \quad (S1b)$$

The contour plot of adopted free energy form (Eq S1b) on a (ϕ_d, ϕ_n) phase space (Figure S2, right), shows a similar location of two energy wells as the Flory-Huggins description. However, the simplification additionally allows (a) easier numerical implementation as the log terms may become undefined due to numerical errors, and (b) a better control on the location of wells, since the well locations are directly defined by the value of the parameter ϕ_h^{max} . Due to these advantages, we use the free energy description given by Eq S1b.

Point 2: The major obstruction in following this manuscript is its mis-references and inconsistent usage of symbols. See for example Eq. 1 and compare it with Eqs. S1 and S1' in the supplementary information (SI). δ is not defined in the main text, ϕ_w should be read as ϕ_n , the tilde over W is missing in Fig. 1b (I guess, if otherwise, please mention it in the caption). There is a repetition of text in col. 150. Check all the texts where subfigures have been referenced (e.g., Fig. 2e should be Fig. 2f on col. 365, and many others). There are two Eq. 6 in the manuscript. I guess, the authors meant to refer the extended methods in SI on col. 144, 212, etc. There is no Sec. 2.5 in methods as being referred to on col. 427. The authors should check the SI as well (e.g., Eq. 13 has been cited on col. 236 of SI, which I could not find). Please note that, I do not intend to provide a complete list of this kind of errors. Better, the authors do a thorough revision of all the text including the figure-captions, check its consistency with the figures, check individual equations, the symbols, and whether the symbols have been defined in the main text (e.g., the rescaled quantity $\tilde{\Gamma}_{me}$ should be defined in the main text).

Response: We have carefully made every effort to eliminate any inconsistencies, mis-references and typographical errors. We apologize that there were so many mistakes in the first place. In addition to correcting the errors pointed out by the reviewer, we also rectified the following:

- We have now consistently kept all the equations in the main text non-dimensional. Earlier some (e.g., Eq 2a, b, 3) were not rescaled (text added to explain the rescaling is highlighted yellow).
- Mis-references highlighted on Pages 16, 18, and 20 of the revised manuscript.
- Updated Figure 6a.
- Rescaling of Eq S10 and S11 is now shown (Page 20, revised SI).
- Label for Eq S19 was missing in the SI (Page 26, revised SI).

Point 3: The phrase ‘diffusion of epigenetic marks’ may not be obvious to all the readers. It has been defined in sec. 2.3-2.4; however, used earlier (see col. 78). The authors may think about better way of presentation on this matter.

Response: We thank the reviewer for pointing out the discrepancy. ‘Diffusion of epigenetic marks’ was mentioned earlier as one of the ingredients of our model. We have rephrased the sentence (Section 1, Page 2):

... drives a separation of hetero- and eu-chromatin phases.

Now, the first mention of the term ‘diffusion of epigenetic marks’ occurs in Section 2.4, where its meaning is explained in more detail now. The following paragraph is added to Section 2.4 (Page 7).

The reaction kinetics however should also incorporate the contribution of chromatin-chromatin interactions, which determine how favorable the euchromatin-heterochromatin interconversion is depending on the epigenetic marks on the other nucleosomes in vicinity. In the SI, we qualitatively (Section S1.6) and theoretically (Section S1.7) describe how neighborhood dependent reaction-kinetics is effectively equivalent to diffusion-like evolution of epigenetic marks, which we incorporate in our model as ‘diffusion of epigenetic marks’ (first term in Eq 2b’). Thus, the reaction kinetics also give rise to an effectively conservative contribution which allows for evolution of epigenetic marks without changing the overall amounts of heterochromatin and euchromatin in the nucleus.

As explained in the added paragraph, we show how the conservative “diffusion of epigenetic marks” can occur as a result of reaction kinetics with the added influence of energetic preference for like-marked nucleosome neighbors. Further explanation of the term ‘diffusion of epigenetic marks’ is also offered in the SI (Section S1.6, Page 10), and Figure S4, which is reproduced here (Figure R2) focusing on the epigenetic mark diffusion.

Figure R2: Schematic depicting the individual roles of the diffusion and reactions kinetics incorporated into the heterochromatin organization model. Reproduction of Figure S4.

In the modified figure, first we discuss the three types of kinetics. The following sections are edited for discussing the diffusion of water and reaction kinetics (Section S1.6, Page 9):

Firstly, we consider the diffusion of nucleoplasm via the conservative kinetics given by Eq S4. If there is no flux of water occurring across the nuclear lamina, the net amount of nucleoplasm in the nucleus $\bar{\phi}_n$ remains a constant. Locally the effect of nucleoplasm movement allows methylated histones to come together resulting in their compaction (Figure S4a). Thus, the conservative diffusion of nucleoplasm allows coarsening of heterochromatin domains keeping the euchromatin phase water rich.

Before discussing the effective diffusion of epigenetic marks via the conservative kinetics (Eq S5), we focus on the epigenetic reaction kinetics described via Eq S6. The role of epigenetic reaction kinetics is non-conservative since it allows an interconversion of euchromatin to heterochromatin via a reaction rate Γ_{me} and heterochromatin to euchromatin via a reaction rate Γ_{ac} (Figure S4b). Note that these interconversions require multiple steps, such as demethylation followed by acetylation, or deacetylation followed by methyltransferase activity (Figure S3). While these reactions do not affect the total amount of water or DNA in the nucleus, they change the individual amounts of heterochromatin and euchromatin. For instance, increase in Γ_{ac} (Γ_{me}) will increase the overall euchromatin (heterochromatin) in the nucleus, without changing the total amount of chromatin.

Next, we discuss what we call the ‘diffusion of epigenetic marks’. The mechanistic evolution is the result of energetics driven neighborhood dependence of reactions, as schematically shown in Figure R2d. The following paragraph is added to the Section S1.6, Pages 10-11:

The presence of epigenetic reactions has an additional energetics influenced component. When considering the reaction kinetics due to epigenetic regulation we have to account for the fact that the nucleosomes that are in close proximity to each other can interact. These interactions might stem from histone-bridging proteins such as HP1 [14, 15], binding complexes such as SAGA [16], or even ionic interactions due to the presence of post-translational modifications. The interactions are captured in our model as chromatin-chromatin interactions and give rise to the energy landscape described in Figure 1b. Because of chromatin-chromatin interaction energetics, heterochromatin-heterochromatin or euchromatin-euchromatin neighbors are more stable than euchromatin-heterochromatin neighbors. Thus, the acetylation and methylation reactions effectively depend on the specific location of the nucleosome within the genome.

The role of energetics can be better understood via the schematic shown in Figure S4d. Say a portion of chromatin polymer at some point in time looks like the initial configuration shown in Figure S4d(i) (in the figure, heterochromatin is shown as red and euchromatin blue). Without loss of generality, let us say that the first step is conversion of heterochromatin into euchromatin (acetylation). There are various heterochromatin sites which can be converted into euchromatin (shown in the grey box), but the configuration shown in Figure S4d (ii) has the highest probability of occurring. This is because it maximizes like-nucleosome neighbors and thus reduces the energy of chromatin-chromatin interactions the most. The next step could either involve converting heterochromatin to euchromatin or vice versa. If a heterochromatin to euchromatin flip occurs, in our meso-scale model its effect is captured by the non-conservative reaction (Eq S6). On the other hand, if a euchromatin to heterochromatin flipping occurs, the possible configurations attainable are shown in the grey box. The configuration with highest probability of occurrence is if all heterochromatic nucleosomes are segregated from the euchromatic ones, as shown in Figure S4d (iii). Thus, over the course of these two steps, we see that an effective conservative rearrangement of heterochromatin and euchromatic nucleosomes can occur which leads to coarsening of the two phases. This conservative evolution effectively is captured in our model as ‘diffusion of epigenetic marks’.

Note that the sequence of events described here gives rise to effective diffusion of epigenetic marks along the chain of polymer – labelled cis-diffusion in Figure S4c. A similar reaction kinetics driven epigenetic diffusion can occur between nucleosomes that are close in the 3D space, although not neighbors on the chromatin polymer chain. Such an event, as shown in Figure S4c, can be called trans-diffusion, and can similarly occur.

To support this qualitative description of events, in the next section we show how the presence of reaction rates dependent on the energetic interactions of nucleosomes can give rise to effective 'diffusion of epigenetic marks'.

Lastly, we also show that the qualitative sequence of events explained schematically in Section S1.6 can be theoretically described by ascribing a neighborhood dependence on the acetylation and methylation reactions. To this end, we have added a new section (Section S1.7, Page 11) to the extended methods. To avoid reproducing the entire section (due to its length) here we give a brief summary -

We consider chromatin to be a 1D polymer of acetylated and methylated nucleosomes corresponding to euchromatin and heterochromatin phases respectively. We consider methylation and acetylation reactions, which respectively convert euchromatin to heterochromatin and vice versa. Lastly, we introduce a neighborhood dependence, such that the rates of acetylation (or equivalently methylation) increase by $\Delta\Gamma$ when one of the neighbors are epigenetically marked similarly to the nucleosome after reaction. The resulting evolution kinetics (Eq S8.3) additively decouples naturally into a reaction-like non-conservative evolution and a diffusion-like conservative evolution – which is equivalent to the 'diffusion of epigenetic marks' in our model.

Point 4: I understand that only the chromatin-chromatin interactions has been considered to draw the contour plot in Fig. 1b. But, I do not understand what the authors meant to say by referring to it in the sentence ending on col. 334.

Response: The reviewer is correct that the energy landscape shows as the contour plot in Figure 1b results from the chromatin-chromatin interactions occurring everywhere in the nucleus. The wells in the landscape dictate the local phase – heterochromatin or euchromatin – of the chromatin organization. However, the path that the chromatin evolution follows towards the wells is dictated by the kinetics of our model, and hence is referenced in the sentence pointed out by the reviewer (now replaced by a short paragraph on Page 11 in the revised manuscript).

As we discuss in Section 3.2 (Eq 3, derived in Section S3), the average amount of heterochromatin or euchromatin in the nucleus, i.e., $\bar{\phi}_h$ or $\bar{\phi}_e$, is determined synergistically by the epigenetic and transcriptional activity. If we start from some initial chromatin organization, reactions will occur over a timescale $1/\Gamma$ to give an overall chromatin organization $(\bar{\phi}_h, \bar{\phi}_e)$ in the nucleus (light blue circle, Figure R1b). This 'composition co-ordinate' gives us an idea as to where we are on the energy landscape, in an average sense (or if the energetics has not yet kicked in) – and is therefore mentioned in col. 334. We have replaced the sentence with the following paragraph for a better explanation (Section 3.2, Page 11):

Thus, the overall mean chromatin composition of the nucleus $(\bar{\phi}_h, \bar{\phi}_e)$ is determined by the reaction kinetics of epigenetic regulation along with transcription. The reaction kinetics alone would drive a homogenous chromatin organization with $(\bar{\phi}_h, \bar{\phi}_e)$. On the (ϕ_a, ϕ_n) phase space we see that the average composition (shown as a light blue circle in Figure 1b) determined by reactions is energetically unfavorable – it does not lie in the energy wells – and hence must evolve in time.

Point 5: I do not understand what the authors meant by 'relative' in Fig. 2g. It has been referred as steady-state radius in Fig. 2f (wrong numbering I believe) on col. 376.

Response: It is the steady state radius just as mentioned in the text. For ease of interpretation, we intended to show this steady state radius value with respect to the maximum possible value. For given $\bar{\Gamma}_{me}$, the maximum possible steady state radius occurs at $\Gamma_a = 0$ (i.e., no chromatin extrusion). So, the value of steady state radius at $\Gamma_a = 0$ is scaled to 1. The value of the steady state radius at any Γ_a is given *relative* to the maximum value. For example, when the relative value is 0.8, the radius at this level of transcription is 80% the size of the radius at no transcription. In the caption to Figure 2g we have now defined the relative sense of the steady state radius:

(g) The evaluation of stable radius (blue) and stable LAD thickness (red) as transcription mediated surface reactions are changed. Here, the relative radius is defined as the steady state radius relative to its value when transcription is zero, i.e., relative radius = $\tilde{R}_d^{SS} / \tilde{R}_d^{SS}|_{\Gamma_a=0}$. The relative LAD thickness is similarly defined.

The following sentence is added to the main text (Section 3.2, Page 12):

The quantitative dependence of the steady state radius on transcriptional kinetics is shown in Figure 2g (blue solid line). Note that the steady state radius shown in Figure 2g is normalized relative to the steady state radius with no transcription.

Point 6: Compare the first the Eq. 6 on col. 659 with the expression written in Fig. 6a.

Response: We are very thankful to the reviewer for pointing out this and other discrepancies. It has been corrected now.

Point 7: I believe some of the preprints cited in the manuscript might have already published.

Response: We thank the reviewer for pointing this out. We found our citations to include the following preprints:

44. Das, R., et al., *How enzymatic activity is involved in chromatin organization*. arXiv preprint arXiv:2112.10460, 2021.
45. Shin, S., et al., *Transcription-induced active forces suppress chromatin motion by inducing a transient disorder-to-order transition*. arXiv preprint arXiv:2205.00353, 2022.

These have now been updated as:

81. Das, R., et al., *How enzymatic activity is involved in chromatin organization*. *Elife*, 2022. **11**: p. e79901.
82. Shin, S., et al., *Transcription-induced active forces suppress chromatin motion by inducing a transient disorder-to-order transition*. *Biophysical Journal*, 2023. **122**(3): p. 19a.

REVIEWER #2

General Comments 1,2: In contrast to present polymer theories of nuclear genome structure and their relation to transcriptional activity, this appears to be the first large-scale attempt to quantitatively predict essential structural features of the condensation/decondensation of nuclear chromatin caused synergistically by epigenetic markers and transcriptional complexes (corresponding to some qualitative models previously discussed in the Literature) within the framework of a unified differential equation of the chemical potential of the entire nuclear genome. This appears also to be the first time that the role of epigenetic structure related elements is postulated to be equivalent to the biochemistry of transcription: Accordingly, genome structure is no longer regarded as a "downstream" phenomenon of transcription processes, but emerges to be also essential for gene regulation. Its results are in accordance with a wealth of experimental results obtained by both electron and super-resolution microscopy.

2) For the readership of Nature Communications (mostly with a biological/biochemical background) it should be helpful to add a short section indicating this highly innovative perspectives, and also to refer to previous qualitative models referred to in the literature.

Response: We thank the reviewer for their encouraging comments, and feedback that has helped us refine the model and its presentation for the better. As advised by the reviewer, to better contextualize our PDE based continuum model with the polymer theories previously presented in the literature, we have added the following discussion in Section 4, Page 23:

Significant inroads into mechanistic modeling of chromatin organization as physically and functionally distinct states with finer architectural sub-features such as topologically associated domains (TADs) and

chromatin loops have been made from a polymer physics perspective. Such models were developed with different levels of fine-graining to capture biophysics of chromatin organization at different length-scales spanning single or multiple nucleosomes [52-54], multiple nucleosome clutches [55-60], single and multiple TADs with salient sub-TAD features [61-64], single and multiple chromosomes [39, 40, 63, 65-68] and the whole genome [69-72]. Depending on the focus on the chromatin functionalities or structure being simulated, any of these models can be adopted. For instance, first-principles thermodynamics driven approach may capture chromatin as a copolymer with two states, whereas a data-driven approach trained on conformation capture (e.g., Hi-C) or sequencing (e.g., CHIP-seq) data may incorporate over 50 states spanning the entire genome.

The experimentally observed role of RNAPII-mediated transcription in DNA supercoiling and subsequent loop extrusion [2, 12, 19, 20, 51, 73-76] has also been studied using molecular dynamics simulations and polymer physics-based models at nanoscale [20, 77-82]. However, quantitative predictions of sizes of heterochromatin domains which organize at a nucleus-wide meso-scale level are beyond the purview of such models. Furthermore, to the best of our knowledge, polymer models lack the far from equilibrium kinetic considerations of active epigenetic regulation, chromatin extrusion and diffusion kinetics, which we find are intricately involved in the spatiotemporal regulation of heterochromatin domain sizes. The current study, incorporating coarse-grained continuum model of chromatin organization at a mesoscale, presents the following novel advantages over previous polymer-based models:

- a. Nucleus-wide characteristic size distribution of heterochromatic domains.
- b. LADs of finite thickness co-existing with interior heterochromatin domains and their dynamic size-regulation.
- c. The kinetic interplay of diffusion, epigenetic reactions and transcription in regulation of meso-scale organization.

General Comment 3: The Theoretical Physics effort used for this purpose is impressive; it is understandable that the detailed theoretical description given in the Supplementary Material will mostly be reserved to specialists in the field. In the main text, however, it might be helpful for readers with an experimental background to explain in a more „basic language“ the general highlights of the theory. In the Supplementary Information, a list of the mathematical symbols (well known to Physicists but perhaps not all familiar to others) might be added.

Response: We agree that a list of mathematical symbols in the SI will be very useful in better presenting our model. The following table is added in the Extended Methods section of the SI (Section S1.4, Page 7) of the SI:

	Symbol	Physical Interpretation	SI Unit	Remarks
	x	Spatial variable, two-dimensional vector	m	
	t	Time variable	s	
Volume Fractions	ϕ_h	Volume fraction of heterochromatin	1	Defined in Section S1.1
	ϕ_e	Volume fraction of euchromatin	1	
	ϕ_n	Volume fraction of nucleoplasm	1	
	ϕ_d	$= \phi_h - \phi_e$, difference between heterochromatin and euchromatin volume fractions, order parameter	1	
Energetic contributions	W	Total free energy density of chromatin organization	J/m^3	Defined via Eq S1
	W_{CCI}	Free energy density due to chromatin-chromatin interactions	J/m^3	Defined via Eq S1.2
	V_L	Strength of chromatin-lamina interactions	J/m^3	Defined in Section S1.2
	d	Distance of a point from the nuclear periphery	m	
	d_0	Length-scale of chromatin-lamina interactions – the distance over which effect of chromatin-anchoring proteins vanishes	m	
	η	Energy penalty on formation of sharp interface	J/m	
Diffusion	$\mu_{n,d}$	Chemical potential = $\delta W / \delta \phi_{n,d}$ – driving force for energy-reducing passive diffusion kinetics	J/m^3	Defined via Eq S3

	$J^{n,d}$	Nominal volumetric flux of mobile species – defined as volume of species flowing per unit local area per unit time	m^3/m^2s	Defined in Section S1.4
	$M_{n,d}$	Mobility of nucleoplasm or epigenetic marks in the nucleus	$\frac{m^2/s}{J/m^3}$	
	$D_{n,d}$	Diffusivity of nucleoplasm of epigenetic marks in the nucleus	m^2/s	
Reaction kinetics	Γ_{me}	Reaction rate of methylation, defined as deacetylation + methyltransferase activity	s^{-1}	Defined in Section S1.4
	Γ_{ac}	Reaction rate of acetylation, defined as demethylation + acetyltransferase activity	s^{-1}	
	Γ_a	Rate of chromatin extrusion driven by transcription-mediated supercoiling	s^{-1}	

In the main text, as suggested by the reviewer, we have tried to simplify the presentation of the theory. In the section 2.1 of the main manuscript (Pages 3-5), we have rewritten the introduction to the model from a biophysical perspective:

The model ingredients are depicted schematically in Figure 1a. While on one hand entropic contributions push chromatin towards a homogenous organization, enthalpy arising from nucleosome-bridging via HP1 proteins [22], via ionic interactions within chromatin phases [23, 24] or local activity [25] oppose it. The entropic-enthalpic competition comprising the chromatin-chromatin interactions drives the phase separation of chromatin domains. The emergent formation of domains occurring thermodynamically in our model is similar to the chromatin domains qualitatively postulated based on super-resolution imaging [5, 26, 27]. Near the nuclear periphery, heterochromatin can be further anchored to the nuclear lamina via anchoring proteins such as LAP2 β [28-30]. This drives the formation of peripheral LADs. The interior and peripheral heterochromatin domain formation occurs spatiotemporally via free energy lowering diffusion of nucleoplasm, and diffusion-like evolution of acetylation or methylation marks on the histones. We incorporate the active interconversion between the eu- and heterochromatic phases in two ways: histone methylation or acetylation reactions can change the epigenetic distribution or transcription mediated supercoiling-driven chromatin loop extrusion.

As shown in Figure 1a, supercoiling-driven DNA loop extrusion, can occur broadly in two regions where RNAPII is present [26, 27, 31, 32]: within the euchromatin domains (red dashed circle in Figure 1a) or at the interface of heterochromatin and euchromatin phases (black circle in Figure 1a). Since the chromatin extrusion in the euchromatin phase maintains its transcriptionally active status and does not lead to any significant mesoscale changes in the epigenetic distribution, we focus on the domain interface. The chromatin extrusion at the interface is instrumental in the regulation of size of heterochromatic domains at the periphery to form euchromatin.

General Comment 4: A peculiarity of the Chemical Potential theory of nuclear chromatin is that so far no sequence-specific distinctions are made. In order to still come up with something useful (decondensation of the surface of domains upon transcription, condensation without transcription), as a starting point the authors assume a qualitative model very similar to Miron et al. (2020) and other papers in the literature as an initial condition, however without reference to them. To convince the biologically/biochemically oriented reader, it should be helpful to explain a little bit more the assumptions made, and to quote some of the earlier, qualitative models some of them fitting well to the results of this MS.

Response: We thank the reviewer for pointing us towards these publications. We do agree that the compacted heterochromatin domains we refer to correspond very closely to ~200 nm chromatin domains reported by other super-resolution imaging techniques – 3D-structured illumination microscopy (Miron et al., 2020), electron microscopy (Markaki et al., 2010; Miron et al., 2020), and photoactivated localization microscopy (Nozaki et al., 2017). Our model also predicts a chromatin organization very similar to the qualitative models based on these imaging techniques (Miron et al., 2020; Nozaki et al., 2017).

We have explained our model in simpler terms in response to **General Comment 3** above. As suggested by the reviewer we have added the following sentence when introducing our model (Section 2.1, Page 3):

The emergent formation of domains occurring thermodynamically in our model is similar to the chromatin domains qualitatively postulated based on super-resolution imaging [5, 26, 27].

General Comment 5: The impressive differential equation for the chemical potential of the nucleus in its present form yielded usable solutions only by additionally applying a set of experimentally determined parameters that were, moreover, "dimensionless": This means that, for example, only relative densities of DNA in the domains (e.g., radial relative distribution) could be calculated; thus, absolute densities (Mbp/ μm^3) could not be given. However, according to polymer based numerical simulations (e.g. Maeshima et al., 2015), as well as to qualitative models referred to in the literature (so far not cited by the authors), absolute densities of chromatin domains may control the accessibility of macromolecular complexes to target sequences essential for the initiation of transcription.

Response: We agree with this and other comments by the reviewer regarding the 'dimensionless' considerations of our model. The only reason this was done was to specifically consider only the parameters which quantitatively affect the chromatin organization – more specifically the size-determination of the heterochromatin domains. While the other parameters are important, they emerge as scaling parameters in our equations, i.e., these parameters merely rescale our results. As suggested by the reviewer here and in future specific comments, for the sake of completeness, in the SI we have now added a list of the parameter values (Table S3, Section S9, Page 29 of the SI) which can be used to translate our dimensionless model into physical parameters. The table is reproduced below:

	Parameter	Meaning	Order of magnitude	Reference	Remarks
Reaction timescale	Γ_{ac}	Rate of histone acetylation reaction	$\sim 10^{-2} s^{-1}$	[27]	Calculated as reciprocal of time-scale of reaction
	Γ_{me}	Rate of histone methylation reaction	$\sim 10^{-3} s^{-1}$	[28]	
Length-scale	D	Diffusivity of nucleosomes	$\sim 10^{-3} \mu\text{m}^2/s$	[29]	Diffusivity of nucleosomes calculated as slope of MSD curve
	ℓ_{RD}	Reaction-diffusion length scale	$\sim 300 \text{ nm}$	Calculated as $\ell_{RD} \sim \sqrt{D/\Gamma_{ac}}$	
	R_d^{SS}	Characteristic steady state size of heterochromatin domains	$\sim 100 \text{ nm}$ [29-34] ¹	Calculated as $\sim \sqrt{\frac{3D}{\Gamma_{ac}} \frac{\Gamma_{me}}{\Gamma_{me} + \Gamma_{ac}}}$ (Eq S16)	
Diffusion timescale		Time for diffusion across inter-domain space	$\sim 5 \text{ min}$		Calculated as $\sim (2\ell_{RD})^2/D$
Energy scale	$\frac{k_B T}{\Omega}$	Energy scale of chromatin-chromatin interactions	$\sim 10^{-24} \text{ J}/\text{nm}^3$		Calculated. Assuming nucleosomes interact with each other over a length scale of 10 nm
	ℓ_{int}	Width of the smooth boundary of heterochromatic domains	$\sim 50 \text{ nm}$		Observed from ChromSTEM imaging (Figure 2c)
	η	Penalty associated with formation of interfaces	$\sim 10^{-21} \text{ J}/\text{nm}$		Calculated as $\ell_{int} = \sqrt{\frac{\eta\Omega}{k_B T}}$, (Section S1.5)

The reviewer is also right in pointing out that the specific density of chromatin within both heterochromatin and euchromatin is not measured. Rather, we use the volume fraction of chromatin which is non-dimensional. The exact choice of the absolute density of chromatin in the compacted phase will serve to relocate the well corresponding to the heterochromatin phase (e.g. in Figure 1b). However, this does not change the underlying physics of the model, nor does it affect the predictions made. This choice of the extent of chromatin compaction in the heterochromatin phase would be better predicted by molecular dynamics-based polymer models at a resolution of nucleosomes. The predicted compaction density would serve as an input for our model to more accurately predict the meso-scale organization.

Exemplifying this, we now report three simulations with three different choices of compaction of chromatin in the heterochromatin phase (Figure S15, reproduced here as Figure R3). We adjust this choice by changing the overall water content of the nucleus, such that the heterochromatin phase becomes less compacted and contains a higher volume fraction of water. The following section is added to the SI (Section S10, Page 30 of the SI) explaining our predictions:

Our model predicts that chromatin exists in two phases which are not only transcriptionally distinct but are also differentially constituted. For instance, the euchromatin phase primarily consists of acetylated chromatin with a volume fraction $\phi_e^{EC} \sim 0.2 - 0.3$ and is water rich with volume fraction of nucleoplasm $\phi_n^{EC} \sim 0.7 - 0.8$. However, the euchromatin phase has no methylated heterochromatin content, i.e., $\phi_n^{EC} \sim 0$. On the other hand, the heterochromatin phase is rich in heterochromatic content $\phi_h^{HC} = \phi_h^{max}$, and has a very little water content, with $\phi_n^{HC} \sim 0.04$.

Note that the heterochromatin phase is only water-poor, and not water-free, i.e., $\phi_n^{HC} \neq 0$. Due to being water-poor the heterochromatin phase is highly compacted, but not so compacted as to exclude water. This distinction becomes important because even the highly compacted heterochromatin phase includes multiple chromatin associated proteins such as HP1 [15] and certain histone methyltransferases [31].

The extent of chromatin compaction in the heterochromatin phase i.e., the maximum heterochromatin volume fraction $\phi_h^{HC} = \phi_h^{max}$ is chosen suitably to allow the presence of some euchromatin and nucleoplasm within the heterochromatic phase. We observe a water content of $\phi_n^{HC} \sim 0.04$. However, the exact values of these parameters, or the exact density (in $\text{Mbp}/\mu\text{m}^3$) of chromatin within the heterochromatin phase, could be obtained by molecular dynamic simulations at a resolution of nucleosomes, calibrated to single

Figure R3: Chromatin organization predicted numerically as the overall water content in the nucleus varies. (a) As the nucleus becomes water rich, the heterochromatin phase becomes less compacted and contains a higher volume fraction of water. (b) For the case of $\bar{\phi}_n = 0.6$, as the rate of chromatin extrusion Γ_α increases, the heterochromatin domains become smaller.

nucleosome live-cell super-resolution imaging (such as [29, 32]). A combined computational and imaging framework involving polymer-based modeling at nucleosome level and super-resolution imaging, called Modeling immuno-OligoSTORM (MiOS) has previously been developed by Neguembor, et al. [35], which could potentially be used for observing such fine-scale features of DNA organization.

As an example, we modify the overall water content of the nucleus ($\bar{\phi}_n$) to modulate the extent of chromatin compaction and water content within the heterochromatin phase (ϕ_n^{HC}). The extent of chromatin compaction can also be

modified in other ways, such as varying ϕ_h^{max} , but here we present one example to show that our model predictions are agnostic to the exact choice of these parameters. The chromatin organization predicted by the model is shown in Figure S15. Note that in Figure S15a the distribution of water in the nucleus is shown (as opposed to that of heterochromatin). However, the red region (with low water content) is still heterochromatin, and blue region (with high water content) is euchromatin. Further, we have annotated the exact value of local volume fraction of nucleoplasm within a heterochromatin domain, i.e., ϕ_n^{HC} . It can be seen that as the nucleus becomes water rich, the heterochromatin phase compaction varies. This is because changing the water content in the nucleus also changes the location of the energy well on the energy landscape.

For a specific case of $\bar{\phi}_n = 0.6$ (different from the value we have chosen in the manuscript), we see that as the transcription increases the heterochromatin domain size scale in the interior and along the periphery reduces – in qualitative agreement with predictions reported in the main manuscript (Figure S15b). Thus, while choosing the exact level of chromatin compaction does quantitatively improve our prediction accuracy, it does not modify our model results, nor does it change the underlying physics of the model.

General Comment 6: A first very rough estimate of the absolute densities relevant for the theory presented might already be derived from the initial conditions assumed by the authors that the heterochromatic domains have such a high density that even water cannot diffuse. From this (and also from the schematic images shown) it is clear that the assumed density of the compact domains will be at least 200 Mbp/ μm^3 , i.e. such domains should be inaccessible for macromolecules. Hence, transcription must take place at the surface of the domains, as the authors also explicitly postulate.

Response: This is a very interesting point raised by the reviewer. While our schematic representations do indeed show a very tight packing of histones within the heterochromatin domain, we do not want to specifically claim, or intend to give an impression, that they are so dense that ‘even water cannot diffuse’ in. This is because condensed chromatin can still be accessible to proteins, albeit small, responsible for several functions such as heterochromatin cross-linking by HP1 α (Sanulli et al., 2019) or DNA repair (review by Feng and Michaels (2015)).

As we discuss in our response to **General Comment 5** by the reviewer, and in a new section (Section S10, Page 30) added to the SI, our model is agnostic to the actual values of chromatin density within the compacted domains. It is completely allowable that there may be some water in the compacted domains, though we expect this value to be quite small. Indeed, the presence of small epigenetic modifiers such as KMTs within the heterochromatin domains (Miron et al., 2020), requires that some water be present in the heterochromatin phase. However, as the reviewer rightly says, larger macromolecules such as those involved in the transcriptional machinery e.g. RNAPII may not be able to traverse through the heterochromatin phase (Maeshima et al., 2015; Miron et al., 2020) and thus be restricted to euchromatin, as we postulate via Eq S7.

General Comment 7: Recent published reports of experimental measurements of the absolute DNA density in compact chromatin domains obtained by single molecule localization microscopy (SMLM) revealed a range of absolute densities sufficient to allow an accessibility controlled fine regulation of transcription.

Altogether, the basic theoretical approach presented offers novel and fascinating perspectives for a substantially improved understanding of functional nuclear genome structure, based on a rigid application of the fundamental laws of Physics and Chemistry.

From the side of "theoretical nuclear genome physics", there are thus two basic approaches that can be further developed:

a) Numerical models based on polymer chains (developed e.g. by the Nicodemi, Safran, de Rosas, Everaers labs); and

b) Differential equation systems built on the thermodynamics of chemical potentials, as presented in this MS.

Within their specific limits, both quantitative modelling approaches will be extremely valuable to provide a theoretical frame to experimental nuclear genome structure research. Another, most important perspective will be the wealth of experimentally testable predictions (an example for this documented also in this ms). This will not only stimulate the application of present methodology, from Hi-C to electron microscopy and super-resolution, but also the development of new experimental approaches.

Eventually, this may even contribute to the development of epigenetic pharmaceuticals, e.g. by specifically changing transcription related chromatin compaction. It might be helpful for the reader to add a short section on the comparison of the Chemical Potential model to polymer based theories.

Response: We thank the reviewer for this really helpful perspective. In Section 4 (Page 23), we have now presented a starker comparison between these two modeling philosophies, and how both possess advantages when it comes to understanding chromatin organization at different scales. As we discuss in our manuscript, we feel that the novelty of our work arises from two aspects which have not been previously addressed via either of the models – (a) far from equilibrium dynamics of chromatin organization due to epigenetic reactions and transcriptional activity, and (b) dynamic size scaling of heterochromatin domains along with its underlying biophysical principles.

To incorporate the perspective offered by the reviewer, and to provide the reader a context for our modeling, we have now added a discussion on the same in Section 4, Page 23. The added text was reproduced in response to **General Comment 1** by Reviewer 2.

Specific Comment 1,2: „Transcription-driven loop extrusion“: This is a favorite topic of biochemically oriented papers. However, models shown on this subject in the literature are extremely “qualitative”. The more quantitative nanostructural data are made available, the more realistic such models should become; eventually it should be possible to understand what the biochemically inspired word „loop extrusion“ really means in terms of space and time.

“Loss of transcription results in increased chromatin compaction and larger heterochromatin domain sizes“: This result would fit extremely well to earlier qualitative models described in the literature which postulate that the probability of transcription and chromatin compaction is intimately correlated.

Response: We agree with the reviewer on the significance of quantitative studies on chromatin organization. With multiple new technologies available from a sequencing, super-resolution imaging and modeling perspectives, a three-pronged inquiry into the role of macromolecular biochemistry and fundamental biophysics in determining the chromatin organization is a must. We firmly believe that our model has the potential to lay the foundation for such an exploration. Polymer-based modeling techniques coupled with meso-scale application advantages of a first principle thermodynamics based PDE model can be a powerful tool on the mathematical modeling front.

In response to reviewer’s views, we have added more references to qualitative models based on experimental observations. The following citations are added:

5. Nozaki, T., et al., *Dynamic organization of chromatin domains revealed by super-resolution live-cell imaging*. Molecular cell, 2017. **67**(2): p. 282-293. e7.
26. Markaki, Y., et al. Functional nuclear organization of transcription and DNA replication a topographical marriage between chromatin domains and the interchromatin compartment. in Cold Spring Harbor symposia on quantitative biology. 2010. Cold Spring Harbor Laboratory Press.
27. Miron, E., et al., Chromatin arranges in chains of mesoscale domains with nanoscale functional topography independent of cohesin. Science advances, 2020. 6(39): p. eaba8811.

Specific Comment 3: „The existence of a multiscale chromatin organization is well-established [1, 2]“: Both initial references describe well the present state of biochemical analysis (Hi-C, Cohesin loops etc.) but do not provide an adequate description of the essential contributions of microscopy. Some additional references might be helpful.

Response: As suggested by the reviewer, we have added more references to take into consideration the very important super-resolution imaging/microscopy contributions. The statement is modified to account for imaging experiments (page 2):

The existence of a multiscale chromatin organization has been observed not only from sequencing and contact-mapping techniques [1, 2], but also super-resolution imaging [3-8].

Following citations are added:

3. Bintu, B., et al., *Super-resolution chromatin tracing reveals domains and cooperative interactions in single cells*. Science, 2018. **362**(6413): p. eaau1783.
4. Neguembor, M.V., et al., *MiOS, an integrated imaging and computational strategy to model gene folding with nucleosome resolution*. Nature Structural & Molecular Biology, 2022. **29**(10): p. 1011-1023.
5. Nozaki, T., et al., *Dynamic organization of chromatin domains revealed by super-resolution live-cell imaging*. Molecular cell, 2017. **67**(2): p. 282-293. e7.
6. Ou, H.D., et al., *ChromEMT: Visualizing 3D chromatin structure and compaction in interphase and mitotic cells*. Science, 2017. **357**(6349): p. eaag0025.
7. Ricci, M.A., et al., *Chromatin fibers are formed by heterogeneous groups of nucleosomes in vivo*. Cell, 2015. **160**(6): p. 1145-1158.
8. Wang, S., et al., *Spatial organization of chromatin domains and compartments in single chromosomes*. Biophysical Journal, 2017. **112**(3): p. 217a.

Specific Comment 4: Altogether, the mathematical theory presented looks very interesting as a first attempt to describe the space-time structure of the nuclear genome in terms of a general theory based on Thermodynamics. To allow some meaningful testable conclusions, some radical “boundary” assumptions had to be made. It may be useful to provide a little more technical information about the calculations made.

Response: We had previously set aside most of the technical information in the SI, but we understand how briefly including it in the main manuscript can help make the story more self-sufficient and useful. For instance, the “domain-boundary” localized reaction captured by transcription-driven chromatin extrusion was expanded on fully in the SI, but now we have explained the motivation and actual form of this equation in the main-manuscript as (Section 2.4, Page 7):

The last term in Eq 2b' accounts for the supercoiling-driven chromatin extrusion kinetics and the chromatin state changes resulting from it. Being transcription mediated, the kinetic rate of supercoiling-driven extrusion $\Gamma_a(x)$ must be spatially dependent on local availability of RNAPII, which is prominently present at the boundaries of the compacted heterochromatin phase [26,27,31]. Although supercoiling-driven loop extrusion may also occur within the euchromatin phase, it does not contribute to interconversion of chromatin phases as euchromatin is already transcriptionally active. In contrast, at the interface of heterochromatin and euchromatin, supercoiling-driven loop extrusion can result in activation of otherwise inactive genes. Considering this spatial localization to the heterochromatin domain boundaries, we rewrite Eq 2b' as,

$$\frac{\partial \phi_d}{\partial t} = \underbrace{M_d \nabla^2 \mu_d}_{\text{diffusion}} + 2 \left(\underbrace{\Gamma_{me} \phi_e - \Gamma_{ac} \phi_h}_{\text{epigenetic regulation}} - \underbrace{\Gamma_a e^{-\left(\frac{\phi_h - \phi_h^{\max}}{2 \Delta \phi}\right)^2}}_{\text{active chromatin extrusion}} \right) \phi_h \quad (2b)$$

Note that $\phi_h^{\max}/2$ is the volume fraction of heterochromatin at the domain boundary. A deviation of $\Delta\phi$ from this value defines the width of the domain boundary, and the supercoiling-driven loop extrusion is restricted to a narrow width surrounding the heterochromatin phase.

As another example, boundary and initial conditions used for solving the PDEs were previously relegated to the SI (Section S8). We now briefly mention them in the main manuscript Section 2.4 (Page 8).

As a boundary condition we ensure no exchange of water and chromatin between the nucleus and the surroundings. This condition can be suitably adjusted to allow flow of water from or into the nucleus.

Lastly, we define the volume fraction more clearly in the SI. This is further discussed in response to **Specific Comment 6** by Reviewer 2.

Specific Comment 5: An alternative basic approach to the Chemical Potential theory would be a more generalized polymeric theory. For example, using Monte Carlo approximations, relations between absolute chromatin density and accessibility (i.e. between Euchromatin and Heterochromatin) may be predicted; assuming interaction potentials between a polymeric chromatin chain and lamins, structural predictions of the distribution of heterochromatin and euchromatin in the nucleus may be calculated (e.g. Bajpai et al. 2021; Amiad-Pavlov et al., 2021): one might add to such schemes also diffusion and binding characteristics of epigenetic factors etc. In the discussion, some references to such alternative theoretical approaches might be useful. Which model is adapted to solve specific problems of nuclear organization has to be explored in future studies. Probably both will have their importance. Concerning this ms, it is suggested to add a section to discuss the relationship of the Chemical Potential model chosen to Polymer Theory approaches.

Response: We agree with this suggestion by the reviewer. There have been many important polymer-based models that have been developed and reported in the literature, each employing a variety of techniques for model development and validation. We discussed this in response to **General Comment 1** by Reviewer 2. From a different perspective, these can be thought to belong in two broad categories:

- (i) Finding the best 3D equilibrium conformation from the Hi-C map of ChIPseq data (Di Pierro et al., 2017; Paulsen et al., 2017; Qi & Zhang, 2019; Shi et al., 2018), and
- (ii) Considering the out of equilibrium dynamics (e.g. via Kawasaki non-conservative algorithm) to study the evolution of the chromatin polymer (Katava et al., 2022; Michieletto et al., 2016).

Our model lies much towards the non-equilibrium end. To the best of our knowledge, none of the models consider how chromatin re-organizes in response to the interplay of epigenetic and transcription modulations which is the focus of this paper. While nucleoplasm diffusion via Brownian dynamics is a common feature in these models, the epigenetic diffusion and reactions – as we have described by in Sections 2.3-4 and S1.6 – are unique to our model and have not been previously employed from a polymer perspective. Thus, the size regulation discussed by us is a novel finding.

We have also (not a part of this manuscript) created a non-equilibrium polymer model at the scale of one chromosome, with a resolution of 10 kbp (Vinayak et al., 2023a; Vinayak et al., 2023b). This work is more data intense and allows for predictions related to specific genes, however the domain size predictions from the polymer model corroborate the those discussed in the continuum model presented here. We are close to submitting this work.

Specific Comment 6: Maintext & Supplementary Material: How is the “local” volume fraction defined ? Is it the ratio of e.g. [nucleoplasm volume]/[nuclear volume] ? Or is it ratio of [nucleoplasm volume]/[domain volume] ? Generally, the legibility of the ms might be improved by a little more „redundancy“ in the explanation of the parameters shown (if due to length limits this is not possible in the main text, it might be done in the Supplementary Information).

Response: The definition of local volume fraction is similar to that of local concentration but is instead dimensionless. Consider a small observation window – say a finite sized small cube in the 3-dimensional

nucleus. The local concentration of heterochromatin is defined as the amount of heterochromatin divided by the volume of this cube. In a similar way, the local volume fraction ϕ would be the volume of heterochromatin divided by the volume of the cube. Say the local observation window has a volume V . Let us say that the volume of heterochromatin within this finite region is V_h , that of euchromatin is V_e and of nucleoplasm is V_n . Then we can say that $V_h + V_e + V_n = V$, assuming that any other constituent is a part of nucleoplasm. Dividing both sides by V , we get $\phi_h + \phi_e + \phi_n = 1$. This equation defines the relationship between the volume fractions of the nuclear constituents at any point in the continuum.

We have now more clearly defined the term ‘volume fraction’ by adding the following paragraph to the SI Section S1.1 (Page 2):

To investigate the organization of chromatin in the nucleus, we develop a mathematical model for the phase separation of heterochromatin and euchromatin considering chromatin-chromatin interactions, chromatin-lamina interactions, epigenetic regulation of chromatin via histone acetylation or methylation, and the role of transcriptional regulators. We consider three nuclear constituents – nucleoplasm and chromatin in either heterochromatin or euchromatin phases.

At any point x in the nucleus, at a time t , consider an infinitesimal observation window of volume $V(x, t)$. Let the volume of nucleoplasm, euchromatin and heterochromatin within this observation window be $V_n(x, t)$, $V_e(x, t)$ and $V_h(x, t)$. Note that $V = V_n + V_e + V_h$. Thus,

$$\frac{V_n}{V} + \frac{V_e}{V} + \frac{V_h}{V} = 1$$

The ratio of volume of each component to the total volume of the infinitesimal observation window is defined as the volume fraction of the component $\phi_i(x, t)$, for $i = n, e, h$, and determines the content of nuclear constituents – nucleoplasm, euchromatin and heterochromatin at each point x and time t in the nucleus. Thus,

$$\phi_n + \phi_e + \phi_h = 1$$

Thus, the physical state of the nucleus at any point can be defined by volume fractions of any two nuclear constituents, with the third constrained via the above equation.

Further, we have ensured that each variable is defined properly at its introduction. Lastly, we have provided a table (Table S1, Page 7) of model variables along with their physical definitions. This table was reproduced in our response to **General Comment 3** by Reviewer 2.

Specific Comment 7: To use the thermodynamic theory of free energy density is very elegant. The consequence of such a general theory using differential operators is that in principle it assumes a very large number of molecules and does not consider the special conditions of molecular configurations (e.g. number of base pairs; number of H2B molecules, proteins, loops etc.). Hence it is expected that the theory by itself will be able to describe general relationships (e.g. relative densities) but not specific molecular conditions. This might be obtained by experimental calibration, e.g. to translate relative radial intensity distributions of compact domains into absolute ones (in terms of Mbp/ μm^3).

Response: We agree with the reviewer. While the continuum model of phase separation works very well at the meso-scale, it glosses over the fine-grained discrete macromolecular spatial refinement which may be offered by, say, the fine-resolution molecular dynamics simulations. These details are however indirectly included via the energy landscape constructed to represent the chromatin-chromatin interaction, or the maximum chromatin compaction volume fraction ϕ_h^{max} .

We note that the variable volume fraction ϕ_i in our model is linearly proportional to the absolute values of chromatin density. However absolute values (in terms of Mbp/ μm^3) of such parameters would be better obtained via experimental observations and molecular dynamic simulations (e.g. single particle tracking in (Gómez-García et al., 2021; Nozaki et al., 2017)) and could then be used as input parameters (such as

ϕ_h^{max}) in our model. A discussion on this was added to the SI (Section S10, Page 30) in response to **General Comment 5** by Reviewer 2.

Specific Comment 8: The assumption of a water devoid heterochromatin phase would physically be possible only at extremely high absolute densities (estimated to be in the range above 200 Mbp/ μm^3). This would imply:

a) that only a small part of the nuclear volume is in this state: Assuming e.g. a density of 200 Mbp/ μm^3 and a total cellular DNA content of 6,000 Mbp, the heterochromatin volume $V = [\text{DNA content}]/\text{density}$ would be $6,000/200 \mu\text{m}^3 = 30 \mu\text{m}^3$, i.e. only a few percent of a typical nuclear volume;

b) that target sequences in the interior of such extremely compact structures would definitely be excluded from any accessibility of transcription factor complexes. In contrast to these considerations, the schematic figures shown suggest a lot of space even between the chromatin fibers in the interior of heterochromatic domains. This appears to be correct if one assumes that the DNA chain is shown without the nucleosomes; but this may not provide an idea of the real density relations which are of utmost importance for accessibility. Hence it should be helpful to state in the legend whether DNA or chromatin fibers (nucleosome chains) are intended.

Response: This is a very insightful analysis by the reviewer. We would again like to clarify that we do not presume the heterochromatin phase to be completely water devoid, although we do consider it to be a water-poor phase. On the other hand, euchromatin has a lot higher water content due to its loose compaction. From our simulations we observe a nearly 4% water content (i.e., $\phi_n^{HC} \approx 0.04$) in the heterochromatin phase. As discussed in response to **General Comment 5** by Reviewer 2 (new Section S10 of the SI), the exact value of chromatin compaction relocates the wells without changing the underlying physics or qualitative predictions of the model.

That said, the reviewer's analysis still holds. Here are our thoughts:

- (a) We agree that only a small part of the nuclear volume exists in a compacted state. It has been previously reported that chromatin occupies nearly 40% of nucleus by volume (Rouquette et al., 2009). Of that, most of the human genome has been shown to be euchromatic (Consortium, 2004).
- (b) We do concur on the thought that relatively larger macromolecules, especially those constituting the transcriptional machinery, would not be readily accessible within the chromatin domains – as has been shown via several imaging techniques (Li et al., 2021; Maeshima et al., 2015; Markaki et al., 2010; Miron et al., 2020).

Via the schematics, we do not intend to show a lot of empty space within the heterochromatin domains. In our figures, whenever nucleosome is shown, the packing of the heterochromatin is kept very tight (e.g. Figure 1a,c, and 6a). However, in a few figures (e.g., Figure 5a, and 7), we wanted to keep the focus on chromatin fibers to better show loop formation. While nucleosomes are not shown, they are never assumed to be absent. If there are no nucleosomes the DNA cannot be packed because of high repulsive charges. For clarification, as advised by the reviewer, in the legends to these figures we have now mentioned that:

Nucleosomes, despite being present, are not represented in these schematics to better display the chromatin loops.

Specific Comment 9: „Interactions between the chromatin and the lamina“: A reference to the polymeric chromatin-lamin interaction model of Bajpal et al. (2021) and Amiad-Pavlov et al. (2021) may be helpful. An advantage of these and related polymer theories to predict the consequences of chromatin-lamin interactions is that their approach is based on a discrete polymer chain with a large but finite number of chromatin particles (“beads”), each “bead” representing a definite small amount of DNA which may be given in absolute units (No. of bp, No. of nucleosomes). On the other side, so far such polymeric models regarded only chromatin-chromatin and chromatin lamin interactions and hence did not include explicitly epigenetic

diffusing factors, as in the present ms. In the future, both approaches will probably be valid, depending on the special problems to be solved.

Response: We agree that these references are very helpful in giving a perspective to the reader on how chromatin-lamina interactions are interpreted from a polymer perspective. We have now added these references to our manuscript when motivating the chromatin-lamina interactions (Section 2.2, Page 6):

Analogous discrete descriptions of chromatin-lamina interactions, via formation of strong bonds when the chromatin is within a characteristic distance from the lamina have been previously implemented [39, 40] in polymer models of chromatin, although without the epigenetic or transcriptional kinetics.

Specific Comment 10: How are the small heterochromatic domains in the nuclear interior obtained? By assuming a certain amount of lamina also in the nuclear interior?

Response: While research has indeed suggested that lamin A/C can interact with euchromatin within the nucleus interior (Gesson et al., 2016), we do not need to rely on such interactions. Moreover, lamin B, which also anchors chromatin, is not found in the interior of the nucleus.

The interior domains are obtained as a result of energetics-driven spontaneous nucleation and subsequently the phase-separation of chromatin. Briefly, the reactions want to drive the chromatin organization to a uniform distribution given as $(\bar{\phi}_h, \bar{\phi}_e)$ given by Eq 3 (shown in Figure 1b). However, this is not energetically favorable, and thus phase-separation of chromatin into the two wells occurs. This requires a nucleation of heterochromatin/euchromatin phase which occurs spontaneously due to the inherent heterogeneities in the active nucleus. Domain nucleation could also be induced due to several reasons such as local nucleosome crowding, or localized concentration of histone methyltransferase, or even lamin A/C in the nucleus interior, as the reviewer suggests. This could be an interesting avenue for future exploration.

Specific Comment 11: Main text & Supplementary Material, „energy penalty associated with forming phase boundaries between the euchromatin and heterochromatin phases“: Might this term be used to simulate various degrees of "chromosome territory intermingling" (see e.g. Pombo et al. 2006). For this, it would be required to specify the "chromatin particles" belonging to a given CT. Generally, it appears to be that in the Chemical Potential theory presented, the existence of chromosome territories (CTs) is neglected. Would it be possible to extend the theory in such a way that individual chromatin particles are assigned to specific CTs (e.g. by assuming that most of the chromatin particles of e.g. #1 form a cluster with a radius of 1 μm in a nuclear volume of radius 5 μm)?

Response: The reviewer is right in saying that the existence of chromosome territories is neglected in the current model. This is because we have considered only two forms in which chromatin can appear i.e., heterochromatin and euchromatin. Thus, our model gives rise to domains of heterochromatin ~ 100 nm radius, as observed in super-resolution images.

The reviewer suggests a very interesting and potentially very useful extension of the model albeit on a different spatial scale. As the reviewer suggests, we'll need to specify an individual mark to the chromatin particles signifying the chromosome to which they belong. Thus, we now have 23×2 species of chromatin which will coalesce to form 23×2 phase-separated chromosome territories. The interface between these territories will not be a sharp, but a smooth interface, with the interface energy parameter determining its width. This parameter could indeed be tuned to obtain an appropriate "intermingling" of chromosome territories.

Further, expanding on the reviewer's idea, within each chromosome territory it could be possible to study the heterochromatin-euchromatin phase-separation to obtain a hierarchical chromatin organization with chromatin domains at $\sim 100\text{nm}$, superseded by chromosome territories at $\sim 1 \mu\text{m}$ length scale. While these are interesting areas to further explore, they are out of the scope of the current manuscript and will be the subject of future studies.

Also we would like to point that increasing the number of chromatin sub-species combinatorically increases the parameters in any computational model (be it the continuum model we present, or a data-driven polymer-based model).

Specific Comment 12: Main text & Supplementary Material: Might a quantitative theory of absolute fluxes (in terms of water volume (nm³) per unit surface area (μm²) be established on the basis of diffusion simulations using experimental absolute density chromatin distributions?

Response: Yes, the flux of water at any point is described in our SI – Section S1.3. Given an experimentally accurate absolute density of chromatin within both the chromatin phases, we can establish the accurate location of wells on the energy density contour plot in Figure 1b. Once the energy landscape is established (we don't need the absolute values of energy measurements), the chemical potential of nucleoplasm is calculated as,

$$\mu_n(\mathbf{x}, t) = \frac{\partial W}{\partial \phi_n} - \nabla \cdot \left(\frac{\partial W}{\partial (\nabla \phi_n)} \right)$$

as shown in Eq S3. Next, the diffusive flux (volume of water flowing across a point per unit surface area per unit time) of nucleoplasm at any point can now be obtained as,

$$\mathbf{j}^n = -M_n \frac{\partial \mu_n}{\partial \mathbf{x}}$$

Note that mobility of water in the nucleus, M_n can be written in terms of diffusivity D_n as $M_n = \frac{D_n \Omega}{k_B T}$, where Ω is the volume of a water molecule, $k_B T$ is the thermodynamic energy scale. Also note that if we write the flux in terms of diffusivity D_n , the energy scale from mobility term cancels out with the energy scale from μ_n . So, the only parameters we actually need are – (a) exact locations of the wells, (b) diffusivity of nucleoplasm in nucleus. Together these can give an estimate of flux of nucleoplasm.

Specific Comment 13: Maintext & Supplementary Material: What would be required to provide time scales also in absolute terms (s)?

Response: All times in our simulation are normalized with respect to the timescale of the reactions – specifically the acetylation of histones. Thus, the timescale for the conversion of a methylated histone into an acetylated histone is what would be required.

We have added a Section S9 to the SI concerning the translation of non-dimensional parameters into physical dimensions. An estimate of these values is provided in Table S3 (Page 26) of SI, as discussed in response to **General Comment 5** by Reviewer 2.

Specific Comment 14: Conversion of Euchromatin to Heterochromatin and vice vs.: This theory of transition of heterochromatin into euchromatin appears to be based on the assumption that the chromatin modelling factors responsible are not substantially impaired by the extremely high density assumed for heterochromatic domains. A possibility to understand this in the context of the theory given would be to assume that the decondensation process starts at the surface of the heterochromatic domains (for a previous qualitative model of this see e.g. Miron et al. 2020). Such a model would also nicely explain the observation of increased condensation of heterochromatic domains: To function on the surface of dense domains, transcription complexes would have to disentangle somewhat the surface chromatin, as shown in Miron et al. (2020), Fig. 6. In the absence of such factors, chromatin-chromatin interactions would result in a “collapse” of these external loops and hence be visible as increased condensation. The model presented by Miron et al. (2020) as well as related qualitative models presented in the literature also suggest a mechanism how the inhibition of transcription may result in an increase of the size of heterochromatic domains.

Response: If the heterochromatic phase is indeed so dense that none of the epigenetic modifiers cannot access the interior of the heterochromatic domains, then the reviewer's idea certainly holds, and the epigenetic modifications would have to initiate at the surface and proceed inwards disentangling the packed chromatin polymer as it progresses.

However, in our model, we do not make this assumption. Epigenetic reactions – both methylation as well as acetylation – can occur globally in our nucleus and are not restricted to the surface. The motivation for transcription-mediated chromatin extrusion being restricted to the domain boundaries comes from the imaging-based observations of exclusion of RNAPII from compacted regions of chromatin (Amiad-Pavlov et al., 2021; Maeshima et al., 2015; Markaki et al., 2010; Miron et al., 2020).

Expanding on the ideas of the reviewer, we find that while RNAPII with weight ~ 550 kD might find it difficult to access the compacted chromatin domains, most epigenetic modifiers such as those belonging to HDAC, KMT and HAT family are small ~ 50 kD. Thus, these modifiers may be able to access the interior of heterochromatin domains. Further, as we pointed out earlier, we do not expect a complete absence of water from the heterochromatin – just that it is water-poor relative to euchromatin due to its tight compaction.

The exact molecular mechanism of the epigenetic modifiers is beyond the scope of the current work. However, in future, it would be interesting to model both (global v/s surface restricted acetylation) mechanisms via our time-dependent phase-field approach and compare the dynamics of chromatin droplet growth/shrinkage with that observed via imaging with a fast temporal resolution.

Specific Comment 15: „Time-dependent mathematical set of governing equations describing the spatio-temporal evolution of chromatin organization in the nucleus“: In the equation presented, the epigenetic regulation appears to be restricted to two factors (for methylation and acetylation). Would it be possible to extend the theory to include also other epigenetic factors, like other DNA/histone modellers, Polycomb proteins etc.?

Response: It is indeed possible, and it is an avenue we have been exploring. However, including the effects of other remodelers requires an inclusion of other stable species of chromatin rather than just hetero- and euchromatin. Despite being more realistic, such a model starts to become parametrically cumbersome very quickly. For instance, considering the role of Polycomb-group proteins, we could now consider two classes of heterochromatin – Polycomb-associated e.g. H3K27me3, and other forms e.g. H3K9me2/3. Further, there could be other stages involving multi-acetylated histones, or even unmarked chromatin. However, adding more stable sub-types of chromatin increases the combinatorial number of kinetic and energetic interaction parameters in the model. That said, given a suitable choice of such parameters and given the kinetic parameters for the chromatin modifying enzymes, the multi-phase separation model could be similarly simulated.

These discussions are useful for understanding the future scope of our model and how its mathematical framework can be extended to understanding more complicated and realistic scenarios. Hence this discussion is added to the SI as a new Section S11 (page 31). The text and figures added as Section S11 are reproduced here:

The model proposed in this paper incorporates three nuclear constituents – nucleoplasm, heterochromatin and euchromatin. These constituents are mixed with each other and form two stable phases – heterochromatin which is compacted and prominently methylated, and euchromatin which is loosely packed and prominently acetylated.

However, euchromatin as well as heterochromatin can have different subtypes depending on location (e.g., lysine site) and extent (e.g., mono-, di- or tri-) of methylation. These post-translational epigenetic variations can induce different functional properties to chromatin. As an example, H3K9me3 is expected to form a core of constitutive heterochromatin, which remains compacted at all stages of development in the cell, and all cell types [36, 37]. On the other hand, H3K27 in its trimethylated form H3K27me3 is a hallmark of Polycomb facultative heterochromatin, which can be reversibly switched between expressive

(H3K27me1/2) or repressive (H3K27me3) forms [36, 37]. The formation of different classes of heterochromatin (constitutive and facultative) can involve different classes of epigenetic enzymes such as methyltransferase SUV39H1 and SUV39H2 for H3K9me3 or EZH2 for H3K27me3 [36].

Figure S16: The key ingredients of a ternary phase field model to capture three stable phases of chromatin – euchromatin, facultative and constitutive heterochromatin.

As a simplified model to capture multiple states of chromatin, let us consider three nuclear constituents – euchromatin (which as a simplification is nucleoplasm rich by definition), constitutive heterochromatin and Polycomb-marked facultative heterochromatin. We consider that the three constituents can mix with different volume fractions to stably form three phases – each phase rich in one of each constituent. The free-energy landscape of this triphasic system can be defined as (analogous to Eq S1) [38],

$$W = \sum_{i=1,2,3} \left[A_i \phi_i^2 (\phi_i^{max} - \phi_i)^2 + \frac{\eta_i}{2} |\nabla \phi_i|^2 \right]$$

where ϕ_i represents the volume fraction of each constituent. The energy coefficients A_i have a scale of $k_B T / \Omega$. The energy landscape is represented on a ternary phase diagram as shown in Figure S16. Note that the energy landscape has three wells, or three minima, corresponding to the three stable phases. The location of these wells can be altered by appropriately choosing the parameters ϕ_i^{max} . For the particular landscape in Figure S16, we have chosen $\phi_i^{max} = 1$. Further, several classes of epigenetic factors such as histone methyltransferases, acetyltransferases, demethylases and deacetylases allow a non-conservative interconversion between these phases, which are captured via the parameters Γ_{ij} denoting the rate of conversion of constituent i into constituent j , for $i, j = 1, 2, 3$, as shown in Figure S16.

The values of the kinetic parameters need to be appropriately chosen. For instance, it could be assumed that the facultative heterochromatin cannot be converted into constitutive heterochromatin directly ($\Gamma_{23} = 0$), although it may happen indirectly via $\Gamma_{21} \times \Gamma_{13}$ pathway.

Note that by incorporating three stable phases, we have now introduced C_2^3 energetic parameters, 3 interfacial energy parameters, and P_2^3 kinetic parameters. As the number of stable phases incorporated increases, the number of energetic and kinetic parameters increases combinatorically. Thus, from a modeling standpoint, it may make economic sense to choose specific stable forms of chromatin as required to reduce the number of parameters. The choice of which stable forms of chromatin are chosen to be modeled would depend on the specific phenomenon being modeled.

Another such choice of stable forms of chromatin could be euchromatin, heterochromatin, and an intermediate unmarked state of chromatin as shown in Figure S17. Note that if the intermediate state of chromatin is unstable and has a very short lifetime, the rate-determining steps are the demethylation and acetyltransferase reactions. Thus, with such an assumption, this multi-phase model simplifies to the model presented in the main manuscript.

Figure S17: Another interpretation of multiple chromatin phases with an unmarked intermediate phase.

Specific Comment 16: „Rescaling the governing equations“: A non-dimensional set of governing equations means that the results are also intrinsically non-dimensional, i.e. provide relations but not absolute figures (e.g. on absolute densities, e.g. in terms of Mbp/ μm^3). What experimentally determined parameters would be required to translate non-dimensional predictions into absolute dimensional ones?

Response: In response to this comment, and **General Comment 5**, we have now added a section in the SI (Section S9, Table S3) detailing the experimentally obtainable parameters required to translate the non-dimensional predictions into those in physical units. Briefly these are:

- (i) Time scale: Requires the rate of acetylation Γ_{ac} . Note that this is the overall timescale of reactions mediated by demethylase and acetyltransferase enzymes. From previous experiments (Waterborg, 2002) we estimate $\Gamma_{ac} \sim 10^{-2} \text{s}^{-1}$.
- (ii) Length scale: All lengths are normalized via the characteristic length $\ell_{RD} = \sqrt{D/\Gamma_a}$ of reaction-diffusion equation (Section S1.5, Section S4). As nucleosomes diffuse in nucleoplasm, their mean-square displacement plots (Nozaki et al., 2023) allow us to extract their diffusivity D giving $\ell_{RD} \sim 300 \text{ nm}$.
- (iii) Energy scale: This will provide the actual values of the energy landscape i.e. $k_B T/\Omega$. Evaluation of the energy values requires knowledge of Ω , i.e. the volume of individual particles of water and chromatin. If this parameter is known, the energy penalty η in Eq S1 can further be evaluated by approximating the width of smooth interface between the heterochromatin and euchromatin. We have now shown this process with approximate estimates in the SI.
- (iv) The density of chromatin in compacted and uncompact state. These correspond to the volume fraction of water in energy wells. As we discussed previously (**General Comment 5**, SI Section S10), this choice does not change our model predictions.

With these four classes of parameters, our model can be fully translated into physical units. We have given a detailed explanation of these parameters, how they can be obtained and an approximation in physical units in the new SI section S9 (Page 26).

Specific Comment 17: „Polymer analogy of the roles played by reaction and diffusion kinetics“: The conservation of the net amount of the DNA of heterochromatin plus euchromatin in the context of the theory presented appears to be evident for a given state of the cell cycle (e.g G1); but what happens during G2? Would it be possible to extend the theory to cell states with replicating DNA? Since in G2 the number of

DNA bp changes, also the terms for chromatin-chromatin interactions and for chromatin-lamin interaction in the equation should be modified. Assuming that the total DNA content remains constant: What is the expected outcome if other molecule numbers change, e.g. of epigenetic diffusing marks (for example methylated or acetylated histone variants and proteins involved): How would this modify the ratio between euchromatin and heterochromatin and their spatial distribution? This would be highly interesting, e.g. with respect to the known change in methylation during ageing. In other words: Would it in principle be possible by this theory to predict age dependent nuclear genome structure?

Response: We agree with the reviewer that the theory in its current form works for interphase cells in G1 phase – where total amount of DNA is unchanging. By the G2 phase, all DNA has been duplicated, but not yet compacted. The DNA replication stage (S phase) requires opening of chromatin, whereas during M phase, all chromatin is compacted into mitotic chromosomes with transcriptional cessation. Thus, it indeed appears that the energy landscape – decided by the strength of the chromatin-chromatin interactions – would be highly dynamic outside the G1 phase.

A discussion on this is beyond the scope of our manuscript and hence is not included so as to not detract from the main focus: *interplay of epigenetic and transcriptional regulation on chromatin organization*.

That said, we can **speculate** a possible way to capture this within our modeling framework, via a modified Flory-Huggins-like energy density:

$$W = \underbrace{\chi \phi^2 (\phi_h^{max} - \phi)^2}_{\text{Entropic Contribution}} + \underbrace{\alpha \phi}_{\text{Compaction Preference}}$$

Here, χ dictates the strength of interaction between the like and unlike phases of chromatin. $\alpha > 0$ denotes the energetic preference for a low-density phase, whereas $\alpha < 0$ denotes the energetic preference for a high-density phase. The following parameters could determine the energy landscape during different phases of cell cycle:

- **G1 Phase:** $\alpha = 0$ – gives the two phases of high and low density (our model).
- **G2 Phase:** $\alpha = 0.5$ – Only a low-density phase is observed, i.e., all chromatin open.
- **M Phase:** $\alpha = -0.5$ – Only a high-density phase is observed, i.e., all chromatin compacted.

Figure R4: Possible constructs of the Flory-Huggins type energy landscape for different phases of the cell cycle. The stable phases are shown as black points.

During the S phase when DNA is dynamically being synthesized, a double-well energy density with varying reaction kinetics could possibly be used to simulate the spatio-temporal DNA duplication and accompanying local chromatin decompaction.

Lastly, the reviewer suggests a scenario where “...other molecule numbers change, e.g. of epigenetic diffusing marks (for example methylated or acetylated histone variants and proteins involved)”. Our model captures the effect of changing molecule number (e.g., epigenetic factors such as HDAC, EZH2, which changes the number of epigenetic marks of each type), via the rate of epigenetic reactions. For instance, TSA treatment inhibits HDAC, reducing the number of HDAC molecules available for histone deacetylation activity. This reduces rate of deacetylation (Γ_{me} , see Figure S2). The change in the overall ratio of acetylation v/s methylation marks due to epigenetic rate change is given (by Eq S11) as,

$$\frac{\bar{\phi}_h}{\bar{\phi}_e} = \frac{\Gamma_{me}}{\Gamma_{ac}}$$

Such framework could “in principle ... predict age dependent nuclear genome structure”, although not without simplifying assumptions which would need to be experimentally motivated.

Specific Comment 18: Schematic Figures: If e.g. in Figure S2 one follows the trajectories of an individual interphase chromosome, one gets the impression that a the enveloping surface of a specific chromosome territory may fill a large part of the nuclear volume, which is in contradiction with the results of experimental observations. To reduce such misleading impressions (often found in biochemical papers on chromatin organisation), the fibers may be just appropriately shortened.

Response: We agree with the reviewer and thank them for pointing out this inconsistency. The schematic was meant as a cartoon representation and did not intend to provide a scaled representation of the domain size. To avoid any such misleading impressions, we have now zoomed into a region of the nucleus rather than showing the entire of it (Figure 1a) or removed the fiber only showing relatively small domains of heterochromatin (Figure S4).

Specific Comment 19: „Conservation of epigenetic factors via Eq (S5) dictates that the total amount of DNA in heterochromatic or euchromatic phase in the nucleus does not change“: This would describe a given transcriptional state of a nucleus. But what happens if gene activity changes ? If by such activation the domains with the sequences to be transcribed have to be made accessible (see e.g. the Model in Miron et al., 2020, or other models, e.g. ANC/INC model, BioEssays 2020), and hence decondensation occurs, THEN the amount of Euchromatin will increase and the amount of heterochromatin will diminish.

Response: Drastic changes to transcription do take place during physiological transformations (e.g., differentiation, reprogramming) or pathological transformations (e.g., aging, disease states like cancer initiation). While the exact mechanisms of how transcriptional states and cell fates change during these transformations are not well understood and are still being explored, to some extent, they involve changes in epigenetic states (e.g., changes in rates and levels of acetylation/methylation) and changes in loop extrusion rates may also potentially be involved. Our model can account for such changes (Section 3.2, Sections S2-S4) and as our model predicts, these changes will change the overall amount of euchromatin and heterochromatin. Similar changes in the total amount of heterochromatin and euchromatin are also predicted upon change in supercoiling-driven chromatin loop extrusion mediated by transcription.

Specific Comments 20: „Relocalization of methylation or acetylation histones without changing the overall amounts of heterochromatin and euchromatin“: This appears to be plausible under the assumption that the number of epigenetic factors remains constant, and all chromatin particles have the same interaction potentials. This is a valid first assumption to predict the general euchromatin-euchromatin distribution. However, how to integrate the fact that cell specific genes are turned on/off in a specific way ? Would it be possible to create instead of one equation a set of coupled differential equations for each set of „particles“ (according to their specific activity status) ?

Response: We would like to point out that the time-dependent evolution of epigenetic markers (see Eq 2b) is written as follows:

$$\frac{\partial \phi_d}{\partial t} = \underbrace{M_d \nabla^2 \mu_d}_{\text{Diffusion of Epigenetic Marks}} + \underbrace{2(\Gamma_{me} \phi_e - \Gamma_{ac} \phi_e)}_{\text{Epigenetic Regulation}} + \text{Active Chromatin Extrusion}$$

The sentence “Relocalization of methylation or acetylation histones without changing the overall amounts of heterochromatin and euchromatin” is from the caption of Figure S3 and is used to only describe the contribution of the diffusion term. This term is indeed a conservative expression and does not change the overall amounts of acetylation or methylation marks. But it is not enough to completely describe the evolution of the said marks. The scenario that “cell specific genes are turned on/off” is intricately important to our model and is captured by the non-conservative epigenetic regulation term (second term) shown in the above equation. This will change the overall amounts of heterochromatin and euchromatin.

Our model differentiates between the activity status of the genes – as described by their inclusion into heterochromatic or euchromatic phase. However, due to the coarse-grained continuum description, we do not distinguish between “specific” genes. That could be possible by considering a multi-phase scenario – with each phase describing a specific gene. However, this could only be done at a much smaller length-scale, not at a meso-scale level of our model. Further, at such a length-scale polymer-based models – which can borrow ingredients (e.g., diffusion and reaction kinetics) from our model – would be much more suitable for specific predictions.

Specific Comments 21: Fig. S2: Using the assumption of very compact heterochromatin domains excluding even water (as indicated in Figure S2a below, then in the schematic depiction of the nuclear DNA density landscape (Fig. S2 abc above), the heterochromatin blocks may graphically better be indicated by homogeneously stained red regions, instead of Spaghetti fibers.

Response: We thank the reviewer for this suggestion. Our adjustment to Fig S2 (now it is Fig S4 in the revised SI on Page 10), in response to **Specific Comment 18**, takes this comment into consideration. As suggested by the reviewer, we have now indicated chromatin in the nucleus as stained red regions rather than spaghetti fibers.

Specific Comments 22: „In addition to methylation and acetylation, transcriptional activity also regulates the sizes of the heterochromatin domains“: Is transcriptional activity regulating the size of the domains; or does the size/structure of the domains regulate transcriptional activity? In other words: Is chromatin nanostructure modification a consequence of the biochemical reactions involved in transcription? Or is the initiation of transcription a consequence of the chromatin nanostructure modification? (e.g. by allowing increased accessibility by decompaction to a target sequence in the interior). In the first case, chromatin structure changes would be just a "down stream effect" produced by otherwise functionally independent biochemical reactions; in the second, the domain nanostructure would play an essential role; chromatin nanostructure would be of fundamental importance for the understanding of transcriptional regulation.

Response: We believe that there is a bidirectional cause and effect, i.e. transcription regulates the size of the domains and at the same time the compacted domains restrict the gene transcription.

Experimental evidence for the role of transcription in regulating the spatial chromatin organization was presented by our co-authors Neguembor et al. (2021). It was shown that transcription-derived supercoiling, in association with the supercoiling alleviation by topoisomerases, closely regulates the formation of chromatin loops, thereby affecting the size of the compacted chromatin domains. Transcription inhibition and modulation of cohesin abundance via WAPL-depletion directly affects the extent of chromatin compaction and its spatial organization.

Conversely, it is known the condensed chromatin domains stay inaccessible for relative larger macromolecules associated with the transcription machinery, thereby restricting the expression of specific genes.

However, while we do think that transcription helps regulate the sizes of heterochromatin domains, domain can form even without it. We see that domains form even in absence of transcription (ActD treatment on multiple cell-lines, Figure 3). Why this happens is theoretically explained via the steps in the phase separation of chromatin delineated in Section 3.2 of our manuscript. Chromatin compaction primarily requires methylation at histone tails (which is also involved in the heterochromatin cross-linking by HP1 α , and lamina interactions). The epigenetic reaction associated with diffusion results in growth of the domains, while acetylation opposes their growth (Figure 2e). Transcription-driven chromatin extrusion adds to this kinetic balance.

Specific Comments 23: List of parameters: Altogether, the parameters used to solve the Chemical Potential equation were chosen to fit the basic experimental expectations. This is of course a valid first approach. The dimension less parameter values indicate that in this way, relative distributions will be obtained. The model would become even more interesting, if it should become possible to introduce

absolute values, e.g. the absolute DNA content, the absolute amount/density of H2B, the absolute number of chromatin-chromatin interactions and chromatin-lamin interactions, the absolute number of epigenetic factors etc. Using appropriately calibrated SMLM approaches, this should become possible at least for some basic parameters, e.g. for the absolute density of eu- and heterochromatin domains (see e.g. a recent analysis in Cell Reports 2023). One might argue that eventually also chromatin structure is regulated by the fundamental laws of Physics and Chemistry. This is of course true. But under evolutionary pressure chromatin nanostructure may have developed in such a way to allow an optimum of transcriptionally useful dynamic chromatin structures.

Recommendation: Nuclear chromatin structure and its relation to gene regulation has emerged as one of the most challenging topics of modern cell biology; for a better understanding of the mechanisms involved, theoretical contributions to this field are highly welcome. Due to the importance of developing theoretical approaches to understand the synergistic action of transcription and epigenetics, a revised version of the ms is recommended for publication.

Response: We agree with the reviewer and appreciate their thought about the role of fundamental physics and chemistry in regulation of chromatin structure. We believe that our model is a step towards unraveling such mechanisms. By referring to various previously published experiments, we have been able to obtain absolute values of several parameters in our model, to allow a more physical interpretation of our predictions. We have added a new section (Section S9) in our SI explaining how such exercise can be carried out. We also list the absolute value of parameters (Table S3), and which experiments are useful for such an exercise.

REVIEWER #3

This manuscript describes a mesoscale whole-nucleus reaction-diffusion model of how chromatin is packed into spatial domains of active and inactive chromatin with various sizes, and how perturbations can alter this organization. The model incorporates: chromatin-chromatin interactions, heterochromatin-lamina interactions, interfacial energy, and kinetic parameters controlling the exchange between heterochromatin and euchromatin. From these, the model generates spatial distributions for heterochromatin, euchromatin, and nucleoplasm and how they evolve over time. The authors compare simulated spatial domain distribution of domains to those obtained experimentally from five cell lines and two imaging technologies, chromSTEM and STORM. Meso-scale biophysical models are of great interest for the interpretation of perturbations on genome folding. In general, the details of this model are nicely presented with both schematics and annotated equations. However, the current presentation is confusing for several major reasons, as well as important missing details.

Major Point 1: First, the proposed model assumes that loop extrusion converts euchromatin (ϕ_e) into heterochromatin (ϕ_h). This is not consistent with descriptions of loop extrusion in previous polymer modeling literature, or in related biology experiments. In previous polymer modeling descriptions of extrusion and phase separation (e.g. Nuebler et al., Conte et al., Salari et al.), the total amount of heterochromatin or euchromatin is not changed by the level of loop extrusion: what is altered is their mixing. That description aligns with experimental results for depletion of cohesin (e.g. Vian et al., Schwarzer et al.), which do not show more inactive chromatin, but instead show a de-mixing of active and inactive chromatin. While the model presented here can make interesting predictions, it will be confusing to use the same words to describe a distinct process. Much clarity could be gained if the authors use a different term to refer to the process modeled throughout the paper (e.g. “active interfacial heterochromatin conversion”). Drawing conclusions about the processes referred to as loop extrusion in the existing literature would require extensive comparative analysis between the models here and previous models in the literature. While such an extensive analysis may be beyond the scope of this paper, clarity and consistency with prior literature should not be.

Response: We thank the reviewer for their positive and encouraging comments. We also welcome the feedback on avoiding the unintended confusion that may arise for a reader due to our presentation. The

reviewer states that ‘model assumes that loop extrusion converts euchromatin (\$\phi_e\$ ) into heterochromatin (\$\phi_h\$ )’. We would like to clarify that the model assumes the *transcription-mediated supercoiling* results in extrusion of chromatin from compacted heterochromatin domains. So, it results in the conversion of heterochromatin to euchromatin and not the other way around. We do agree with the reviewer that increased loop extrusion, as seen in WAPL mutant nuclei (Neguembor et al., 2021), there are longer loops and increased mixing of compartments.

We believe that there are two aspects to the reviewer’s comments:

1. General phenomena of loop extrusion where at a length-scale of TADs or below loops of chromatin are formed – as identified through peaks on HiC/MicroC – can occur anywhere on the genome. The mechanisms of such loop extrusion could be distinct, such as SMC motor activity (Davidson et al., 2019; Kim et al., 2019). Parallel to this, our co-authors (Neguembor et al., 2021) have shown that RNAPII driven transcription can increase DNA supercoiling, in turn mediating the formation of chromatin loops. Importantly, *we do not presume to replace one mechanism with the other*. It may very well be possible that these occur convergently.

Briefly here we describe the mechanism of chromatin extrusion by transcription:

- RNAPII in the nucleus is found to colocalize (Figure R5a) with the low density chromatin phase (Castells-Garcia et al., 2022; Li et al., 2021; Markaki et al., 2010; Miron et al., 2020; Neguembor et al., 2021; Otterstrom et al., 2019; Stasevich et al., 2014). We would like to point out that these observations are consistent on different cell lines – M248 ovarian cancer, mouse C127, human skin fibroblasts and HeLa cells in the cited papers.
- It is well established that RNAPII activity during transcription process results in DNA supercoiling (Liu & Wang, 1987; Naughton et al., 2013).
- Via mathematical modeling, the torsional stress accompanying the supercoiling has been shown to drive extrusion of loops of chromatin fibers through cohesin (Figure R5b, adapted from (Racko et al., 2018)). Further, Neguembor et al. (2021) confirmed experimentally that transcription increases DNA supercoiling, extrudes chromatin and changes its spatial organization.

Figure R5: Contrasting the RNAPII-dependent mechanisms of chromatin extrusion considered in our paper (left) versus the mechanism of extrusion of chromatin loops observed via Hi-C or Micro-C (right). (a) RNAPII typically is found to localize along the periphery of compacted heterochromatin domains (Li et al., 2021). (b) Supercoiling that is associated with transcription has been shown to result in extrusion of loops of DNA (Racko et al., 2018), also see for experimental evidence (Neguembor et al., 2021).

To what extent do the SMC motor activity or other extrusion mechanisms cooperate with the transcription-driven extrusion remains an open question. However, in our work we focus on the ‘*transcription-driven*’ aspect, localized to the heterochromatin domain boundaries. We wholeheartedly agree with the reviewer

that this clarification, while discussed by us in the paper, becomes muddled as we used the ‘same words to describe a distinct process’. To circumvent any possible confusion with the more general ‘loop extrusion’ discussed in the literature, we have changed all occurrences of ‘loop extrusion’ to ‘supercoiling-driven loop extrusion’. To elaborate on this distinction, the following paragraph is added to Section 4 (Page 26):

Beyond transcription induced supercoiling-driven loop extrusion, cohesin itself can play an active role in the formation, extrusion and maintenance of chromatin loops at different physiological length scales. Loops of chromatin, identified as peaks in chromatin contact mapping techniques like Hi-C and Micro-C [9-11, 86] are formed at a length scale of topologically associated domains (TADs) or below. Such loops are observed within both active A [86] and inactive B [92] compartments. The mechanism of extrusion of such loops, could be distinct from RNAPII transcription induced supercoiling-driven loop extrusion, such as cohesin subunit SMC motor activity [20, 93, 94] or a passive cohesin diffusion along the chromatin polymer [95, 96]. These multiple mechanisms of loop formation could be convergently cooperating forming chromatin loops at multiple physiological length scales.

However, here we are specifically interested in the meso-scale roles played by supercoiling-driven extrusion of chromatin loops from the silenced heterochromatin phase into transcriptionally active euchromatin region (Figure 7a).

2. Conversion of heterochromatin into euchromatin. Transcription mediated supercoiling-driven extrusion of chromatin from heterochromatin into euchromatic phase allows an interchange of the two phases, thereby altering the overall extent of chromatin compaction in the nucleus. In agreement, global and local chromatin compaction by various RNAPII inhibitors has been previously reported (Müller et al., 2001; Naughton et al., 2013; Neguembor et al., 2021; Nickerson et al., 1989). This is further confirmed in our manuscript by PWS and ChromSTEM imaging on multiple live and fixed cells.

In our current model, we have only included *two stable forms* (energy wells) of chromatin – euchromatin (less dense, prominently acetylated) and heterochromatin (compacted, prominently methylated). Seeing as increased DNA supercoiling drives extrusion of compacted chromatin into less compacted loops (Neguembor et al., 2021), in our simplified two-well model such extrusion “converts heterochromatin into euchromatin”.

Even with this simplifying assumption our model explains several experimental observations as reported by us (Sections 3.3-3.6), and others in the literature, specifically via contact mapping by Zhang et al. (2023). We will discuss this in more detail in **Major Point 2** next. A possible option to avoid this simplifying assumption is considering the existence of multiple stable states of chromatin. A discussion on how such a model can be approached within our modeling framework is added to the SI, Section S11 (Page 31). Briefly, we can consider additional stable states of chromatin, for instance different forms of heterochromatin (also discussed in response to **Specific Comment 15** by Reviewer 2), or possibly an intermediate unmarked state of chromatin. Unfortunately, adding more stable sub-types of chromatin increases the combinatorial number of kinetic and energetic interaction parameters in the model.

To the Section S11 of the SI (Page 33), the following paragraph is added in response to the reviewer comments:

Another such choice of stable forms of chromatin could be euchromatin, heterochromatin, and an intermediate unmarked state of chromatin as shown in Figure S17. Note that if the intermediate state of chromatin is unstable and has a very short lifetime, the rate-determining steps are the demethylation and acetyltransferase reactions. Thus, with such an assumption, this multi-phase model simplifies to the model presented in the main manuscript.

Major Point 2: The biological motivation behind the assumption that cohesin converts euchromatin into heterochromatin at phase boundaries is also unclear. Recent work (Zhang et al., <https://www.nature.com/articles/s41588-023-01364-4>) shows that RNAPII counteracts extrusion as assayed by Hi-C. Other recent work shows experimental Hi-C data are consistent with cohesin modifying

folding inside the heterochromatin domains, and not just at their edges (<https://www.nature.com/articles/s41594-022-00892-7>). The authors should address such existing literature when motivating their modeling approach and presenting their conclusions.

Response: We thank the reviewer for pointing out these references, which are very useful for motivating our model. We believe that the reviewer’s comments stem from the two points addressed in the first comment. Briefly, we reiterate,

- (a) Supercoiling-mediated chromatin extrusion is a subset of loop extrusion: We agree that ‘cohesin modifying folding’ occurs ‘inside the heterochromatin domains, and not just at their edges’, as is seen via Hi-C. We do not discount this but focus only on transcription-driven extrusion at the edges.
- (b) Chromatin decompaction due to extrusion is captured as the conversion of heterochromatin to euchromatin: Chromatin decompaction due to supercoiling mediated extrusion (Müller et al., 2001; Naughton et al., 2013; Neguembor et al., 2021; Nickerson et al., 1989) is captured in our model as a phase change from compacted heterochromatin into uncompactd euchromatin.

We would like to specifically address the reviewer’s observation that Zhang et al. (2023) “shows that RNAPII counteracts extrusion as assayed by Hi-C”. The Table R1 lists the significant observations by Zhang et al. (2023) and how they compare with our predictions.

Table R1: Comparison of contact mapping observations and model predictions after RNAPII depletion

	Hi-C/Micro-C observation (Zhang et al., 2023)	Predictions from our model
1.	Micro-C reveals that at least 40% of the loops are associated with RNAPII.	Our model pertains to these RNAPII associated loops, not the others.
2.	RNAPII depletion (via AID system) resulted in genome-wide reduction in H3K27ac levels.	Agrees with our prediction via Eq (3), where overall euchromatin content in the nucleus reduces after transcription inhibition.
3.	RNAPII depletion (via AID system) resulted in locally reduced chromatin accessibility.	Agrees with predicted change in local chromatin compaction quantified as domain size (Figure 2,3). Also reported experimentally (Figure 2). Neguembor (2021) earlier quantified the change in ‘chromatin compaction’ due to ActD treatment.
4.	Of the loops that are lost after RNAPII depletion, most frequent are the enhancer anchored loops.	This is consistent with our model. These could correspond to the genes on chromatin extruded from heterochromatin into euchromatin (thus active), but after RNAPII removal, no extrusion results in these chromatin segments becoming heterochromatic. Thus, the genes were upregulated before and downregulated after RNAPII removal.
	These were typically associated with genes which were upregulated before and downregulated after RNAPII depletion.	
	H3K27ac levels around these loops dropped by more than 50%.	
5	The new loops gained/rewired due to RNAPII depletion “typically arose in and around domains with active genes that became silenced upon RNAPII depletion” (Page 833 of the paper for context).	i. Portions of DNA that were active became suppressed after RNAPII depletion. This is totally consistent with our model. ii. They emerge as new loops in B compartment – not associated with RNAPII. Our model does address such non-RNAPII loops.
6	While gained loops are formed in vicinity of H3K27me3 peaks, there is no global increase in methylation.	Our model says that global methylation level would increase. This disagreement is due to the limitation discussed in point (b) above. We address this in the newly added SI Section S11.

Although we may not expect a one-to-one correspondence between our coarse-grained continuum model, spatial super-resolution imaging and the fine-resolution Micro-C predictions, we are still able to explain many local and genome-wide changes due to loss in transcription, which are consistent between all three approaches. To compare our model predictions with chromatin conformation capture techniques, which was missing from our previous manuscript, we have added the following sentences in Section 4, Page 26:

In addition to imaging techniques using multiple modalities, previously reported observations of chromatin reorganization via chromatin conformation capture studies further confirm our model predictions [85, 86]. While the loss of RNAPII only has a very limited effect on the presence of chromatin loops, observed as off diagonal peaks on Hi-C contact maps [87-89], investigations of contact maps at a finer resolution using Micro-C [86, 90, 91] reveals the existence of RNAPII-associated chromatin loops which are indeed disrupted upon transcription inhibition. Specifically, in agreement with our predictions, it was reported that depletion of RNAPII decreased genome-wide histone acetylation (specifically H3K27ac) levels, reduced local chromatin accessibility, and lead to loss of chromatin loops with their downregulation after RNAPII depletion [86]. Further, Hi-C contact maps reveal an increase in chromatin loops in WAPL deficient nuclei [85]. However, it should be noted that not all the loops observed using Hi-C or Micro-C are RNAPII associated.

Point 6 in Table R1 appears to be a point of disagreement between the model and reported observations, in line with the reviewer's comments. This occurs because we considered only two species of chromatin. Further discussion on this is needed for a complete presentation of our model. We have now added Section S11 in our SI (Page 28) which discusses how our model can be extended to capture the presence of multiple stable configurations of chromatin corresponding to the abounding types of post-translational modifications possible on DNA and histone. However, we note that this combinatorically increases the number of parameters, which we would like to avoid in the current model. Furthermore, if these stable phases have a short kinetic lifetime, we will recover our biphasic model.

Major Point 3a: Finally, the authors do not clearly show how their analysis support changes in extrusion versus changes in the rate of acetylation following transcription inhibition. The model's similar dependence on acetylation and active chromatin extrusion (Γ_{ac} and Γ_a equation 2b), raises uncertainty if observed changes following ActD treatment are due to lowered extrusion versus lowered acetylation. Are there differences in distribution of size of modeled heterochromatin domains in these two scenarios?

Response: The reviewer raises an important point about the distinction between the effects of histone acetylation (Γ_{ac}) and supercoiling-driven active chromatin extrusion (Γ_a).

Even within the framework of the current model, although the qualitative effects of both of these are similar, their spatial localization is very different. Acetylation can occur anywhere in the nucleus, but primarily converts heterochromatin inside the domain into euchromatin. Transcription-driven chromatin extrusion

Figure R6: Change in steady state domain radius \tilde{R}_d^{ss} as rate of acetylation Γ_{ac} or rate of chromatin extrusion $\tilde{\Gamma}_a$ increase. Reproduced from Figure S8 added to the SI.

also decompacts chromatin, however requires the presence of RNAPII which is present in the uncompact region. Thus transcription-driven chromatin extrusion decondenses chromatin prominently at the domain boundary. Their quantitative effect on domain size is very different (SI, Eq S15, S16). Roughly, from Eq S14, the radius follows a linearly inverse relation with transcription-driven extrusion, but an inverse square root relation with acetylation. The following paragraph is added to SI, Section S4 (Page 23), where the domain size is theoretically quantified:

Note that although the qualitative effect of increasing acetylation is the same as increasing rate of supercoiling-drive chromatin extrusion, the quantitative ways in which their effects are felt are different (different scales in Eq S13, S14). This is immediately visible from Eq S18 and S16'. With $\tilde{\Gamma}_{me} = \Gamma_{me}/\Gamma_{ac}$, the domain radius scales as, $\tilde{R}_d^{ss} \sim \sqrt{\frac{\tilde{\Gamma}_{me}}{1+\tilde{\Gamma}_{me}}}$. On the other hand, the scaling of domain radius with $\tilde{\Gamma}_a$ is more complex, but if the effect of acetylation is ignored, from Eq S14, we see that $\tilde{R}_d^{ss} \sim \frac{1}{\tilde{\Gamma}_a}$. This difference in scales can be exemplified by graphically seeing the change in \tilde{R}_d^{ss} as these rates change, as shown in Figure S8. We see that the effect of acetylation is more pronounced than the effect of chromatin extrusion. However, this result is derived theoretically for a single domain. Since a single domain has only one continuous boundary where transcription occurs, this scaling holds. In a numerical simulation, there are many boundaries where transcription will drive chromatin extrusion, and thus change the scaling derived theoretically. The numerical effect of changing epigenetic reaction rate, and extrusion rate was discussed in Section S2, Figure S5. Theoretically this can be captured by increasing the value of κ in Eq S18. The parameter κ , as discussed in Section S3, includes, amongst other effects, the role of multiple domain boundaries.

Further we would like to clarify that all treatments shown in our manuscript, at least directly, only *alter the process of chromatin extrusion*. Specifically, ActD treatment inhibits RNAPII while WAPL Δ reduced cohesin unloading and increases extrusion. Similarly, in our modeling results, we have made a point to ensure that the reaction rate of epigenetic-driven acetylation Γ_{ac} is not altered in any simulation throughout the study. This ensures that, in correspondence with the experiments, all changes occurring to the chromatin organization are purely due to the transcription mediated supercoiling-driven chromatin extrusion. We had previously stressed (Section 3.3, Page 13):

The choice of the parameters for rates of acetylation $\tilde{\Gamma}_{ac}$, methylation $\tilde{\Gamma}_{me}$, and the strength of chromatin-lamina interactions \tilde{V}_L , were held constant for all the following simulations, ...

To the methods section (Section 2.4, Page 8), we had added the following sentence to further stress this implementation:

Note that the epigenetic rates $\tilde{\Gamma}_{me}$ and the strength of chromatin-lamina affinity \tilde{V}_L are not modified throughout any of the simulations carried out, unless explicitly stated. This is to ensure that any predicted changes in chromatin organization occur specifically due to change in supercoiling-driven chromatin loop extrusion.

Major Point 3b: To draw firm conclusions, it would be informative to decrease the expected activity of extrusion (e.g. via RAD21 depletion) and then inhibiting transcription. Such model inquiries and experiments may be beyond the scope of this study, but without it the authors should heavily revise how their conclusions are presented.

Response: We thank the reviewer for their suggestion of comparing the effects of inhibition of cohesin activity (via RAD21 depletion) and transcription. We would like to point out that a very similar inquiry was performed by Neguembor et al. (2021), the co-authors of the current manuscript. We describe this briefly here.

Figure R7: Confocal images of SMC3 in WAPL Δ nuclei (a) without SMC3 kd and normal transcription, (b) with SMC3 kd, (c) with only transcription inhibited, and (d) with transcription inhibition and SMC3 kd (Reproduction of Figure S6D Neguembor et al 2021). Quantified in Figure 5G, Neguembor et al. (2021).

Cohesin in HeLa nuclei with WAPL mutation was altered via a partial knockdown of SMC1A and SMC3 subunits via shRNA lentiviral transduction. This was followed by an inhibition of transcription via ActD treatment. The confocal images of SMC3 distribution (see Figure S6D in (Neguembor et al., 2021)) thus obtained for the four cases are reproduced in Figure R7. Note that WAPL deficiency results in formation of vermicelli-like structures of cohesin indicating accumulation of cohesin on DNA and increased loop extrusion (Figure R7a). Even if cohesin levels are partially altered via SMC3kd, vermicelli form at a rate comparable to non-silencing (NS) shRNA controls (Figure 5G, Neguembor et al 2021) (Figure R7b). However, upon transcription inhibition vermicelli formation is strongly reduced in cells with normal or reduced cohesin level (NS or SMC3 kd respectively) (Figure 5G Neguembor et al 2021) (Figure R7c,d). Thus, reduced cohesin level is not sufficient per se to promote chromatin compaction or loss of vermicelli formation.

This experiment offers evidence that at least a partial loss of cohesin does not significantly change the role that transcription is playing in the supercoiling-driven extrusion of chromatin. If we wished to account for the contribution of cohesin motor activity on the transcription-driven extrusion, we would modify the active extrusion rate Γ_a as,

$$\Gamma_a = \left(\underbrace{\Gamma_{tr}^{active}}_{\text{supercoiling driven extrusion by RNAPII}} + \underbrace{\Gamma_{coh}^{active}}_{\text{Cohesin motor driven extrusion}} \right) \times \underbrace{(\Gamma_l - \Gamma_{ul})}_{\text{Abundance of cohesin (effect of WAPL and NIPBL)}}$$

However, considering the above experimental observations we would expect that $\Gamma_{coh}^{active} \ll \Gamma_{tr}^{active}$, at least under partial knock-down of cohesin. This then reduces to Eq 6 of our main text.

However, note that this observation *does not imply that cohesin's transcription independent activity has no role in chromatin organization*. Indeed, even a partial loss of SMC3 affects the spatial chromatin organization observed via STORM imaging promoting partial chromatin decompaction (refer Figure 5E (Neguembor et al., 2021)). While we are focusing on the supercoiling-driven extrusion only at the domain boundaries, explaining this observation requires understanding and modeling the global effects of cohesin in loop extrusion, and is beyond the scope of this work.

Key Details:

- as numerical solution of the reaction-diffusion PDEs are central to this manuscript, code and software used should be specified in the manuscript and supplement.

Response: We thank the reviewer for pointing this out. The PDEs required to model the chromatin organization are mentioned in the SI and the manuscript. Further, we converted the PDEs into weak-form equations suitable for finite-element solver implementation. We then used COMSOL Multiphysics with a

'Weak Form PDE' module for the solution of the equations. We have now taken care to specify these details (Section S1.5, Page 9 of SI):

For the purposes of numerical implementation, these equations are converted into weak-form suitable for a suitable for finite-element solver implementation. We then used COMSOL Multiphysics with a 'Weak Form PDE' module for the solution of the equations. The boundary conditions, and the list of non-dimensional parameters used in our model are discussed in detail in Section S8.

- the reporting summary has a link to a GitHub repository, but the repository does not provide an explanation of how the files could be used. This repository should include a sample dataset of how an image is processed. An example simulated image would be helpful as well. Image data can be stored on OSF, Zenodo, or elsewhere.

Response: We have updated the repository (https://github.com/ShenoyLab/STORM_Analysis). We have now added a sample dataset (in the folder "Input_LocsLib") along with the expected image output (folder: "Output_VoronoiDensityPlot"). We have also added a 'README' file delineating the implementation of the Matlab code.

- it would be helpful to justify the exponential decay used to model the chromatin-lamina interaction.

Response: The schematic in Figure R8 shows the chromatin-lamina interactions which may be mediated by a host of proteins such as LAP2 β , emerin, LBR etc. (Luperchio et al., 2014; Zullo et al., 2012). Since these interactions are mediated by proteins anchored on the lamina, we take an interaction strength that decays exponentially away from the lamina. Note that the exact decay characteristics do not really affect the model qualitatively. We choose exponential form as a generalized minimal-parameter decay. The decay parameter d_0 is a characteristic length scale of the exponential decay and is dependent on the strength of interactions and molecular structure of the anchoring proteins. We have added the above schematic and the following paragraph to the SI (Section S1.2, Page 3):

Figure R8: Schematic representation of the energetic interactions between chromatin and the lamina mediated by anchoring proteins like LAP2 β .

Since these interactions are mediated by proteins anchored on the lamina, we take an interaction strength that is most robust at the nuclear periphery (distance from lamina $d = 0$) and decays exponentially away from the lamina over a length scale d_0 (schematically shown in Figure S1). Note that the exact decay characteristics do not really affect the model qualitatively. We choose exponential form as a generalized minimal-parameter decay.

- it is important to indicate the resolution in nanometers at which they model chromatin or solve their equations.

Response: Since we have a coarse-grained continuum model, unlike for a polymer model, a 'resolution' is not defined. However, we can estimate an approximate scale of the characteristic lengths in our model. We have added a list of non-dimensional parameters (Section S9, Table S3) and what their real-world values correspond to.

At a spatial scale, we have non-dimensionalized all lengths with respect to the reaction-diffusion length scale $\ell_{RD} \sim \sqrt{D/\Gamma_{ac}}$. By estimating the diffusion and reaction parameters (Table S3), we estimate ℓ_{RD} to be approximately 300 nm. Physically, this is the order of the spacing between the heterochromatin domains,

under the assumptions held *only for analytical scaling*, i.e. domains are far enough to not interact with each other.

We could define the resolution of the continuum model as the smallest length-scale that emerges, which is the width of the smooth domain boundaries ℓ_{int} in our model. As shown in SI Section S1.5, this emerges from the competition between the interfacial energy and the bulk mixing energy. We estimate $\ell_{int} \sim 30\text{-}50$ nm from comparison with ChromSTEM imaging (Table S3, SI).

- while experimental data appears to display a broad range of domain sizes, the model appears to predict domains with a characteristic size. How can this discrepancy be resolved? If it cannot with the existing approach, the authors should mention this as a limitation in the discussion.

Response: This is an important observation by the reviewer. Despite seeing a comparatively broad range of domains sizes, we do observe a characteristic length scale from the experiments too. That is how we quantify the effects of transcription. That said, the deviation from this mean characteristic length scale is indeed much larger in experiments than in the model. This is because, in the model, we have assumed a uniform (with Gaussian noise) spatial abundance of epigenetic factors such as HDAC, HATs etc. We have kept the standard deviation of the Gaussian noise at $\sim 20\%$ of the mean.

For a more realistic spatiotemporal distribution of the reaction rates $\Gamma_{ac}(x, t)$ and $\Gamma_{me}(x, t)$, we should ideally solve a reaction diffusion equation for the nucleocytoplasmic shuttling of epigenetic factors such as HDAC. Another option could be to introduce a spatiotemporally correlated thermodynamic noise for the distribution of these parameters with parameters trained on multiple control STORM images.

Although, these steps would increase the visual similarity between our simulation results and experimental images, it does not affect the fundamental scaling relations, as well as RNAPII/WAPL inhibition results we have derived.

- the distribution of chromatin is not necessarily at steady-state. Can the authors provide evidence that this is a reasonable simplifying assumption?

Response: We agree that the distribution of chromatin does not attain a steady state. That is because of a continuous presence of spatiotemporal activity in a live cell, as well as a spatiotemporal thermodynamic noise – as is evident from the vast literature on molecular dynamic simulations of chromatin as a heteropolymer. Due to these local fluctuations and persistent dynamics, locally the domains are never ‘static’, although in an average sense – over the entire nucleus – they should still exhibit a characteristic domain size. Indeed, we observe a characteristic chromatin domain size from live cell (PWS) as well as fixed cell (STORM, ChromSTEM) imaging. Further the local dynamics explain the high standard-deviations in the experimental domain size measurements – as discussed in the previous comment.

- if Γ_{tr} is small but non-zero, then Γ_a for combined WAPL and ActD should be smaller than ActD alone, and domain sizes may be further increased. The authors should discuss how their conclusions would differ if this is the case.

Response: The reviewer raises an important question about an intermediate value of Γ_{tr} , i.e. between unhindered transcription and switched-off transcription. As we show below, in such a scenario the size of the domains would decrease and not increase. Recapitulating the relevant equation (Eq 6):

$$\Gamma_a = \underbrace{\Gamma_{tr}}_{\text{active supercoiling-mediated extrusion rate}} \times \underbrace{\Gamma_{coh}}_{\text{Rate of cohesin loading-unloading}}$$

After ActD treatment, we choose $\Gamma_{tr} = 0$. However, now consider the case when Γ_{tr} is small, but non-zero. Say, $\Gamma_{tr} \sim 0.01$. As a result of WAPL Δ , there is more cohesin abundance on chromatin such that $\Gamma_{coh}^{WAPL\Delta} > \Gamma_{coh}^{control}$. As a corollary, we can also say that $\Gamma_{coh}^{WAPL\Delta+ActD} > \Gamma_{coh}^{ActD}$ because ActD treatment does not affect this parameter (at least not directly, in the current model).

WAPLΔ + ActD

$$\Gamma_a^{WAPL\Delta+ActD} = 0.01 \times \Gamma_{coh}^{WAPL\Delta+ActD}$$

ActD

$$\Gamma_a^{ActD} = 0.01 \times \Gamma_{coh}^{ActD}$$

Thus, $\Gamma_a^{WAPL\Delta+ActD}$ is going to be slightly greater than Γ_a^{ActD} . When extrusion rate is greater, the size of the heterochromatin domain decreases. Thus, size of domains after combined WAPLΔ and ActD treatment would be smaller than the size of the domains for ActD treatment alone.

Physically this makes complete sense. Γ_{tr} small but non-zero corresponds to the case when some transcription-driven supercoiling activity is still happening, albeit to a lesser extent. WAPLΔ makes sure that less cohesin is unloaded and therefore associates more with chromatin than the case of ActD alone. Thus, if some supercoiling occurs, there is more extrusion than in the case of ActD alone. Thus, size of domains goes down with WAPLΔ. The maximal domain size is attained only as $\Gamma_a \rightarrow 0$. The domain size cannot increase beyond this limit, given that all other parameters are unchanged.

Minor Details:

- 2.4: Unclear what authors mean by: "However, locally, the diffusion kinetics can result in exchange of epigenetic marks – either acetylation or methylation – between neighboring nucleosomes, as shown in the inset in Figure S1. ". Is it really the nucleosomes that are changing from diffusion or the spatial density / positions of the two types of marks?

Response: We understand that the explanation of diffusion of epigenetic marks was not very clear in our model description. To explain the 'diffusion of epigenetic marks' we have now adopted a more descriptive approach in the SI. We discuss the mechanistic and theoretical details to justify the role played by the conservative evolution of epigenetic marks.

We begin by explaining the role of epigenetic reactions in interconverting the acetylation and methylation marks on the histones, thereby interconverting the heterochromatin and euchromatin phases. However, the interconversion of heterochromatin and euchromatin must also be favorable energetically. This energetic contribution takes into account the epigenetic marks on the nucleosomes surrounding the reaction site. For instance, methylation of a nucleosome is energetically more favorable if the surrounding nucleosomes are all methylated, than when the surrounding nucleosomes are prominently acetylated.

Figure R9: Schematic description of 'diffusion of epigenetic marks'.

We give a schematic evolution via Figure S4d (reproduced here as Figure R9). Consider the initial configuration shown in Figure R9(i). If acetylation reaction occurs, the configuration shown

in Figure R9(ii) has the highest probability of being attained, since it maximizes the like-like nucleosome interactions. Similarly, if methylation occurs next, the configuration shown in Figure R9(iii) is most likely to be attained. Hence, effectively, we see that a conservative diffusion of epigenetic marks occurs if the neighborhood dependence of reaction kinetics is incorporated – which occurs due to the chromatin-chromatin interactions. Such conservative evolution leads to the coarsening of heterochromatin and euchromatin phases.

The text added to the main manuscript (Page 6-7) and SI (Section S1.6, Pages 9-11) were reproduced in response to the **Point 3** raised by Reviewer 1.

Lastly, we also provide theoretical evidence of the conservative diffusion-like evolution of epigenetic marks. As in our mechanistic and schematic explanation, we only constrain the reactions to be neighborhood dependent, so that like-marked nucleosomes interact with a higher binding energy (thereby lowering free energy of the nucleus) than unlike-marked nucleosomes. We find that the resulting evolution of epigenetic marks closely follows the Eq 2b in the main text. To this end, we have added a new section (Section S1.7, Page 11) to the extended methods. To avoid reproducing the entire section here we give a brief summary -

We consider chromatin to be a 1D polymer of acetylated and methylated nucleosomes corresponding to euchromatin and heterochromatin phases respectively. We consider methylation and acetylation reactions, which respectively convert euchromatin to heterochromatin and vice versa. Lastly, we introduce a neighborhood dependence, such that the rates of acetylation (or equivalently methylation) increase by $\Delta\Gamma$ when one of the neighbors are epigenetically marked similarly to the nucleosome after reaction. The resulting evolution kinetics (Eq S8.3) additively decouples naturally into a reaction-like non-conservative evolution and a diffusion-like conservative evolution – which is equivalent to the ‘diffusion of epigenetic marks’ in our model.

- 2.4: the spatial dependence (x) in $\Gamma_a(x)$, is easy to miss, and it would be useful to emphasize that this makes this term distinct from the epigenetic regulation term in the first sentence after equation 2b.

Response: We agree with the reviewer. We have now added more details on the motivation for the spatial localization of supercoiling-mediated chromatin extrusion and choice of $\Gamma_a(x)$ in the main manuscript. The difference between transcriptional and epigenetic regulation is now clearer by explicitly writing the form of $\Gamma_a(x)$ in Eq 2b' and 2b (Page 7, Section 2.4). The text added to the manuscript was reproduced in response to **Specific Comment 4** by Reviewer 2.

- Page 12 line 12: instead of figure 2f it should be figure 2g; S11: gamma_a is in the legend, but gamma_me is in the panel.

Response: We are grateful to the reviewer for pointing out these mistakes. They have been corrected.

- Figure 4b-e: It is not clear how the blue line in experiments, which stays quite high, is analogous to the blue line in simulations. It is also not clear why (e) indicates euchromatin rather than total DNA (heterochromatin + euchromatin) as seems to be plotted for experiments (caption for (a)).

Response: The comparison between the experiments and simulation for Figure 4 is more qualitative in nature. The blue line in Figure 4c is the plot of DNA density over two heterochromatin domains, such that even between the domains, there is the presence of low-density DNA. This shows that the euchromatin phase (between the heterochromatin domains) in control has more DNA than the euchromatin phase in ActD treated nuclei. In simple words, DNA is more diffused in control and compacted in ActD nucleus (similar to quantification by Neguembor et al., 2021). Similarly, in the model (Figure 4d, 4e) we see that in control conditions, between the domains there is a DNA presence, which is higher than in the control conditions.

The plots (d) and (e) for the model still plots the total DNA ($\phi_h + \phi_e$). The dashed line shows the level of total DNA in the euchromatin phase in DMSO (blue) and ActD (red). In Figure 4(e) we have changed the label to clarify the meaning of the dashed lines and to the caption we add the following sentence:

The black dashed line shows the level of total DNA predicted in the euchromatin phase of DMSO and ActD treated nuclei.

REFERENCES

Amiad-Pavlov, D., Lorber, D., Bajpai, G., Reuveny, A., Roncato, F., Alon, R., Safran, S., & Volk, T. (2021). Live imaging of chromatin distribution reveals novel principles of nuclear architecture and chromatin compartmentalization. *Science advances*, 7(23), eabf6251.

- Castells-Garcia, A., Ed-Daoui, I., González-Almela, E., Vicario, C., Ottestrom, J., Lakadamyali, M., Neguembor, M. V., & Cosma, M. P. (2022). Super resolution microscopy reveals how elongating RNA polymerase II and nascent RNA interact with nucleosome clutches. *Nucleic acids research*, *50*(1), 175-190.
- Chen, L.-Q. (2002). Phase-field models for microstructure evolution. *Annual review of materials research*, *32*(1), 113-140.
- Consortium, I. H. G. S. (2004). Finishing the euchromatic sequence of the human genome. *Nature*, *431*(7011), 931-945.
- Davidson, I. F., Bauer, B., Goetz, D., Tang, W., Wutz, G., & Peters, J.-M. (2019). DNA loop extrusion by human cohesin. *Science*, *366*(6471), 1338-1345.
- De Gennes, P.-G. (1979). *Scaling concepts in polymer physics*. Cornell university press.
- Di Pierro, M., Cheng, R. R., Lieberman Aiden, E., Wolynes, P. G., & Onuchic, J. N. (2017). De novo prediction of human chromosome structures: Epigenetic marking patterns encode genome architecture. *Proceedings of the National Academy of Sciences*, *114*(46), 12126-12131.
- Feng, W., & Michaels, S. D. (2015). Accessing the inaccessible: the organization, transcription, replication, and repair of heterochromatin in plants. *Annual Review of Genetics*, *49*, 439-459.
- Gesson, K., Rescheneder, P., Skoruppa, M. P., von Haeseler, A., Dechat, T., & Foisner, R. (2016). A-type lamins bind both hetero-and euchromatin, the latter being regulated by lamina-associated polypeptide 2 alpha. *Genome research*, *26*(4), 462-473.
- Gómez-García, P. A., Portillo-Ledesma, S., Neguembor, M. V., Pesaresi, M., Oweis, W., Rohrlich, T., Wieser, S., Meshorer, E., Schlick, T., & Cosma, M. P. (2021). Mesoscale modeling and single-nucleosome tracking reveal remodeling of clutch folding and dynamics in stem cell differentiation. *Cell reports*, *34*(2).
- Katava, M., Shi, G., & Thirumalai, D. (2022). Chromatin dynamics controls epigenetic domain formation. *Biophysical Journal*, *121*(15), 2895-2905.
- Kim, K., Roy, A., Gururajan, M., Wolverton, C., & Voorhees, P. (2017). First-principles/Phase-field modeling of θ' precipitation in Al-Cu alloys. *Acta Materialia*, *140*, 344-354.
- Kim, Y., Shi, Z., Zhang, H., Finkelstein, I. J., & Yu, H. (2019). Human cohesin compacts DNA by loop extrusion. *Science*, *366*(6471), 1345-1349.
- Li, Y., Eshein, A., Virk, R. K., Eid, A., Wu, W., Frederick, J., VanDerway, D., Gladstein, S., Huang, K., & Shim, A. R. (2021). Nanoscale chromatin imaging and analysis platform bridges 4D chromatin organization with molecular function. *Science advances*, *7*(1), eabe4310.
- Liu, L. F., & Wang, J. C. (1987). Supercoiling of the DNA template during transcription. *Proceedings of the National Academy of Sciences*, *84*(20), 7024-7027.
- Luperchio, T. R., Wong, X., & Reddy, K. L. (2014). Genome regulation at the peripheral zone: lamina associated domains in development and disease. *Current opinion in genetics & development*, *25*, 50-61.
- Maeshima, K., Kaizu, K., Tamura, S., Nozaki, T., Kokubo, T., & Takahashi, K. (2015). The physical size of transcription factors is key to transcriptional regulation in chromatin domains. *Journal of Physics: Condensed Matter*, *27*(6), 064116.
- Markaki, Y., Gunkel, M., Schermelleh, L., Beichmanis, S., Neumann, J., Heidemann, M., Leonhardt, H., Eick, D., Cremer, C., & Cremer, T. (2010). Functional nuclear organization of transcription and DNA replication a topographical marriage between chromatin domains and the interchromatin compartment. Cold Spring Harbor symposia on quantitative biology,
- Michieletto, D., Orlandini, E., & Marenduzzo, D. (2016). Polymer model with epigenetic recoloring reveals a pathway for the de novo establishment and 3D organization of chromatin domains. *Physical Review X*, *6*(4), 041047.
- Miron, E., Oldenkamp, R., Brown, J. M., Pinto, D. M., Xu, C. S., Faria, A. R., Shaban, H. A., Rhodes, J. D., Innocent, C., & De Ornellas, S. (2020). Chromatin arranges in chains of mesoscale domains with nanoscale functional topography independent of cohesin. *Science advances*, *6*(39), eaba8811.
- Müller, W. G., Walker, D., Hager, G. L., & McNally, J. G. (2001). Large-scale chromatin decondensation and recondensation regulated by transcription from a natural promoter. *The Journal of cell biology*, *154*(1), 33-48.
- Naughton, C., Avlonitis, N., Corless, S., Prendergast, J. G., Mati, I. K., Eijk, P. P., Cockroft, S. L., Bradley, M., Ylstra, B., & Gilbert, N. (2013). Transcription forms and remodels supercoiling domains unfolding large-scale chromatin structures. *Nature structural & molecular biology*, *20*(3), 387-395.
- Neguembor, M. V., Martin, L., Castells-García, Á., Gómez-García, P. A., Vicario, C., Carnevali, D., Abed, J. A., Granados, A., Sebastian-Perez, R., & Sottile, F. (2021). Transcription-mediated supercoiling regulates genome folding and loop formation. *Molecular Cell*, *81*(15), 3065-3081. e3012.
- Nickerson, J. A., Krochmalnic, G., Wan, K. M., & Penman, S. (1989). Chromatin architecture and nuclear RNA. *Proceedings of the National Academy of Sciences*, *86*(1), 177-181.
- Nozaki, T., Imai, R., Tanbo, M., Nagashima, R., Tamura, S., Tani, T., Joti, Y., Tomita, M., Hibino, K., & Kanemaki, M. T. (2017). Dynamic organization of chromatin domains revealed by super-resolution live-cell imaging. *Molecular Cell*, *67*(2), 282-293. e287.
- Nozaki, T., Shinkai, S., Ide, S., Higashi, K., Tamura, S., Shimazoe, M. A., Nakagawa, M., Suzuki, Y., Okada, Y., & Sasai, M. (2023). Condensed but liquid-like domain organization of active chromatin regions in living human cells. *Science advances*, *9*(14), eadf1488.

- Otterstrom, J., Castells-Garcia, A., Vicario, C., Gomez-Garcia, P. A., Cosma, M. P., & Lakadamyali, M. (2019). Super-resolution microscopy reveals how histone tail acetylation affects DNA compaction within nucleosomes in vivo. *Nucleic acids research*, *47*(16), 8470-8484.
- Paulsen, J., Sekelja, M., Oldenburg, A. R., Barateau, A., Briand, N., Delbarre, E., Shah, A., Sørensen, A. L., Vigouroux, C., & Buendia, B. (2017). Chrom3D: three-dimensional genome modeling from Hi-C and nuclear lamin-genome contacts. *Genome biology*, *18*, 1-15.
- Qi, Y., & Zhang, B. (2019). Predicting three-dimensional genome organization with chromatin states. *PLoS computational biology*, *15*(6), e1007024.
- Racko, D., Benedetti, F., Dorier, J., & Stasiak, A. (2018). Transcription-induced supercoiling as the driving force of chromatin loop extrusion during formation of TADs in interphase chromosomes. *Nucleic acids research*, *46*(4), 1648-1660.
- Ramanarayan, H., & Abinandanan, T. (2003). Phase field study of grain boundary effects on spinodal decomposition. *Acta Materialia*, *51*(16), 4761-4772.
- Rouquette, J., Genoud, C., Vazquez-Nin, G. H., Kraus, B., Cremer, T., & Fakan, S. (2009). Revealing the high-resolution three-dimensional network of chromatin and interchromatin space: a novel electron-microscopic approach to reconstructing nuclear architecture. *Chromosome research*, *17*, 801-810.
- Sanulli, S., Gross, J. D., & Narlikar, G. J. (2019). Biophysical properties of HP1-mediated heterochromatin. Cold Spring Harbor symposia on quantitative biology.
- Shi, G., Liu, L., Hyeon, C., & Thirumalai, D. (2018). Interphase human chromosome exhibits out of equilibrium glassy dynamics. *Nature communications*, *9*(1), 3161.
- Stasevich, T. J., Sato, Y., Nozaki, N., & Kimura, H. (2014). Quantifying histone and RNA polymerase II post-translational modification dynamics in mother and daughter cells. *Methods*, *70*(2-3), 77-88.
- Tanaka, H. (2000). Viscoelastic phase separation. *Journal of Physics: Condensed Matter*, *12*(15), R207.
- Vinayak, V., Basir, R., Sant'Anna, L., & Shenoy, V. (2023a). Polymer model predicts history dependent epigenetic and lamin-associated domain sizes. *Bulletin of the American Physical Society*.
- Vinayak, V., Basir, R., Sant'Anna, L., & Shenoy, V. B. (2023b). Polymer model predicts history dependent epigenetic and lamin-associated domain sizes. *Biophysical Journal*, *122*(3), 306a-307a.
- Waterborg, J. H. (2002). Dynamics of histone acetylation in vivo. A function for acetylation turnover? *Biochemistry and cell biology*, *80*(3), 363-378.
- Zhang, S., Übelmesser, N., Barbieri, M., & Papantonis, A. (2023). Enhancer-promoter contact formation requires RNAPII and antagonizes loop extrusion. *Nature Genetics*, *55*(5), 832-840.
- Zullo, J. M., Demarco, I. A., Piqué-Regi, R., Gaffney, D. J., Epstein, C. B., Spooner, C. J., Luperchio, T. R., Bernstein, B. E., Pritchard, J. K., & Reddy, K. L. (2012). DNA sequence-dependent compartmentalization and silencing of chromatin at the nuclear lamina. *Cell*, *149*(7), 1474-1487.

REVIEWERS' COMMENTS

Reviewer #1 (Remarks to the Author):

I appreciate the efforts made by the authors to take all my previous concerns seriously and thereby revising their manuscript. I have no objection to the authors' responses to my queries, and I find the modifications made on the manuscript satisfactory. However, I have another minor concern which should be taken care of before publication.

The authors assured my understanding that only the chromatin-chromatin interactions energetics has been used to plot Fig. 1b. I therefore recommend to omit the \tilde{W} symbols from Fig. 1b as it creates unnecessary confusion with Eq. 1 and the definition of the chemical potentials in col. 200. Also, I recommend to omit the citation to Fig. 1b from col 272 as the definition of the chemical potentials considers all the terms on the right-hand-side of Eq. 1 (as understood from Eqs. S1 and S3 in SI).

I would like to express another personal opinion on the referencing style used by the authors in the SI. See for example, eqs. numbered as S1a and S1b between eqs. S3 and S4. The consecutive eqs. on page 8 and 9 are numbered as S1', S3', and S8'. These unusual style of numbering unnecessarily distracted me from the interesting scientific endeavor they have presented in their texts. Of course, the authors need not to oblige to this personal opinion of the reviewer, at least for the current manuscript.

Reviewer #2 (Remarks to the Author):

Thanks to the modifications made by the authors, the ms has substantially gained in clarity, and I have no objections to its speedy publication.

Only one last suggestion to the authors:

In various comments (e.g. 7, 23) the problem of absolute density measurements as an important parameter in heterochromatin dependent transcription regulation has been expressed: "...Using appropriately calibrated SMLM approaches, this should become possible at least for some basic parameters, e.g. for the absolute density of eu-and heterochromatin domains (see e.g. a recent analysis in Cell Reports 2023)."

In their response, the authors reply

"However absolute values (in terms of Mbp/ μm^3) of such parameters would be better obtained via experimental observations and molecular dynamic simulations (e.g. single particle tracking in (Gómez-García et al., 2021; Nozaki et al., 2017)) and could then be used as input parameters (such as ϕ_{hmax}) in our model."

The two papers mentioned contain indeed valuable information about DNA densities.

While Nozaki et al. (2017) appears not to contain absolute DNA density (Mbp/ μm^3) estimates, such data might be obtained from Fig. 2 in Gomez-Garcia et al. (2021), under the assumption that a) the abscissa refers to nm² instead of the " μm^2 " stated there; and b) that the SMLM microscope was used in the astigmatic mode to allow an optical section thickness of 0.1 μm ; then these data may be converted to absolute density estimates up to 50 Mbp/ μm^3 , i.e. a density which according to Maeshima et al. (2015) should be sufficient for a substantial reduction of accessibility.

The existence of heterochromatic domains with equally high (and for small domains sometimes even higher) absolute DNA densities was also confirmed by a recent paper in (Gelleri et al., Cell Reports 42, 112567, June 27, 2023) where the entire nuclear landscape was mapped in terms of Mbp/ μm^3 . This

further supports some basic conclusions of the theoretical modelling approach presented in the revised ms. It is recommended to add this paper (and additional related ones, if already existing at the necessary level of nanoscale resolution) to the references.

We again sincerely thank the referees for their careful review of the manuscript, and their constructive feedback on it. We have now addressed the last remaining concerns raised by the referees below. (The reviewer comments are in blue, our response in plain text and the edits in the manuscript are highlighted in yellow.)

Reviewer #1

I appreciate the efforts made by the authors to take all my previous concerns seriously and thereby revising their manuscript. I have no objection to the authors' responses to my queries, and I find the modifications made on the manuscript satisfactory. However, I have another minor concern which should be taken care of before publication.

The authors assured my understanding that only the chromatin-chromatin interactions energetics has been used to plot Fig. 1b. I therefore recommend to omit the \tilde{W} symbols from Fig. 1b as it creates unnecessary confusion with Eq. 1 and the definition of the chemical potentials in col. 200. Also, I recommend to omit the citation to Fig. 1b from col 272 as the definition of the chemical potentials considers all the terms on the right-hand-side of Eq. 1 (as understood from Eqs. S1 and S3 in SI).

Response: We would like to thank the reviewer for their valuable feedback. Following their suggestion, we have removed the \tilde{W} symbol from Fig 1b and have only kept the chromatin-chromatin interaction energy expression. Further, as suggested by the reviewer, we have omitted the citation to Fig 1b from col 272.

I would like to express another personal opinion on the referencing style used by the authors in the SI. See for example, eqs. numbered as S1a and S1b between eqs. S3 and S4. The consecutive eqs. on page 8 and 9 are numbered as S1', S3', and S8'. These unusual style of numbering unnecessarily distracted me from the interesting scientific endeavor they have presented in their texts. Of course, the authors need not to oblige to this personal opinion of the reviewer, at least for the current manuscript.

Response: With the different referencing style, we intended to demonstrate the relation between Eq S1 and Eq S1', and so on. The 'prime' equations are normalized versions. However, as the reviewer suggests, the referencing style can be distracting, making the equations harder to find. We have revised the equation numbers to be serial throughout the SI.

Reviewer #2

Thanks to the modifications made by the authors, the ms has substantially gained in clarity, and I have no objections to its speedy publication.

Only one last suggestion to the authors:

In various comments (e.g. 7, 23) the problem of absolute density measurements as an important parameter in heterochromatin dependent transcription regulation has been expressed: "...Using appropriately calibrated SMLM approaches, this should become possible at least for some basic parameters, e.g. for the absolute density of eu-and heterochromatin domains (see e.g. a recent analysis in Cell Reports 2023)."

In their response, the authors reply

"However absolute values (in terms of Mbp/ μm^3) of such parameters would be better obtained via experimental observations and molecular dynamic simulations (e.g. single particle tracking in (Gómez-García et al., 2021; Nozaki et al., 2017)) and could then be used as input parameters (such as ϕ_{hmax}) in

our model.”

The two papers mentioned contain indeed valuable information about DNA densities.

While Nozaki et al. (2017) appears not to contain absolute DNA density (Mbp/ μm^3) estimates, such data might be obtained from Fig. 2 in Gomez-Garcia et al. (2021), under the assumption that a) the abscissa refers to nm^2 instead of the “ μm^2 ” stated there; and b) that the SMLM microscope was used in the astigmatic mode to allow an optical section thickness of 0.1 μm ; then these data may be converted to absolute density estimates up to 50 Mbp/ μm^3 , i.e. a density which according to Maeshima et al. (2015) should be sufficient for a substantial reduction of accessibility.

The existence of heterochromatic domains with equally high (and for small domains sometimes even higher) absolute DNA densities was also confirmed by a recent paper in (Gelleri et al., Cell Reports 42, 112567, June 27, 2023) where the entire nuclear landscape was mapped in terms of Mbp/ μm^3 . This further supports some basic conclusions of the theoretical modelling approach presented in the revised ms. It is recommended to add this paper (and additional related ones, if already existing at the necessary level of nanoscale resolution) to the references.

Response: We are very grateful to the reviewer for their feedback and the new citation suggested. The exact spatial distribution of DNA densities observed in the referred paper would be very helpful in choosing the model parameter ϕ_h^{max} . The following sentence is added to the SI (Page 31):

Single molecule localization microscopy (SMLM), which allows visualization of individual nucleosome clusters combined with super-resolution fluctuation-assisted binding-activated localization microscopy (fBALM) has recently emerged as a promising avenue towards extracting the nucleus-wide 3D spatial DNA density (in Mbp/ μm^3) [35].